# EQUIVARIANT SUBGRAPH AGGREGATION NETWORKS

**Beatrice Bevilacqua**[*]
Purdue University
bbevilac@purdue.edu

**Fabrizio Frasca**[*]
Imperial College London & Twitter
ffrasca@twitter.com

**Derek Lim**[*]
MIT CSAIL
dereklim@mit.edu

**Balasubramaniam Srinivasan**
Purdue University
bsriniv@purdue.edu

**Chen Cai**
UCSD CSE
c1cai@ucsd.edu

**Gopinath Balamurugan**
University of Tuebingen
gbm0998@gmail.com

**Michael M. Bronstein**
Imperial College London & Twitter
mbronstein@twitter.com

**Haggai Maron**
NVIDIA Research
hmaron@nvidia.com

## ABSTRACT

Message-passing neural networks (MPNNs) are the leading architecture for deep learning on graph-structured data, in large part due to their simplicity and scalability. Unfortunately, it was shown that these architectures are limited in their expressive power. This paper proposes a novel framework called Equivariant Subgraph Aggregation Networks (ESAN) to address this issue. Our main observation is that while two graphs may not be distinguishable by an MPNN, they often contain distinguishable subgraphs. Thus, we propose to represent each graph as a set of subgraphs derived by some predefined policy, and to process it using a suitable equivariant architecture. We develop novel variants of the 1-dimensional Weisfeiler-Leman (1-WL) test for graph isomorphism, and prove lower bounds on the expressiveness of ESAN in terms of these new WL variants. We further prove that our approach increases the expressive power of both MPNNs and more expressive architectures. Moreover, we provide theoretical results that describe how design choices such as the subgraph selection policy and equivariant neural architecture affect our architecture's expressive power. To deal with the increased computational cost, we propose a subgraph sampling scheme, which can be viewed as a stochastic version of our framework. A comprehensive set of experiments on real and synthetic datasets demonstrates that our framework improves the expressive power and overall performance of popular GNN architectures.

## 1 INTRODUCTION

Owing to their scalability and simplicity, Message-Passing Neural Networks (MPNNs) are the leading Graph Neural Network (GNN) architecture for deep learning on graph-structured data. However, Morris et al. (2019); Xu et al. (2019) have shown that these architectures are at most as expressive as the Weisfeiler-Lehman (WL) graph isomorphism test (Weisfeiler & Leman, 1968). As a consequence, MPNNs cannot distinguish between very simple graphs (See Figure 1). In light of this limitation, a question naturally arises: **is it possible to improve the expressiveness of MPNNs?**

Several recent works have proposed more powerful architectures. One of the main approaches involves higher-order GNNs equivalent to the hierarchy of the $k$-WL tests (Morris et al., 2019; 2020b; Maron et al., 2019b;a; Keriven & Peyré, 2019; Azizian & Lelarge, 2021; Geerts, 2020), offering a trade-off between expressivity and space-time complexity. Unfortunately, it is already difficult to implement 3rd order networks (which offer the expressive power of 3-WL). Alternative approaches use standard MPNNs enriched with structural encodings for nodes and edges (e.g. based on cycle or clique counting) (Bouritsas et al., 2022; Thiede et al., 2021) or lift graphs into simplicial- (Bodnar et al., 2021b) or cell complexes (Bodnar et al., 2021a), extending the message passing mechanism to these higher-order structures. Both types of approaches require a precomputation stage that, though reasonable in practice, might be expensive in the worst case.

---

[*]Equal contribution, authors are in alphabetical order.

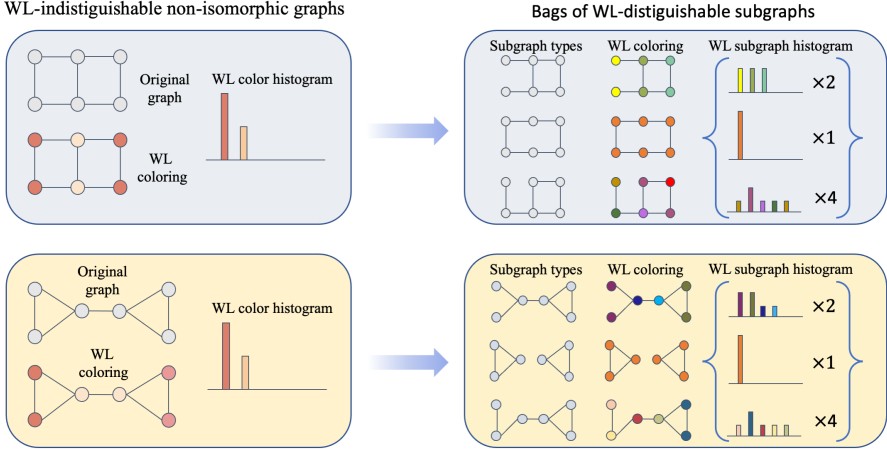

Figure 1: We present a provably expressive graph learning framework based on representing graphs as bags of subgraphs and processing them with an equivariant architecture composed of GNNs and set networks. **Left:** A pair of graphs not distinguishable by the WL test. **Right:** The corresponding bags (multisets) of edge-deleted subgraphs, which can be distinguished by our framework.

**Our approach.** In an effort to devise simple, intuitive and more flexible provably expressive graph architectures, we develop a novel framework, dubbed *Equivariant Subgraph Aggregation Networks* (ESAN), to enhance the expressive power of existing GNNs. Our solution emerges from the observation that while two graphs may not be distinguishable by an MPNN, it may be easy to find distinguishable subgraphs. More generally, instead of encoding multisets of *node* colors as done in MPNNs and the WL test, we opt for encoding bags (multisets) of subgraphs and show that such an encoding can lead to a better expressive power. Following that observation, we advocate representing each graph as a *bag of subgraphs* chosen according to some predefined policy, e.g., all graphs that can be obtained by removing one edge from the original graph. Figure 1 illustrates this idea.

Bags of subgraphs are highly structured objects whose symmetry arises from both the structure of each constituent graph as well as the multiset on the whole. We propose an equivariant architecture specifically tailored to capture this object's symmetry group. Specifically, we first formulate the symmetry group for a set of graphs as the direct product of the symmetry groups for sets and graphs. We then construct a neural network comprising layers that are equivariant to this group. Motivated by Maron et al. (2020), these layers employ two *base graph encoders* as subroutines: The first encoder implements a Siamese network processing each subgraph independently; The second acts as an information sharing module by processing the aggregation of the subgraphs. After being processed by several such layers, a set learning module aggregates the obtained subgraph representations into an invariant representation of the original graph that is used in downstream tasks.

An integral component of our method, with major impacts on its complexity and expressivity, is the subgraph selection policy: a function that maps a graph to a bag of subgraphs, which is then processed by our equivariant neural network. In this paper, we explore four simple — yet powerful — subgraph selection policies: node-deleted subgraphs, edge-deleted subgraphs, and two variants of ego-networks. To alleviate the possible computational burden, we also introduce an efficient stochastic version of our method implemented by random sampling of subgraphs according to the aforementioned policies.

We provide a thorough theoretical analysis of our approach. We first prove that our architecture can implement novel and provably stronger variants of the well-known WL test, capable of encoding the multiset of subgraphs according to the base graph encoder (e.g., WL for MPNNs). Furthermore, we study how the expressive power of our architecture depends on different main design choices like the underlying base graph encoder or the subgraph selection policy. Notably, we prove that our framework can separate 3-WL indistinguishable graphs using only a 1-WL graph encoder, and that it can enhance the expressive power of stronger architectures such as PPGN (Maron et al., 2019a).

We then present empirical results on a wide range of synthetic and real datasets, using several existing GNNs as base encoders. Firstly, we study the expressive power of our approach using the synthetic datasets introduced by Abboud et al. (2020) and show that it achieves perfect accuracy.

We then evaluate ESAN on popular graph benchmarks and show that they consistently outperform their base GNN architectures, and perform better or on-par with state of the art methods.

**Main contributions.** This paper offers the following main contributions: (1) A general formulation of learning on graphs as learning on bags of subgraphs; (2) An equivariant framework for generating and processing bags of subgraphs; (3) A comprehensive theoretical analysis of the proposed framework, subgraph selection policies, and their expressive power; (4) An in-depth experimental evaluation of the proposed approach, showing noticeable improvements on real and synthetic data. We believe that our approach is a step forward in the development of simple and provably powerful GNN architectures, and hope that it will inspire both theoretical and practical future research efforts.

## 2 EQUIVARIANT SUBGRAPH AGGREGATION NETWORKS (ESAN)

In this section, we introduce the ESAN framework. It consists of (1) Neural network architectures for processing bags of subgraphs (DSS-GNN and DS-GNN), and (2) Subgraph selection policies. We refer the reader to Appendix A for a brief introduction to GNNs, set learning, and equivariance.

**Setup and overview.** We assume a standard graph classification/regression setting.[1] We represent a graph with $n$ nodes as a tuple $G = (A, X)$ where $A \in \mathbb{R}^{n \times n}$ is the graph's adjacency matrix and $X \in \mathbb{R}^{n \times d}$ is the node feature matrix. The main idea behind our approach is to represent the graph $G$ as a *bag* (multiset) $S_G = \{\!\{G_1, \ldots, G_m\}\!\}$ of its subgraphs, and to make a prediction on a graph based on this subset, namely $F(G) := F(S_G)$. Two crucial questions pertain to this approach: (1) Which architecture should be used to process bags of graphs, i.e., *How to define $F(S_G)$?*, and (2) Which subgraph selection policy should be used, i.e., *How to define $S_G$?*

### 2.1 BAG-OF-GRAPHS ENCODER ARCHITECTURE

To address the first question, we start from the natural symmetry group of a bag of graphs. We first devise an equivariant architecture called DSS-GNN based on this group, and then propose a particularly interesting variant (DS-GNN) obtained by disabling one of the components in DSS-GNN.

**Symmetry group for sets of graphs.** The bag $S_G = \{\!\{G_1, \ldots, G_m\}\!\}$ of subgraphs of $G$ can be represented as tensor $(\mathcal{A}, \mathcal{X}) \in \mathbb{R}^{n \times n \times m} \times \mathbb{R}^{n \times d \times m}$, assuming the number of nodes $n$ and the number of subgraphs $m$ are fixed. Here, $\mathcal{A} \in \mathbb{R}^{n \times n \times m}$ represents a set of $m$ adjacency matrices, and $\mathcal{X} \in \mathbb{R}^{n \times d \times m}$ represents a set of $m$ node feature matrices. Since the order of nodes in each subgraph, as well as the order of these subgraphs in $S_G$, are arbitrary, our architecture must be equivariant to changes of both of these orders. *Subgraph symmetries*, or node permutations, are represented by the symmetric group $S_n$ that acts on a graph via $(\sigma \cdot A)_{ij} = A_{\sigma^{-1}(i)\sigma^{-1}(j)}$ and $(\sigma \cdot X)_{il} = X_{\sigma^{-1}(i)l}$. *Set symmetries* are the permutations of the subgraphs in $S_G$, represented by the symmetric group $S_m$ that acts on sets of graphs via $(\tau \cdot \mathcal{A})_{ijk} = \mathcal{A}_{ij\tau^{-1}(k)}$ and $(\tau \cdot \mathcal{X})_{ilk} = \mathcal{X}_{il\tau^{-1}(k)}$. These two groups can be combined using a direct product into a single group $H = S_m \times S_n$ that acts on $(\mathcal{A}, \mathcal{X})$ in the following way:

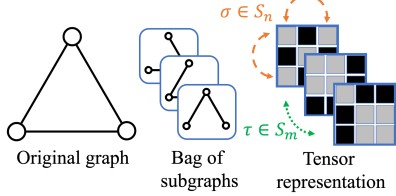

$$((\tau, \sigma) \cdot \mathcal{A})_{ijk} = \mathcal{A}_{\sigma^{-1}(i)\sigma^{-1}(j)\tau^{-1}(k)}, \quad ((\tau, \sigma) \cdot \mathcal{X})_{ilk} = \mathcal{X}_{\sigma^{-1}(i)l\tau^{-1}(k)}$$

Figure 2: The symmetry structure of a bag of subgraphs, in this case the set of all $m = 3$ edge-deleted subgraphs. This set of subgraphs is represented as an $m \times n \times n$ tensor $\mathcal{A}$ (and additional node features that are not illustrated here). $(\tau, \sigma) \in S_m \times S_n$ acts on the tensor $\mathcal{A}$ by permuting the subgraphs ($\tau$) and the nodes in the subgraphs ($\sigma$), which are assumed to be ordered consistently.

In other words, $\tau$ permutes the subgraphs in the set and $\sigma$ permutes the nodes in the subgraphs, as depicted in Figure 2. Importantly, since all the subgraphs originate from the same graph, we can order their nodes consistently throughout, i.e., the $i$-th node in all subgraphs represents the $i$-th node in the original graph.[2] This setting can be seen as a particular instance of the DSS framework (Maron et al., 2020) applied to a bag of graphs.

---

[1] The setup can be easily changed to support node and edge prediction tasks.

[2] This consistency assumption justifies the fact that we apply the same permutation $\sigma$ to all subgraphs, rather than having a different node permutation $(\sigma_1, \ldots, \sigma_m)$ for each subgraph.

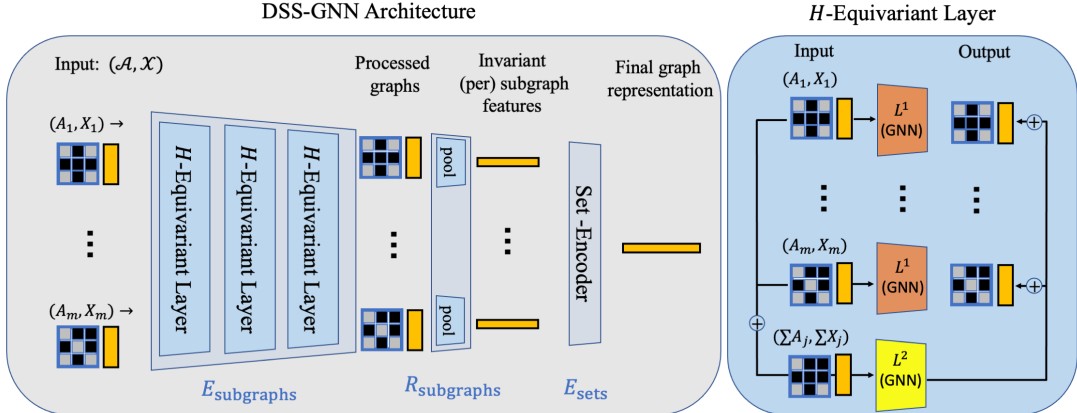

Figure 3: DSS-GNN layers and architecture. **Left panel**: the DSS- GNN architecture is composed of three blocks: a Feature Encoder, a Readout Layer and a Set Encoder. **Right panel**: a DSS-GNN layer is constructed from a Siamese part (orange) and an information-sharing part (yellow).

$H$**-equivariant layers.** Our goal is to propose an $H$-equivariant architecture that can process bags of subgraphs accounting for their natural symmetry (the product of node and subgraph permutations). The main building blocks of such equivariant architectures are $H$-equivariant layers. Unfortunately, characterizing the set of all equivariant functions can be a difficult task, and most works limit themselves to characterizing the spaces of *linear* equivariant layers. In our case, Maron et al. (2020) characterized the space of linear equivariant layers for sets of symmetric elements (such as graphs) with a permutation symmetry group $P$ ($S_n$ in our case). Their study shows that each such layer is composed of a Siamese component applying a linear $P$-equivariant layer to each set element independently and an information sharing component applying a different linear $P$-equivariant layer to the aggregation of all the elements in the set (see Appendix A).

Motivated by the linear characterization, we adopt the same layer structure with Siamese & information sharing components, and parameterize these ones using any equivariant GNN layer such as MPNNs. These layers $L : \mathbb{R}^{n \times n \times m} \times \mathbb{R}^{n \times d \times m} \to \mathbb{R}^{n \times n \times m} \times \mathbb{R}^{n \times d' \times m}$ map bags of subgraphs to bags of subgraphs, as follows:

$$(L(\mathcal{A}, \mathcal{X}))_i = L^1(A_i, X_i) + L^2 \left( \sum_{j=1}^{m} A_j, \sum_{j=1}^{m} X_j \right). \tag{1}$$

Here, $A_j, X_j$ are the adjacency and feature matrices of the $j$-th subgraph (the $j$-th components of the tensors $\mathcal{A}, \mathcal{X}$, respectively), and $(L(\mathcal{A}, \mathcal{X}))_i$ denotes the output of the layer on the $i$-th subgraph. $L^1, L^2 : \mathbb{R}^{n \times n} \times \mathbb{R}^{n \times d} \to \mathbb{R}^{n \times n} \times \mathbb{R}^{n \times d'}$ represent two graph encoders and can be any type of GNN layer. We refer to $L^1, L^2$ as the base graph encoders. When MPNN encoders parameterise $L^1, L^2$, the adjacency matrices of the subgraphs do not change, i.e., the $H$-equivariant layer outputs the processed node features with the same adjacency matrix. These layers can also transform the adjacency structure of the subgraphs, e.g., when using Maron et al. (2019a) as the base encoder.

While the encoder $L^1$ acts on every subgraph independently, $L^2$ allows the sharing of information across all the subgraphs in $S_G$. In particular, the $L^2$ module operates on the sum of adjacency and feature matrices across the bag. This is a meaningful operation because the nodes in all the subgraphs of a particular graph are consistently ordered. As in the original DSS paper, this sum aggregator for the adjacency and feature matrices ($\sum_{j=1}^{m} A_j$ and $\sum_{j=1}^{m} X_j$) can be replaced with (1) Sums that exclude the current subgraph ($\sum_{j \neq i}^{m} A_j$ and $\sum_{j \neq i}^{m} X_j$) or with (2) Other aggregators such as max and mean. We note that for the subgraph selection policies we consider in Subsection 2.2, an entrywise max aggregation over the subgraph adjacency matrices, $A = \max_j A_j$, recovers the original graph connectivity. This is a convenient choice in practice and we use it throughout the paper. Finally, the $H$-equivariance of the layer follows directly from the $S_n$-equivariance of the base-encoder, and the $S_m$-equivariance of the aggregation. Figure 3 (right panel) illustrates our suggested $H$-equivariant layer.

**DSS-GNN** comprises three components:

$$F_{\text{DSS-GNN}} = E_{\text{sets}} \circ R_{\text{subgraphs}} \circ E_{\text{subgraphs}} \tag{2}$$

The first component is an *Equivariant Feature Encoder*, $E_{\text{subgraphs}} : \mathbb{R}^{n \times n \times m} \times \mathbb{R}^{n \times d \times m} \to \mathbb{R}^{n \times n \times m} \times \mathbb{R}^{n \times d' \times m}$ implemented as a composition of several $H$-equivariant layers (Equation 1).

The purpose of the subgraph encoder is to learn useful node features for all the nodes in all subgraphs. On top of the graph encoder, we apply a *Subgraph Readout Layer* $R_{\text{subgraphs}} : \mathbb{R}^{n \times n \times m} \times \mathbb{R}^{n \times d' \times m} \to \mathbb{R}^{d' \times m}$ that generates an invariant feature vector for each subgraph independently by aggregating the graph node (and/or edge) data, for example by summing the node features. The last layer, $E_{\text{sets}} : \mathbb{R}^{d' \times m} \to \mathbb{R}^{d''}$, is a universal *Set Encoder*, for example DeepSets (Zaheer et al., 2017) or PointNet (Qi et al., 2017). Figure 3 (left panel) illustrates the DSS-GNN architecture.

Intuitively, DSS-GNN encodes the subgraphs (while taking into account the other subgraphs) into a set of $S_n$-invariant representations ($E_{\text{subgraphs}} \circ R_{\text{subgraphs}}$) and then encodes the resulting set of subgraph representations into a single $H$-invariant representation $E_{\text{sets}}$ of the original graph $G$.

**DS-GNN** is a variant of DSS-GNN obtained by disabling the information sharing component, i.e., by setting $L^2 = 0$ in Equation 1 for each layer in $E_{\text{subgraphs}}$. As a result, the subgraphs are encoded independently, and $E_{\text{subgraphs}}$ is effectively a Siamese network. DS-GNN can be interpreted as a Siamese network that encodes each subgraph independently, followed by a set encoder that encodes the set of subgraph representations. DS-GNN is invariant to a larger symmetry group obtained from the *wreath product* $S_n \wr S_m$.[3]

In Section 3, we will show that both DS-GNN and DSS-GNN are powerful architectures and that in certain cases DSS-GNN has superior expressive power.

## 2.2 SUBGRAPH SELECTION POLICIES

The second question, *How to select the subgraphs?*, has a direct effect on the expressive power of our architecture. Here, we discuss subgraph selection policies and present simple but powerful schemes that will be analyzed in Section 3.

Let $\mathcal{G}$ be the set of all graphs with $n$ nodes or less, and let $\mathbb{P}(\mathcal{G})$ be its power set, i.e. the set of all subsets $S \subseteq \mathcal{G}$. A subgraph selection policy is a function $\pi : \mathcal{G} \to \mathbb{P}(\mathcal{G})$ that assigns to each graph $G$ a subset of the set of its subgraphs $\pi(G)$. We require that $\pi$ is invariant to permutations of the nodes in the graphs, namely that $\pi(G) = \pi(\sigma \cdot G)$, where $\sigma \in S_n$ is a node permutation, and $\sigma \cdot G$ is the graph obtained after applying $\sigma$. We will use the following notation: $\pi(G) := S_G^\pi$. As before, $S_G^\pi$ can be represented in tensor form, where we stack subgraphs in arbitrary order while making sure that the order of the nodes in the subgraphs is consistent.

In this paper, we explore four simple subgraph selection policies that prove to strike a good balance between complexity (the number of subgraphs) and the resulting expressive power: node-deleted subgraphs (ND), edge-deleted subgraphs (ED), and ego-networks (EGO, EGO+), as described next. In the **node-deleted policy**, a graph is mapped to the set containing all subgraphs that can be obtained from the original graph by removing a single node.[4] Similarly, the **edge-deleted policy** is defined by removing a single edge. The **ego-networks policy** EGO maps each graph to a set of ego-networks of some specified depth, one for each node in the graph (a $k$-Ego-network of a node is its $k$-hop neighbourhood with the induced connectivity). We also consider a variant of the ego-networks policy where the root node holds an identifying feature (EGO+). For DS-GNN, in order to guarantee at least the expressive power of the basic graph encoder used, one can add the original graph $G$ to $S_G^\pi$ for any policy $\pi$, obtaining the augmented version $\hat{\pi}$. For example, $\widehat{\text{EGO}}$ is a new policy that outputs for each graph the standard EGO policy described above and the original graph.

**Stochastic sampling.** For larger graphs, using the entire bag of subgraphs for some policies can be expensive. To counteract this, we sometimes employ a stochastic subgraph sub-sampling scheme which, in every training epoch, selects a small subset of subgraphs $\overline{S}_G^\pi \subset S_G^\pi$ independently and uniformly at random. In practice, we use $|\overline{S}_G^\pi|/|S_G^\pi| \in \{0.05, 0.2, 0.5\}$. Besides enabling running on larger graphs, sub-sampling also allows larger batch sizes, resulting in a faster runtime per epoch. During inference, we combine $\ell$ such subsets of subgraphs (we use $\ell = 5$) and a majority voting scheme to make predictions. While the resulting model is no longer invariant, we show that this efficient variant of our method increases the expressive power of the base encoder and performs well empirically.

---

[3]The wreath product models a setup in which the subgraphs are not aligned, see Maron et al. (2020); Wang et al. (2020) for further discussion.

[4]For all policies, node removal is implemented by removing all the edges to the node, so that all the subgraphs are kept aligned and have $n$ nodes.

## 2.3 RELATED WORK

Broadly speaking, the efforts in enhancing the expressive power of GNNs can be clustered along three main directions: (1) Aligning to the k-WL hierarchy (Morris et al., 2019; 2020b; Maron et al., 2019b;a); (2) Augmenting node features with exogenous identifiers (Sato et al., 2021; Dasoulas et al., 2020; Abboud et al., 2020); (3) Leveraging on structural information that cannot provably be captured by the WL test (Bouritsas et al., 2022; Thiede et al., 2021; de Haan et al., 2020; Bodnar et al., 2021b;a). Our approach falls into the third category. It describes an architecture that uncovers discriminative structural information within subgraphs of the original graph, while preserving equivariance – typically lacking in (2) – and locality and sparsity of computation – generally absent in (1). Prominent works in category (3) count or list substructures from fixed, predefined banks. This preprocessing step has an intractable worst-case complexity and the choice of substructures is generally task dependent. In contrast, as we show in Section 3, our framework is provably expressive with scalable and non-domain specific policies. Lastly, ESAN can be combined with these related methods, as they can parameterize the $E_{\text{subgraphs}}$ module whenever the application domain suggests so. We refer readers to Appendix B for a more detailed comparison of our work to relevant previous work, and to Appendix F for an in-depth analysis of its computational complexity.

## 3 THEORETICAL ANALYSIS

In this section, we study the expressive power of our architecture by its ability to provably separate non-isomorphic graphs. We start by introducing WL analogues for ESAN and then study how different design choices affect the expressive power of our architectures.

### 3.1 A WL ANALOGUE FOR ESAN

We introduce two color-refinement procedures inspired by the WL isomorphism test (Weisfeiler & Leman, 1968). These variants, which we refer to as DSS-WL and DS-WL variants, are inspired by DSS-GNN and DS-GNN with a 1-WL equivalent base graph encoder (e.g., an MPNN), accordingly. The goal is to formalize our intuition that graphs may be more easily separated by properly encoding (a subset of) their contained subgraphs. Our variants are designed in a way to extract and leverage this discriminative source of information. Details and proofs are provided in Appendix D.

*(Init.)* Inputs to the DSS-WL test are two input graphs $G^1, G^2$ and a subgraph selection policy $\pi$. The algorithm generates the subgraph bags over the input graph by applying $\pi$. If initial colors are provided, each node in each subgraph is assigned its original label in the original graph. Otherwise, an initial, constant color is assigned to each node in each subgraph, independent of the bag.

*(Refinement)* On subgraph $S$, the color of node $v$ is refined according to the rule: $c_{v,S}^{t+1} \leftarrow \text{HASH}(c_{v,S}^t, N_{v,S}^t, C_v^t, M_v^t)$. Here, $N_{v,S}^t$ denotes the multiset of colors in $v$'s neighborhood over subgraph $S$; $C_v^t$ represents the multiset of $v$'s colors across subgraphs; $M_v^t$ is the multiset collecting these cross-bag aggregated colors for $v$'s neighbors according to the original graph connectivity.

*(Termination)* At any step, each subgraph $S$ is associated with color $c_S$, representing the multiset of node colors therein. Input graphs are described by the multisets of subgraph colors in the corresponding bag. The algorithm terminates as soon as the graph representations diverge, deeming the inputs non-isomorphic. The test is inconclusive if the two graphs are assigned the same representations at convergence, i.e., when the color refinement procedure converges on each subgraph.

DS-WL is obtained as a special setting of DSS-WL whereby the refinement rule neglects inputs $C_v^t$ and $M_v^t$. These inputs account for information sharing across subgraphs, and by disabling them, the procedure runs the standard WL on each subgraph independently. We note that since DS-WL is a special instantiation of DSS-WL, we expect the latter to be as powerful as the former; we formally prove this in Appendix D. Finally, both DSS- and DS-WL variants reduce to trivial equivalents of the standard WL test with policy $\pi : G \mapsto \{G\}$.

### 3.2 WL ANALOGUES AND EXPRESSIVE POWER

Our first result confirms our intuition on the discriminative information contained in subgraphs:

**Theorem 1** (DS(S)-WL strictly more powerful than 1-WL)**.** *There exist selection policies such that DS(S)-WL is* strictly more powerful *than 1-WL in distinguishing between non-isomorphic graphs.*

The idea is that it is possible to find policies[5] such that (i) our variants refine WL, i.e. any graph pair distinguished by WL is also distinguished by our variants, and (ii) there exist pairs of graphs indistinguishable by WL but separated by our variants. These exemplary pairs include graphs from a specific family, i.e. Circulant Skip Link (CSL) graphs (Murphy et al., 2019; Dwivedi et al., 2020). We show that DS-WL and DSS-WL can distinguish pairs of these graphs:

**Lemma 1.** $\text{CSL}(n, 2)$ *can be distinguished from any* $\text{CSL}(n, k)$ *with* $k \in [3, n/2 - 1]$ *by DS-WL and DSS-WL with either the* ND, EGO, *or* EGO+ *policy.*

It is possible to also find exemplary pairs for the ED policy: one is included is Figure 1, while another is found in Appendix D. Further examples to the expressive power attainable with our framework are reported in the next subsection. Next, the following result guarantees that the expressive power of our variants is attainable by our proposed architectures.

**Theorem 2** (DS(S)-GNN at least as powerful as DS(S)-WL; DS-GNN at most as powerful as DS-WL)**.** *Let* $\mathcal{F}$ *be any family of bounded-sized graphs endowed with node labels from a finite set. There exist selection policies such that, for any two graphs* $G^1, G^2$ *in* $\mathcal{F}$*, distinguished by DS(S)-WL, there is a DS(S)-GNN model in the form of Equation* (2) *assigning* $G^1, G^2$ *distinct representations. Also, DS-GNN with MPNN base graph encoder is at most as powerful as DS-WL.*

In particular, this theorem applies to policies such that each edge in the original graph appears at least once in the bag. This is the case of the policies in Section 2.2 and their augmented versions. The theorem states that, under the aforementioned assumptions, DSS-GNNs (DS-GNNs) are at least as powerful as the DSS-WL (DS-WL) test, and transitively, in view of Theorem 1, that there exist policies making ESAN more powerful than the WL test. This result is important, because it gives provable guarantees on the representation power of our architecture w.r.t. standard MPNNs:

**Corollary 1** (DS(S)-GNN is strictly more powerful than MPNNs)**.** *There exists subgraph selection policies such that DSS-GNN and DS-GNN architectures are strictly more powerful than standard MPNN models in distinguishing non-isomorphic graphs.*

The corollary is an immediate consequence of Theorem 1, Theorem 2 and the results in Xu et al. (2019); Morris et al. (2019), according to which MPNNs are at most as powerful as the WL test.

### 3.3 THEORETICAL EXPRESSIVENESS OF DESIGN CHOICES

Our general ESAN architecture has several important design choices: choice of DSS-GNN or DS-GNN for the architecture, choice of graph encoder, and choice of subgraph selection policy. Thus, here we study how these design choices affect expressiveness of our architecture.

**DSS vs. DS matters.** To continue the discussion of the last section, we show that DSS-GNN is at least as powerful as DS-GNN, and is in fact strictly stronger than DS-GNN for a specific policy.

**Proposition 1.** *DSS-GNN is at least as powerful as DS-GNN for all policies. Furthermore, there are policies where DSS-GNN is strictly more powerful than DS-GNN.*

**Base graph encoder choice matters.** Several graph networks with increased expressive power over 1-WL have been proposed (Sato, 2020; Maron et al., 2019a). Thus, one may desire to use a more expressive graph network than MPNNs. The following proposition analyzes the expressivity of a generalization of DS-GNN to use a 3-WL base encoder; details are given in the proof in Appendix E.

**Proposition 2.** *Let the subgraph policy be depth-1* $\widehat{\text{EGO}}$ *or* $\widehat{\text{EGO}+}$*. Then: (1) DS-GNN with a 3-WL base graph encoder is strictly more powerful than 3-WL. (2) DS-GNN with a 3-WL base graph encoder is strictly more powerful than DS-GNN with a 1-WL base graph encoder.*

Part (1) shows that our DS-GNN architecture improves the expressiveness of 3-WL graph encoders over 3-WL itself, so expressiveness gains from our ESAN architecture are not limited to the 1-WL base graph encoder case. Part (2) shows that our architecture can gain in expressive power when increasing the expressive power of the graph encoder.

**Subgraph selection policy matters.** Besides complexity (discussed in Appendix F), different policies also result in different expressive power of our model. We analyze the case of 1-WL (MPNN)

---

[5]In the case of DS-WL it is required that the original graph is included in the bag.

Table 1: TUDatasets. Gray background indicates that ESAN outperforms the base encoder. SoTA line reports results for the best performing model for each dataset.

| Method ↓ / Dataset → | MUTAG | PTC | PROTEINS | NCI1 | NCI109 | IMDB-B | IMDB-M |
|---|---|---|---|---|---|---|---|
| SoTA | 92.7±6.1 | 68.2±7.2 | 77.2±4.7 | 83.6±1.4 | 84.0±1.6 | 77.8±3.3 | 54.3±3.3 |
| GIN (Xu et al., 2019) | 89.4±5.6 | 64.6±7.0 | 76.2±2.8 | 82.7±1.7 | 82.2±1.6 | 75.1±5.1 | 52.3±2.8 |
| **DS-GNN (GIN) (ED)** | 89.9±3.7 | 66.0±7.2 | 76.8±4.6 | 83.3±2.5 | 83.0±1.7 | 76.1±2.6 | 52.9±2.4 |
| **DS-GNN (GIN) (ND)** | 89.4±4.8 | 66.3±7.0 | 77.1±4.6 | 83.8±2.4 | 82.4±1.3 | 75.4±2.9 | 52.7±2.0 |
| **DS-GNN (GIN) (EGO)** | 89.9±6.5 | 68.6±5.8 | 76.7±5.8 | 81.4±0.7 | 79.5±1.0 | 76.1±2.8 | 52.6±2.8 |
| **DS-GNN (GIN) (EGO+)** | 91.0±4.8 | 68.7±7.0 | 76.7±4.4 | 82.0±1.4 | 80.3±0.9 | 77.1±2.6 | 53.2±2.8 |
| **DSS-GNN (GIN) (ED)** | 91.0±4.8 | 66.6±7.3 | 75.8±4.5 | 83.4±2.5 | 82.8±0.9 | 76.8±4.3 | 53.5±3.4 |
| **DSS-GNN (GIN) (ND)** | 91.0±3.5 | 66.3±5.9 | 76.1±3.4 | 83.6±1.5 | 83.1±0.8 | 76.1±2.9 | 53.3±1.9 |
| **DSS-GNN (GIN) (EGO)** | 91.0±4.7 | 68.2±5.8 | 76.7±4.1 | 83.6±1.8 | 82.5±1.6 | 76.5±2.8 | 53.3±3.1 |
| **DSS-GNN (GIN) (EGO+)** | 91.1±7.0 | 69.2±6.5 | 75.9±4.3 | 83.7±1.8 | 82.8±1.2 | 77.1±3.0 | 53.2±2.4 |

graph encoder for different choices of policy, restricting our attention to strongly regular (SR) graphs. These graphs are parameterized by 4 parameters $(n, k, \lambda, \mu)$. SR graphs are an interesting 'corner case' often used for studying GNN expressivity, as any SR graphs of the same parameters cannot be distinguished by 1-WL or 3-WL (Arvind et al., 2020; Bodnar et al., 2021b;a).

We show that our framework with the ND and depth-$n$ EGO+ policies can distinguish any SR graphs of different parameters. However, just like 3-WL, we cannot distinguish SR graphs of the *same* parameters. On the other hand, the ED policy can distinguish certain pairs of non-isomorphic SR graphs of the same parameters, while still being able to distinguish all SR graphs of different parameters. We formalize the above statements in Proposition 3.

**Proposition 3.** *Consider DS-GNN on SR graphs: (1) ND, EGO+, ED can distinguish any SR graphs of different parameters, (2) ND, EGO+ cannot distinguish any SR graphs of the same parameters, (3) ED can distinguish some SR graphs of the same parameters. The EGO+ policies are at depth-$n$. Thus, for distinguishing SR graphs, ND and depth-$n$ EGO+ are as strong as 3-WL, while ED is strictly stronger than 3-WL.*

## 4 EXPERIMENTS

We perform an extensive set of experiments to answer five main questions: (1) *Is our approach more expressive than the base graph encoders in practice?* (2) *Can our approach achieve better performance on real benchmarks?* (3) *How do the different subgraph selection policies compare?* (4) *Can our efficient stochastic variant offer the same benefits as the full approach?* (5) *Are there any empirical advantages to DSS-GNN versus DS-GNN?*. In the following, we report our main experimental results. We defer readers to Appendices C.2 and C.3 for additional details and results, including timings. An overall discussion of the experimental results is found at Appendix C.4. Our code is also available.[6]

**Expressive power: EXP, CEXP, CSL.** In an effort to answer question (1), we first tested the benefits of our framework on the synthetic EXP, CEXP (Abboud et al., 2020), and CSL datasets (Murphy et al., 2019; Dwivedi et al., 2020), specifically designed so that any 1-WL GNN cannot do better than random guess (50% accuracy in EXP, 75% in CEXP, 10% in CSL). As for EXP and CEXP, we report the mean and standard deviation from 10-fold cross-validation in Table 6 in Appendix C.2. While GIN and GraphConv perform no better than a random guess, DS- and DSS-GNN perfectly solve the task when using these layers as base graph encoders, with all the policies working equally well. Table 8 in Appendix C.3 shows the performance of our stochastic variant. In all cases our approach yields almost perfect accuracy, where DSS-GNN works better than DS-GNN when sampling only 5% of the subgraphs. On the CSL benchmark, both DS- and DSS-GNN achieve perfect performance for any considered 1-WL base encoder and policy, even stochastic ones. More details are found in Appendix C.2 and Appendix C.3.

**TUDatasets.** We experimented with popular datasets from the TUD repository. We followed the widely-used hyperparameter selection and experimental procedure proposed by Xu et al. (2019). Salient results are reported in Table 1 where the best performing method for each dataset is reported as SoTA; in the following we discuss the full set of results, reported in Table 4 in Appendix C.2. We observe that: (1) ESAN variants achieve excellent results with respect to SoTA methods: they score first, second and third in, respectively, two, four and one dataset out of seven; (2) ESAN consistently

---

[6]https://github.com/beabevi/ESAN

improves over the base encoder, up to almost 5% (DSS-GNN: 51/56 cases, DS-GNN: 42/56 cases, see gray background); (3) ESAN consistently outperforms other methods aimed at increasing the expressive power of GNNs, such as ID-GNN (You et al., 2021) and RNI (Abboud et al., 2020). Particularly successful configurations are: (i) DSS-GNN (EGO+) + GraphConv/GIN; (ii) DS-GNN (ND) + GIN. Table 7 in Appendix C.3 compares the performance of our stochastic sampling variant with the base encoder (GIN) and the full approach. Our stochastic version provides a strong and efficient alternative, generally outperforming the base encoder. Intriguingly, stochastic sampling occasionally outperforms even the full approach.

**OGB.** We tested the ESAN framework on the OGBG-MOLHIV and OGBG-MOLTOX21 datasets from Hu et al. (2020) using the scaffold splits. Table 2 shows the results of all ESAN variants to the performance of its GIN base encoder, while in Table 5 in Appendix C.2 we report results obtained when employing a GCN base encoder. When using GIN, all ESAN variants improve the base encoder accuracy by up to 2.4% on OGBG-MOLHIV and 3% on OGBG-MOLTOX21. In addition, DSS-GNN performs better that DS-GNN with GCN. In general, DSS-GNN improves the base encoder in 14/16 cases, whereas a DS-GNN improves it in 10/16 cases. Finally, Table 8 in Appendix C.3 shows the performance of our stochastic variant.

Table 2: Test results for OGB datasets. Gray background indicates that ESAN outperforms the base encoder.

| Method | OGBG-MOLHIV ROC-AUC (%) | OGBG-MOLTOX21 ROC-AUC (%) |
|---|---|---|
| GIN (Xu et al., 2019) | 75.58±1.40 | 74.91±0.51 |
| **DS-GNN (GIN) (ED)** | 76.43±2.12 | 75.12±0.50 |
| **DS-GNN (GIN) (ND)** | 76.19±0.96 | 75.34±1.21 |
| **DS-GNN (GIN) (EGO)** | 78.00±1.42 | 76.22±0.62 |
| **DS-GNN (GIN) (EGO+)** | 77.40±2.19 | 76.39±1.18 |
| **DSS-GNN (GIN) (ED)** | 77.03±1.81 | 76.71±0.67 |
| **DSS-GNN (GIN) (ND)** | 76.63±1.52 | 77.21±0.70 |
| **DSS-GNN (GIN) (EGO)** | 77.19±1.27 | 77.45±0.41 |
| **DSS-GNN (GIN) (EGO+)** | 76.78±1.66 | 77.95±0.40 |

Table 3: ZINC12k dataset. Gray background indicates that ESAN outperforms the base encoder.

| Method | ZINC (MAE ↓) |
|---|---|
| PNA (Corso et al., 2020) | 0.188±0.004 |
| DGN (Beaini et al., 2021) | 0.168±0.003 |
| SMP (Vignac et al., 2020) | 0.138±? |
| GIN (Xu et al., 2019) | 0.252±0.017 |
| HIMP (Fey et al., 2020) | 0.151±0.006 |
| GSN (Bouritsas et al., 2022) | 0.108±0.018 |
| CIN-SMALL (Bodnar et al., 2021a) | 0.094±0.004 |
| **DS-GNN (GIN) (ED)** | 0.172±0.008 |
| **DS-GNN (GIN) (ND)** | 0.171±0.010 |
| **DS-GNN (GIN) (EGO)** | 0.126±0.006 |
| **DS-GNN (GIN) (EGO+)** | 0.116±0.009 |
| **DSS-GNN (GIN) (ED)** | 0.172±0.005 |
| **DSS-GNN (GIN) (ND)** | 0.166±0.004 |
| **DSS-GNN (GIN) (EGO)** | 0.107±0.005 |
| **DSS-GNN (GIN) (EGO+)** | 0.102±0.003 |

**ZINC12k.** We further experimented with an additional large scale molecular benchmark: ZINC12K (Sterling & Irwin, 2015; Gómez-Bombarelli et al., 2018; Dwivedi et al., 2020), where, as prescribed, we impose a 100k parameter budget. We observe from Table 3 that all ESAN configurations significantly outperform their base GIN encoder. In general, our method achieves particularly competitive results, irrespective of the chosen subgraph selection policy. It always outperforms the provably expressive PNA model (Corso et al., 2020), while other expressive approaches (Beaini et al., 2021; Vignac et al., 2020) are outperformed when our model is equipped with EGO(+) policies. With the same policy, ESAN also outperforms HIMP (Fey et al., 2020), and GSN (Bouritsas et al., 2022) which explicitly employ information from rings and their counts, a predictive signal on this task (Bodnar et al., 2021a). In conclusion, ESAN is the best performing model amongst all provably expressive, domain agnostic GNNs, while being competitive with and often of superior performance than (provably expressive) GNNs which employ domain specific structural information.

## 5 CONCLUSION

We have presented ESAN, a novel framework for learning graph-structured data, based on decomposing a graph into a set of subgraphs and processing this set in a manner respecting its symmetry structure. We have shown that ESAN increases the expressive power of several existing graph learning architectures and performs well on multiple graph classification benchmarks. The main limitation of our approach is its increased computational complexity with respect to standard MPNNs. We have proposed a more efficient stochastic version to mitigate this limitation and demonstrated that it performs well in practice. In follow-up research, we plan to investigate the following directions: (1) Is it possible to *learn* useful subgraph selection policies? (2) Can higher order structures on subgraphs (e.g., a graph of subgraphs) be leveraged for better expressive power? (3) A theoretical analysis of the stochastic version and of different policies/aggregation functions.

ACKNOWLEDGEMENTS

We would like to thank all those people involved in the 2021 London Geometry and Machine Learning Summer School (LOGML), where this research project started. In particular, we would like to express our gratitude to the members of the organizing committee: Mehdi Bahri, Guadalupe Gonzalez, Tim King, Dr Momchil Konstantinov and Daniel Platt. We would also like to thank Stefanie Jegelka, Cristian Bodnar, and Yusu Wang for helpful discussions. MB is supported in part by ERC Consolidator grant no 724228 (LEMAN).

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

## A BACKGROUND: LEARNING ON SETS AND GRAPHS

In this section we present a brief introduction to GNNs, equivariance and set learning.

**Graph neural networks.** GNNs are a type of neural networks that process graphs as input. Their roots can be traced back to the field of computational chemistry Kireev (1995); Baskin et al. (1997) and the pioneering works of Sperduti (1994); Goller & Kuchler (1996); Sperduti & Starita (1997); Gori et al. (2005); Scarselli et al. (2008). In recent years, GNNs have become a major research direction and are used in many application fields such as social network analysis (Fan et al., 2019), molecular biology (Jin et al., 2018), physics (Shlomi et al., 2020; Cai et al., 2021) and computer vision (Johnson et al., 2018). Over the years, several ideas have served as inspiration for GNNs, including spectral graph theory (Bruna et al., 2013; Bronstein et al., 2017) and equivariance (Maron et al., 2019b; Bronstein et al., 2021). Message-passing neural networks (MPNNs) (Gilmer et al., 2017; Morris et al., 2019; Xu et al., 2019) have been the most popular approach to this problem. These models are inspired by convolutional networks on images because of their locality and stationary characteristics. These networks are composed of several layers, which update node (and edge, if available) features based on the local graph connectivity. There are several ways to define a message passing layer. For example, the following is a popular formulation (Morris et al., 2019):

$$x_v^t = W_1^t x_v^{t-1} + W_2^t \sum_{u \sim v} x_u^{t-1} + b^t \tag{3}$$

Here, $x_v^t \in R^{d_t}$ is the feature associated with a node $v$ after after applying the $t$-th layer, $W_i \in R^{d_t \times d_{t-1}}, b^t \in R^{d_t}$ are the learnable parameters, and $u \sim v$ means that the nodes are adjacent.

**Expressive power of GNNs.** There is a particular interest in the expressive power of GNNs (Sato, 2020). The seminal works of Morris et al. (2019); Xu et al. (2019) analyzed the expressive power of MPNNs and showed that their ability to distinguish graphs is the same as the first version of the WL graph isomorphism test introduced by Weisfeiler & Leman (1968). Importantly, this test cannot distinguish rather simple pairs of graphs (see Figure 1). In general, WL tests are a polynomial time hierarchy of graph isomorphism tests: for $k > k'$, $k$-WL is strictly more powerful than $k'$-WL (Cai et al., 1992), but requires more time and space complexity which is exponential in $k$. These works inspired a plethora of novel GNNs with improved expressive power (Maron et al., 2019a; Bouritsas et al., 2022; Bodnar et al., 2021b;a; Vignac et al., 2020; Murphy et al., 2019; Chen et al., 2019). See Appendix B for a detailed discussion.

**Invariance and equivariance** The notion of symmetry, expressed through the action of a group on signals defined on some domain, is a key concept of 'geometric deep learning', allowing to derive from first principles many modern deep representation learning architectures (Bronstein et al., 2021). Let $H$ be a group acting on vector spaces $V, W$. We say that a function $f : V \to \mathbb{R}$ is *H-invariant* if $f(h \cdot x) = f(x)$ for all $x \in V$, $h \in H$. Similarly, a function $f : V \to W$ is *equivariant* if the function commutes with the group action, namely $f(h \cdot x) = h \cdot f(x)$ for all $x \in V$, $h \in H$. Constructing invariant and equivariant learning models has become a central design paradigm in machine learning, see e.g., Cohen & Welling (2016); Ravanbakhsh et al. (2017); Kondor & Trivedi (2018); Esteves et al. (2018). In particular, graphs typically do not have a canonical ordering of nodes, forcing GNN layers to be *permutation-equivariant*.

**Deep learning on Sets.** Sets can be considered as a special case of graphs with no edges, and learning on set-structured data has recently become an important topic with applications such as computer graphics and vision dealing with 3D point clouds (Qi et al., 2017). A set of $n$ items can be encoded in an $n \times d$ feature matrix $X$, and since the order of rows is arbitrary, one is interested in functions that are invariant or equivariant under element permutations. The works of Zaheer et al. (2017); Ravanbakhsh et al. (2016); Qi et al. (2017) were the first to suggest universal architectures for set data. They are based on element-wise functions and global pooling operations. For example, a Deep Sets layer takes the form

$$x_i^t = W_1^t x_i^{t-1} + W_2^t \sum_{j=1}^{n} x_j^{t-1} + b^t \tag{4}$$

Here, $x_i^t$ is the feature of the $i$-th set element after applying layer $t$. Alternatively, this layer can be described as a message passing layer on a complete graph with Equation 3.

Recently, Maron et al. (2020) suggested the *Deep Sets for Symmetric elements* framework (DSS), a generalization of Deep Sets to a learning setup in which each element has its own symmetry structure, represented by another group $G$ acting on the $d$-dimensional vectors in the input (the rows of $X$), such as translations in the case of images. The paper characterizes the space of linear equivariant layers which boil down to replacing the unconstrained fully-connected matrices $W_i$ in the equation above with corresponding linear equivariant layers, like convolutions in the case of images. We observe that when $W_2 = 0$, we obtain a simple Siamese network. Importantly, the paper shows that when $W_2 \neq 0$, the network exhibits a greater expressive power. Intuitively this is due its enhanced information-sharing abilities within the set.

## B    RELATION TO PREVIOUS WORKS

**Processing subgraphs for graph learning.** Our approach is related to multiple techniques in graph learning. Introduced for semi-supervised node-classification tasks, DropEdge (Rong et al., 2019) can be considered as a stochastic version of the ED policy that processes one single edge-deleted subgraph at a time. Ego-GNNs (Sandfelder et al., 2021) resemble our EGO policy, as messages are passed within each node's ego-net, and are aggregated from each ego-net that a node is contained in. ID-GNNs (You et al., 2021) resemble the EGO+ policy, the difference being the way in which the root node is identified and the way the information between the ego-nets is combined. RNP-GNNs (Tahmasebi et al., 2020) learn functions over recursively-defined node-deleted ego-nets with augmented features, which allows for high expressive power for counting substructures and computing other local functions. FactorGCN (Yang et al., 2020) is a model that disentangles relations in graph data by learning subgraphs containing edges that capture latent relationships. The graph automorphic equivalence network (Xu et al., 2020) compares ego-nets to subgraph templates in a GNN architecture. Furthermore, Hamilton et al. (2017) introduce a minibatching procedure that samples neighborhoods for each GNN layer — in large part for the sake of efficiently processing large graphs. Other works like Cluster-GCN (Chiang et al., 2019) and GraphSAINT (Zeng et al., 2020) have followed by developing methods for sampling whole subgraphs at a time, and then processing these subgraphs with a GNN.

**Provably expressive GNNs.** The inability of WL to disambiguate even simple graph pairs has recently sparked particular interest in provably expressive GNNs. A fruitful streamline of works (Morris et al., 2019; 2020b; Kondor et al., 2018; Maron et al., 2019b;a; Keriven & Peyré, 2019; Azizian & Lelarge, 2021) has proposed architectures aligning to the k-WL hierarchy; the expressiveness of these models is well characterized in terms of this last, but these guarantees come at the cost of a generally intractable computational cost for $k \geq 3$. Additionally, most of these architectures lack an important feature contributing to the recent success of GNNs: *locality* of computation. In an effort to maintain the inductive bias of locality, recent works have proposed to provably enhance the expressiveness of GNNs by augmenting node features with random identifiers (Sato et al., 2021; Loukas, 2020; Abboud et al., 2020; Puny et al., 2020) or auxiliary colourings (Dasoulas et al., 2020). Although of scalable and immediate implementation, these approaches may not preserve equivariance w.r.t. node permutations and random features in particular have been observed to under-perform in generalising outside of training data.

A family of works focused on building local and equivariant provably powerful architectures and, for this reason, is especially related to our approach. Of particular relevance are works which argued for the use of (higher-order) structural information. Structural node identifiers obtained as subgraph isomorphism counting were introduced in Bouritsas et al. (2022). Alternatively, Srinivasan & Ribeiro (2020); de Haan et al. (2020); Thiede et al. (2021) argue for the use the automorphism group of edges and substructures to obtain more expressive representations. These features cannot be computed by standard MPNNs (Chen et al., 2020) for substructures different than trees and they can provably enhance the expressiveness of standard MPNNs while retaining their local inductive bias. Bodnar et al. (2021b;a) introduced the idea of lifting graphs to more general topological spaces such as simplicial and cellular complexes. Higher-order structures are first-class objects thereon and message passing is directly performed on them. In general, listing (or counting) substructures has an intractable worst-case complexity and the design of the substructure bank may be difficult in certain cases. This makes these models of less obvious application outside of the molecular and social domain, where the importance of specific substructures is not well understood and their number may rapidly grow.

Our framework, on the contrary, attains locality, equivariance and provable expressiveness with scalable and non-domain specific policies. Any of the ED, ND, and EGO(+) policies make our approach strictly more powerful than standard MPNNs (see Section D), and the generation of the bags is computationally tractable, independently of the graph distribution at hand (more in Section F). Either way, we remark that the aforementioned works are compatible with our overall framework, and can be employed to parameterize the $E_{\mathrm{subgraphs}}$ module in those cases where our approach may benefit from provably powerful and domain-specific base graph encoders.

As a final note, we would like to point out the presence of two recent works which focused on the problem of automatically extracting relevant graph substructures in an automated way. Bouritsas et al. (2021) propose a model that can be trained to partition graphs into substructures for lossless compression; Chen et al. (2021) show how the model introduced in Chen et al. (2020) can achieve competitive results while retaining interpretability by the detection of the graph substructures employed by such model to make its decisions. These approaches may potentially be used to design more advanced subgraph selection policies.

## B.1 Comparison to Reconstruction GNNs

Contemporaneous work by Cotta et al. (2021) takes motivation from the node reconstruction conjecture (Kelly, 1957) in developing a model that processes node-deleted subgraphs of a given graph by independent MPNNs. Thus, there are some connections between our work and theirs. Nevertheless, we develop a more general framework; in fact, the Reconstruction GNN (Cotta et al., 2021) is exactly our DS-GNN with a $k$-node-deletion subgraph selection policy. There are multiple substantial differences between these works, which we discuss below.

**Architecture for processing bags of subgraphs:** In the terminology of our paper, Cotta et al. (2021) use a DS-GNN architecture, while we perform a rigorous and more specific analysis of equivariance, and argue in favor of a different and more powerful DSS-GNN architecture. We show that DS-GNN is theoretically too restrictive in terms of the invariance it enforces (see the DS-GNN paragraph in Section 2.1) and we prove that DSS-GNN is strictly stronger than DS-GNN for certain policies (see Proposition 1). Moreover, our DSS-GNN achieves better empirical results than DS-GNN in most of the tested settings, so this theoretical benefit leads to practical gains.

**General subgraph policies:** We advocate general subgraph policies, four of which are studied in the paper (node-deletion, edge-deletion, ego-nets and ego-nets+). There are many other possible subgraph policies that fit into our framework, such as those based on trees. In Section 4, we show that the node-deletion policy is not the optimal subgraph selection policy in many practical settings (e.g., on the ZINC12K dataset, the EGO and EGO+ policies perform significantly better).

**More general theoretical analysis:** We provide a more general theoretical analysis. First, since we consider general subgraph selection policies, we are able to prove results for many policies in the same way (e.g. as in Proposition 3), not just for node-deletion policies. Second, we consider higher-order graph encoders in our general framework, whereas Cotta et al. (2021) considers MPNNs and universal feed forward networks. We prove results on the integration of higher-order encoders in our ESAN framework in Proposition 2.

**Formulation of new WL variants:** We also develop and analyze new variants of the Weisfeiler-Lehman test (DS-WL and DSS-WL). We then characterize the expressive power of DS-GNN and DSS-GNN in terms of these variants (see Theorem 2), which helps us to prove other results in Section 3.3, and will help in proofs for future work. In comparison, there is no such WL variant developed by Cotta et al. (2021).

**Basic motivation:** The paper of Cotta et al. (2021) is mainly motivated by the Reconstruction Conjecture in graph theory. In contrast, our paper is motivated by the recent work by Maron et al. (2020) on architectures that are equivariant to product symmetry groups arising when dealing with sets of objects containing internal symmetries.

Table 4: TUDatasets. The top three are highlighted by **First**, **Second**, **Third**. Gray background indicates that ESAN outperforms the base encoder.

| Dataset | MUTAG | PTC | PROTEINS | NCI1 | NCI109 | IMDB-B | IMDB-M |
|---|---|---|---|---|---|---|---|
| RWK (Gärtner et al., 2003) | 79.2±2.1 | 55.9±0.3 | 59.6±0.1 | >3 days | N/A | N/A | N/A |
| GK ($k = 3$) (Shervashidze et al., 2009) | 81.4±1.7 | 55.7±0.5 | 71.4±0.3 | 62.5±0.3 | 62.4±0.3 | N/A | N/A |
| PK (Neumann et al., 2016) | 76.0±2.7 | 59.5±2.4 | 73.7±0.7 | 82.5±0.5 | N/A | N/A | N/A |
| WL KERNEL (Shervashidze et al., 2011) | 90.4±5.7 | 59.9±4.3 | 75.0±3.1 | **86.0**±1.8 | N/A | 73.8±3.9 | 50.9±3.8 |
| DCNN (Atwood & Towsley, 2016) | N/A | N/A | 61.3±1.6 | 56.6±1.0 | N/A | 49.1±1.4 | 33.5±1.4 |
| DGCNN (Zhang et al., 2018) | 85.8±1.8 | 58.6±2.5 | 75.5±0.9 | 74.4±0.5 | N/A | 70.0±0.9 | 47.8±0.9 |
| IGN (Maron et al., 2019b) | 83.9±13.0 | 58.5±6.9 | 76.6±5.5 | 74.3±2.7 | 72.8±1.5 | 72.0±5.5 | 48.7±3.4 |
| PPGNs (Maron et al., 2019a) | 90.6±8.7 | 66.2±6.6 | **77.2**±4.7 | 83.2±1.1 | 82.2±1.4 | 73.0±5.8 | 50.5±3.6 |
| NATURAL GN (de Haan et al., 2020) | 89.4±1.6 | **66.8**±1.7 | 71.7±1.0 | 82.4±1.3 | N/A | 73.5±2.0 | 51.3±1.5 |
| GSN (Bouritsas et al., 2022) | **92.2**±7.5 | **68.2**±7.2 | 76.6±5.0 | 83.5±2.0 | N/A | **77.8**±3.3 | **54.3**±3.3 |
| SIN (Bodnar et al., 2021b) | N/A | N/A | 76.4±3.3 | 82.7±2.1 | N/A | 75.6±3.2 | 52.4±2.9 |
| CIN (Bodnar et al., 2021a) | **92.7**±6.1 | **68.2**±5.6 | **77.0**±4.3 | **83.6**±1.4 | **84.0**±1.6 | 75.6±3.7 | 52.7±3.1 |
| GIN (Xu et al., 2019) | 89.4±5.6 | 64.6±7.0 | 76.2±2.8 | 82.7±1.7 | 82.2±1.6 | 75.1±5.1 | 52.3±2.8 |
| GIN + ID-GNN (You et al., 2021) | 90.4±5.4 | 67.2±4.3 | 75.4±2.7 | 82.6±1.6 | 82.1±1.5 | 76.0±2.7 | 52.7±4.2 |
| DROPEDGE (Rong et al. (2019)) | 91.0±5.7 | 64.5±2.6 | 73.5±4.5 | 82.0±2.6 | 82.2±1.4 | **76.5**± 3.3 | 52.8± 2.8 |
| **DS-GNN (GIN) (ED)** | 89.9±3.7 | 66.0±7.2 | 76.8±4.6 | 83.3±2.5 | 83.0±1.7 | 76.1±2.6 | 52.9±2.4 |
| **DS-GNN (GIN) (ND)** | 89.4±4.8 | 66.3±7.0 | 77.1±4.6 | **83.8**±2.4 | 82.4±1.3 | 75.4±2.9 | 52.7±2.0 |
| **DS-GNN (GIN) (EGO)** | 89.9±6.5 | 68.6±5.8 | 76.7±5.8 | 81.4±0.7 | 79.5±1.0 | 76.1±2.8 | 52.6±2.8 |
| **DS-GNN (GIN) (EGO+)** | 91.0±4.8 | 68.7±7.0 | 76.7±4.4 | 82.0±1.4 | 80.3±0.9 | **77.1**±2.6 | 53.2±2.8 |
| **DS-GNN (GIN) ($\widehat{\text{ED}}$)** | 90.5±5.8 | 66.0±6.4 | 77.0±3.2 | 82.8±1.1 | 82.9±1.2 | 76.2±2.4 | 52.9±2.8 |
| **DS-GNN (GIN) ($\widehat{\text{ND}}$)** | 90.0±6.0 | 66.0±8.7 | **77.3**±3.8 | 83.0±1.9 | 82.3±0.7 | 75.5±2.2 | 52.6±2.9 |
| **DS-GNN (GIN) ($\widehat{\text{EGO}}$)** | 91.0±5.3 | 68.4±7.4 | 76.7±4.4 | 81.5±1.1 | 79.6±1.5 | 76.6±2.5 | 52.6±2.8 |
| **DS-GNN (GIN) ($\widehat{\text{EGO+}}$)** | 90.5±4.7 | 68.1±7.8 | 76.6±4.0 | 82.0±1.2 | 80.4±1.4 | 76.8±2.7 | 53.3±2.1 |
| **DSS-GNN (GIN) (ED)** | 91.0±4.8 | 66.6±7.3 | 75.8±4.5 | 83.4±2.5 | 82.8±0.9 | 76.8±4.3 | 53.5±3.4 |
| **DSS-GNN (GIN) (ND)** | 91.0±3.5 | 66.3±5.9 | 76.1±3.4 | 83.6±1.5 | **83.1**±0.8 | 76.1±2.9 | 53.3±1.9 |
| **DSS-GNN (GIN) (EGO)** | 91.0±4.7 | 68.2±5.8 | 76.7±4.1 | 83.6±1.8 | 82.5±1.6 | 76.5±2.8 | 53.3±3.1 |
| **DSS-GNN (GIN) (EGO+)** | 91.1±7.0 | **69.2**±6.5 | 75.9±4.3 | 83.7±1.8 | 82.8±1.2 | **77.1**±3.0 | 53.2±2.4 |
| GRAPHCONV (Morris et al., 2019) | 90.5±4.6 | 64.9±10.4 | 73.9±6.1 | 82.4±2.7 | 81.7±1.0 | 76.1±3.9 | **53.1**±2.9 |
| GRAPHCONV + ID-GNN (You et al., 2021) | 89.4±4.1 | 65.4±7.1 | 71.9±4.6 | 83.4±2.4 | **82.9**±1.2 | 76.1±2.5 | **53.7**±3.3 |
| RNI (Abboud et al., 2020) | 91.0±4.9 | 64.3±6.1 | 73.3±3.3 | 82.1±1.7 | 81.7±1.0 | 75.5±3.3 | **53.1**±1.9 |
| **DS-GNN (GRAPHCONV) (ED)** | 90.4±4.1 | 65.7±5.2 | 76.3±5.2 | 82.7±1.9 | 82.4±1.5 | 75.3±2.3 | 53.5±2.3 |
| **DS-GNN (GRAPHCONV) (ND)** | 88.3±5.1 | 66.6±7.8 | 76.8±3.9 | 82.9±2.5 | 82.7±1.3 | 75.7±2.9 | 53.5±2.1 |
| **DS-GNN (GRAPHCONV) (EGO)** | 89.4±5.4 | 66.6±6.5 | 76.7±5.4 | 81.3±1.9 | 79.6±2.0 | 76.6±4.0 | 53.1±1.5 |
| **DS-GNN (GRAPHCONV) (EGO+)** | 90.4±5.8 | 67.4±4.7 | 76.8±4.3 | 82.8±2.5 | 80.6±1.3 | 76.0±1.6 | 53.3±2.4 |
| **DS-GNN (GRAPHCONV) ($\widehat{\text{ED}}$)** | 90.0±4.3 | 66.0±7.1 | 76.6±3.7 | 83.0±1.8 | 82.6±1.1 | 76.3±3.1 | 53.2±1.9 |
| **DS-GNN (GRAPHCONV) ($\widehat{\text{ND}}$)** | 90.4±4.0 | 66.0±5.0 | 76.5±4.8 | 83.0±2.3 | 82.7±1.4 | 75.1±1.5 | **53.7**±2.1 |
| **DS-GNN (GRAPHCONV) ($\widehat{\text{EGO}}$)** | 90.0±5.0 | 67.8±5.9 | 76.8±4.8 | 81.9±1.7 | 80.4±2.0 | 76.4±3.2 | 53.1±2.7 |
| **DS-GNN (GRAPHCONV) ($\widehat{\text{EGO+}}$)** | 91.5±5.3 | 67.0±5.5 | 76.5±4.0 | 82.0±1.2 | 80.5±1.7 | 76.4±2.7 | 53.3±2.6 |
| **DSS-GNN (GRAPHCONV) (ED)** | 91.0±5.8 | 66.3±7.7 | 75.7±3.6 | 83.1±2.3 | 82.9±1.0 | 75.8±2.8 | **53.7**±2.8 |
| **DSS-GNN (GRAPHCONV) (ND)** | 90.6±5.2 | 65.4±5.8 | 76.2±5.0 | 83.7±1.7 | 82.4±1.3 | 75.1±3.2 | 53.3±2.6 |
| **DSS-GNN (GRAPHCONV) (EGO)** | 91.5±4.9 | 68.0±6.1 | 76.6±4.6 | 83.5±1.1 | 82.5±1.6 | 76.3±3.6 | 53.1±2.8 |
| **DSS-GNN (GRAPHCONV) (EGO+)** | **92.0**±5.0 | 67.7±5.7 | 77.0±5.4 | 83.4±1.8 | 82.6±1.5 | 76.6±2.8 | 53.6±2.8 |

# C  EXPERIMENTS: FURTHER DETAILS AND DISCUSSION

## C.1  DATASETS AND MODELS IN COMPARISON

We conducted experiments on thirteen graph classification datasets originating from five data repositories: (1) RNI (Abboud et al., 2020) and CSL (Murphy et al., 2019; Dwivedi et al., 2020) (measuring expressive power) (2) TUD repository (social network analysis and bioinformatics) (Morris et al., 2020a) (3) Open Graph Benchmark (molecules) (Hu et al., 2020) and (4) ZINC12k (molecules) (Dwivedi et al., 2020). These datasets are diverse on a number of important levels, including the number of graphs (188-41,127), existence of input node and edge features, and sparsity.

We compared our approach to a wide range of methods, including maximally expressive MPNNs (Xu et al., 2019; Morris et al., 2019). We particularly focused on more expressive GNNs (Maron et al., 2019a; de Haan et al., 2020; Bouritsas et al., 2022; Abboud et al., 2020; Bodnar et al., 2021b; Abboud et al., 2020) and also compared to DropEdge (Rong et al., 2019) and ID-GNN (You et al., 2021), that share some similarity with our ED and EGO policies.[7]

We experimented with both DSS-GNN and DS-GNN on all the datasets and used popular MPNN-type GNNs as base encoders: GIN (Xu et al., 2019), GraphConv (Morris et al., 2019), and GCN

---

[7]We implemented the fast version of ID-GNN with 10 dimensional augmented node features.

Table 5: Results for OGBG-MOLHIV and OGBG-MOLTOX21. Gray background indicates that ESAN outperforms the base encoder. Note that ESAN achieves much higher training accuracies with respect to base graph encoders.

| Method | OGBG-MOLHIV ROC-AUC (%) | | | OGBG-MOLTOX21 ROC-AUC (%) | | |
|---|---|---|---|---|---|---|
| | Training | Validation | Test | Training | Validation | Test |
| GCN (Kipf & Welling, 2017) | 88.65±2.19 | 82.04±1.41 | 76.06±0.97 | 92.06±1.81 | 79.04±0.19 | 75.29±0.69 |
| **DS-GNN (GCN) (ED)** | 86.25±2.77 | 82.36±0.75 | 74.70±1.94 | 90.97±1.70 | 80.03±0.59 | 74.86±0.92 |
| **DS-GNN (GCN) (ND)** | 86.82±4.26 | 81.90±0.82 | 74.40±2.48 | 89.28±0.99 | 80.56±0.61 | 75.79±0.30 |
| **DS-GNN (GCN) (EGO)** | 91.91±3.83 | 83.51±0.95 | 74.00±2.38 | 89.29±1.64 | 80.48±0.52 | 75.41±0.72 |
| **DS-GNN (GCN) (EGO+)** | 86.85±3.57 | 81.95±0.69 | 73.84±2.58 | 90.24±1.20 | 80.77±0.51 | 74.74±0.96 |
| **DSS-GNN (GCN) (ED)** | 99.52±0.34 | 83.44±1.10 | 76.00±1.41 | 94.60±1.10 | 80.05±0.40 | 75.34±0.69 |
| **DSS-GNN (GCN) (ND)** | 97.40±3.52 | 82.88±1.29 | 75.17±1.35 | 93.82±2.47 | 81.22±0.52 | 75.56±0.59 |
| **DSS-GNN (GCN) (EGO)** | 98.56±2.08 | 84.34±1.02 | 76.16±1.02 | 93.06±2.54 | 81.51±0.43 | 76.14±0.53 |
| **DSS-GNN (GCN) (EGO+)** | 98.47±2.20 | 84.45±0.65 | 76.50±1.38 | 91.60±1.81 | 81.55±0.63 | 76.29±0.78 |
| GIN (Xu et al., 2019) | 88.64±2.54 | 82.32±0.90 | 75.58±1.40 | 93.06±0.88 | 78.32±0.48 | 74.91±0.51 |
| **DS-GNN (GIN) (ED)** | 91.71±3.50 | 83.32±0.83 | 76.43±2.12 | 92.38±1.57 | 78.98±0.45 | 75.12±0.50 |
| **DS-GNN (GIN) (ND)** | 89.70±3.20 | 83.21±0.87 | 76.19±0.96 | 91.23±2.15 | 79.61±0.59 | 75.34±1.21 |
| **DS-GNN (GIN) (EGO)** | 93.43±2.17 | 84.28±0.90 | 78.00±1.42 | 92.08±1.79 | 81.28±0.54 | 76.22±0.62 |
| **DS-GNN (GIN) (EGO+)** | 90.53±3.01 | 84.63±0.83 | 77.40±2.19 | 90.07±1.65 | 81.29±0.57 | 76.39±1.18 |
| **DSS-GNN (GIN) (ED)** | 91.69±3.47 | 83.33±0.98 | 77.03±1.81 | 95.85±2.08 | 80.83±0.41 | 76.71±0.67 |
| **DSS-GNN (GIN) (ND)** | 90.71±4.28 | 83.62±1.20 | 76.63±1.52 | 96.90±1.45 | 81.22±0.23 | 77.21±0.70 |
| **DSS-GNN (GIN) (EGO)** | 97.05±3.50 | 85.72±1.21 | 77.19±1.27 | 95.58±1.61 | 81.80±0.20 | 77.45±0.41 |
| **DSS-GNN (GIN) (EGO+)** | 94.47±2.13 | 85.51±0.79 | 76.78±1.66 | 96.06±1.61 | 81.82±0.21 | 77.95±0.40 |

Table 6: Results for RNI datasets. Gray background indicates that ESAN outperforms the base encoder. DS/DSS-GNN can boost the performance of both base graph encoders, GIN and GraphConv, from random to perfect.

| | EXP | CEXP |
|---|---|---|
| GIN (Xu et al., 2019) | 51.1±2.1 | 70.2±4.1 |
| GIN + ID-GNN (You et al., 2021) | 100±0.0 | 100±0.0 |
| **DS-GNN (GIN) (ED/ND/EGO/EGO+)** | 100±0.0 | 100±0.0 |
| **DSS-GNN (GIN) (ED/ND/EGO/EGO+)** | 100±0.0 | 100±0.0 |
| GRAPHCONV (Morris et al., 2019) | 50.3±2.6 | 72.9±3.6 |
| GRAPHCONV + ID-GNN (You et al., 2021) | 100±0.0 | 100±0.0 |
| **DS-GNN (GRAPHCONV) (ED/ND/EGO/EGO+)** | 100±0.0 | 100±0.0 |
| **DSS-GNN (GRAPHCONV) (ED/ND/EGO/EGO+)** | 100±0.0 | 100±0.0 |

(Kipf & Welling, 2017) using all four selection policies (ND, ED, EGO, and EGO+). We used max for aggregating the adjacency matrices for DSS-GNN.

## C.2 FURTHER EXPERIMENTAL DETAILS

We implemented our approach using the PyG framework (Fey & Lenssen, 2019) and ran the experiments on NVIDIA DGX V100 stations. We performed hyper parameter tuning using the Weights and Biases framework (Biewald, 2020) for all the methods, including the baselines.

Details of hyper parameter grids and architectures are discussed in the subsections below.

### C.2.1 TUDATASETS

We considered the evaluation procedure proposed by Xu et al. (2019). Specifically, we conducted 10-fold cross validation and reported the validation performances at the epoch achieving the highest averaged validation accuracy across all the folds. We used Adam optimizer with learning rate decayed by a factor of 0.5 every 50 epochs. The training is stopped after 350 epochs. As for DS-GNN, we implemented $R_{subgraphs}$ with summation over node features, while module $E_{sets}$ is parameterized with a two-layer DeepSets with final mean readout; layers are in the form of Equation (4). In DSS-GNN, we considered the mean aggregator for the feature matrix and used the adjacency matrix of

the original graph. We implemented $R_{\text{subgraphs}}$ by averaging node representations on each subgraph; $E_{\text{sets}}$ is either a two-layer DeepSets in the form of Equation (4) with final mean readout, or a simple averaging of subgraphs representations (the choice is driven by hyper parameter tuning). We considered two baseline models, namely GIN (Xu et al., 2019) and GraphConv (Morris et al., 2019) and based our hyper parameter search on the same parameters of the corresponding baseline, as detailed below.

**GIN.** We considered the same architecture as in Xu et al. (2019), with 4 GNN layers with all MLPs having 1 hidden layer. We used the Jumping Knowledge scheme from Xu et al. (2018) and employed batch normalization (Ioffe & Szegedy, 2015). As in Xu et al. (2019), we tuned the batch size in $\{32, 128\}$, the embedding dimension of the MLPs in $\{16, 32\}$ and the initial learning rate in $\{0.01, 0.001\}$. We tuned the embedding dimension of the DeepSets layers of $E_{\text{sets}}$ in $\{32, 64\}$.

**GraphConv.** We considered the same architecture as in Morris et al. (2019), with 3 GNN layers and embedding dimension of 64. We added batch normalization (Ioffe & Szegedy, 2015) after each layer in all models, as we noticed a gain in performances. For a fair comparison, we tuned batch size, learning rate and DeepSets hidden channels over the same grid employed for GIN.

**Additional comparisons.** We additionally implemented and compared our models to Dropedge (Rong et al., 2019), RNI (Abboud et al., 2020) and ID-GNN (You et al., 2021). For Dropedge, we used a drop rate of $20\%$ to drop edges while training. For RNI, we considered Partial-RNI where $50\%$ of node features are randomized and the remaining are deterministically set to the initial node features. For ID-GNN, we implemented the fast version with 10 dimensional augmented node features, due to the good performances presented in the original paper. Table 4 shows the full results on the TUDatasets, including a comparison to the graph kernels (Gärtner et al., 2003; Shervashidze et al., 2009; 2011; Neumann et al., 2016). As discussed in the main paper, to guarantee at least the expressive power of the basic graph encoder used for DS-GNN one can add the original graph $G$ to $S_G^\pi$ for any policy $\pi$, obtaining the augmented version $\widehat{\pi}$. We included results for our DS-GNN on the augmented policies in the same table. As can be seen, DS-GNN obtains very similar results for any policy and its corresponding augmented version.

### C.2.2 OGB DATASETS

We used the evaluation procedure proposed in Hu et al. (2020). Specifically, we ran each experiment with 10 different initialization seeds and we reported the mean performances (along with standard deviation) corresponding to the best validation result. We used Adam optimizer and trained for 100 epochs. We considered two baseline models: GIN (Xu et al., 2019) and GCN (Kipf & Welling, 2017) and we kept the proposed parameters (Hu et al., 2020), i.e., 5 GNN layers with embedding dimension 300. We tuned the learning rate in $\{0.001, 0.0001\}$. DS-GNN implements $R_{\text{subgraphs}}$ as either averaging or summation of node representations, the choice is made based on validation ROC-AUC. $E_{\text{sets}}$ is a two-layer DeepSets (Equation (4)) with final mean readout. Its hidden dimension is tuned in $\{32, 64\}$. As for DSS-GNN, we used the mean aggregator for the feature matrix and used the adjacency matrix of the original graph. Module $R_{\text{subgraphs}}$ applies averaging over node representations and, in turn, $E_{\text{sets}}$ averages the obtained subgraph encodings. Table 5 shows the full results on the OGB datasets, reporting training, validation and test performances.

### C.2.3 RNI DATASETS

We used the same training procedure and hyper parameter search scheme considered for TUDatasets. The only architectural difference is the number of GNN layers that we increased to 6 and the embedding dimension that we fixed to 32. Table 6 reports the results on the RNI datasets.

### C.2.4 CSL

We tuned the batch size in $\{8, 16\}$, the embedding dimension of the MLPs in $\{32, 64\}$ the initial learning rate in $\{0.01, 0.001\}$ and number of layers in $\{4, 8\}$. Interestingly, we find that, when employing the ND selection policy, DS-GNN with less than 5 layers can not differentiate $\text{CSL}(41, 9)$ from $\text{CSL}(41, 12)$ and only achieves $90\%$ accuracy. Increasing the number of layers is crucial for DS-GNN to achieve perfect performance in this case. Similarly, ego networks of $\text{CSL}(41, 9)$ and

Table 7: Results of our stochastic sampling approach on TUDatasets, where each graph sees 100%, 50%, 20%, 5% of the subgraphs in the selected policy (100% corresponds to full bags). Gray background indicates that ESAN outperforms the base encoder.

| | | MUTAG | PTC | PROTEINS | NCI1 | NCI109 | IMDB-B | IMDB-M | RDT-B |
|---|---|---|---|---|---|---|---|---|---|
| GIN (Xu et al., 2019) | | 89.4±5.6 | 64.6±7.0 | 76.2±2.8 | 82.7±1.7 | 82.2±1.6 | 75.1±5.1 | 52.3±2.8 | 92.4±2.5 |
| **DS-GNN (GIN) (ED)** | 100% | 89.9±3.7 | 66.0±7.2 | 76.8±4.6 | 83.3±2.5 | 83.0±1.7 | 76.1±2.6 | 52.9±2.4 | N/A |
| | 50% | 89.9±5.6 | 67.7±5.9 | 76.9±3.8 | 83.0±1.8 | 83.0±1.0 | 76.3±2.6 | 52.9±2.6 | N/A |
| | 20% | 90.5±5.2 | 66.3±5.3 | 77.1±4.5 | 83.6±1.9 | 82.9±1.6 | 76.2±2.4 | 53.2±2.4 | N/A |
| | 5% | 88.8±6.5 | 65.7±7.3 | 76.8±4.6 | 83.8±1.9 | 82.7±1.8 | 76.2±2.8 | 53.3±2.4 | 92.7±1.0 |
| **DS-GNN (GIN) (ND)** | 100% | 89.4±4.8 | 66.3±7.0 | 77.1±4.6 | 83.8±2.4 | 82.4±1.3 | 75.4±2.9 | 52.7±2.0 | N/A |
| | 50% | 88.9±4.3 | 66.2±6.8 | 76.8±5.2 | 83.1±1.5 | 82.7±1.6 | 75.5±2.8 | 52.9±2.2 | N/A |
| | 20% | 88.9±5.5 | 66.0±6.1 | 77.2±2.9 | 84.0±1.6 | 82.7±0.8 | 75.5±1.6 | 52.5±2.9 | N/A |
| | 5% | 88.3±4.5 | 66.3±8.7 | 77.3±4.1 | 83.7±2.2 | 82.8±0.7 | 75.5±3.9 | 52.1±1.8 | 92.4±1.2 |
| **DS-GNN (GIN) (EGO)** | 100% | 89.9±6.5 | 68.6±5.8 | 76.7±5.8 | 81.4±0.7 | 79.5±1.0 | 76.1±2.8 | 52.6±2.8 | N/A |
| | 50% | 89.9±5.6 | 67.5±5.8 | 76.9±4.6 | 81.7±1.4 | 80.3±0.8 | 76.8±2.3 | 52.4±2.7 | N/A |
| | 20% | 88.3±5.8 | 67.5±8.8 | 77.3±4.1 | 80.7±1.1 | 78.3±2.1 | 76.9±2.6 | 52.8±2.6 | N/A |
| | 5% | 86.8±7.1 | 67.7±8.6 | 76.9±4.4 | 77.0±1.9 | 74.0±1.6 | 76.9±3.5 | 52.3±2.8 | 89.0±2.0 |
| **DS-GNN (GIN) (EGO+)** | 100% | 91.0±4.8 | 68.7±7.0 | 76.7±4.4 | 82.0±1.4 | 80.3±0.9 | 77.1±2.6 | 53.2±2.8 | N/A |
| | 50% | 91.6±5.4 | 67.8±8.4 | 77.5±4.7 | 82.5±1.1 | 80.4±1.1 | 76.4±3.1 | 53.1±3.6 | N/A |
| | 20% | 91.5±6.0 | 66.3±8.0 | 77.1±4.4 | 81.5±1.6 | 79.0±2.1 | 76.2±2.3 | 52.6±1.6 | N/A |
| | 5% | 88.4±5.1 | 67.7±6.4 | 77.2±4.2 | 77.2±1.7 | 75.2±0.7 | 77.2±2.3 | 53.2±3.9 | 89.1±2.3 |
| **DSS-GNN (GIN) (ED)** | 100% | 91.0±4.8 | 66.6±7.3 | 75.8±4.5 | 83.4±2.5 | 82.8±0.9 | 76.8±4.3 | 53.5±3.4 | N/A |
| | 50% | 90.5±5.1 | 66.3±6.7 | 75.9±4.3 | 83.0±2.4 | 82.4±1.6 | 76.1±3.1 | 53.7±2.4 | N/A |
| | 20% | 90.4±5.3 | 66.6±6.3 | 76.3±3.8 | 83.1±1.7 | 82.9±1.9 | 76.5±3.3 | 52.9±1.8 | N/A |
| | 5% | 89.4±6.2 | 65.7±7.0 | 75.4±4.5 | 83.2±2.1 | 83.3±1.3 | 77.0±3.2 | 53.3±3.4 | 92.7±1.5 |
| **DSS-GNN (GIN) (ND)** | 100% | 91.0±3.5 | 66.3±5.9 | 76.1±3.4 | 83.6±1.5 | 83.1±0.8 | 76.1±2.9 | 53.3±1.9 | N/A |
| | 50% | 88.9±6.1 | 65.1±4.8 | 76.4±3.9 | 83.1±2.3 | 82.4±1.6 | 75.6±2.4 | 53.1±2.6 | N/A |
| | 20% | 89.9±4.3 | 66.0±9.0 | 76.2±4.8 | 83.2±2.2 | 82.3±1.4 | 75.9±1.8 | 53.5±3.0 | N/A |
| | 5% | 89.9±5.2 | 67.2±7.0 | 77.0±3.9 | 82.8±2.0 | 82.6±1.8 | 76.3±4.6 | 53.1±2.8 | 92.7±1.3 |
| **DSS-GNN (GIN) (EGO)** | 100% | 91.0±4.7 | 68.2±5.8 | 76.7±4.1 | 83.6±1.8 | 82.5±1.6 | 76.5±2.8 | 53.3±3.1 | N/A |
| | 50% | 91.5±5.5 | 68.8±7.7 | 77.3±3.5 | 83.5±1.8 | 83.0±0.8 | 76.3±3.2 | 52.9±3.1 | N/A |
| | 20% | 91.5±4.8 | 66.3±5.9 | 77.1±4.4 | 83.4±1.4 | 82.8±1.0 | 76.9±3.0 | 52.8±3.1 | N/A |
| | 5% | 89.4±5.2 | 67.1±7.5 | 76.6±4.3 | 83.3±2.0 | 83.0±1.4 | 77.2±3.2 | 52.9±2.4 | 92.8±1.6 |
| **DSS-GNN (GIN) (EGO+)** | 100% | 91.1±7.0 | 69.2±6.5 | 75.9±4.3 | 83.7±1.8 | 82.8±1.2 | 77.1±3.0 | 53.2±2.4 | N/A |
| | 50% | 91.0±5.8 | 66.9±7.4 | 77.1±3.0 | 83.7±1.4 | 83.0±1.7 | 76.4±4.1 | 53.0±3.1 | N/A |
| | 20% | 90.5±5.2 | 66.6±7.2 | 77.4±4.5 | 83.7±1.3 | 82.9±1.1 | 77.7±3.4 | 53.2±2.9 | N/A |
| | 5% | 89.4±5.2 | 67.2±6.3 | 76.8±3.1 | 83.3±1.4 | 83.2±1.3 | 77.1±1.9 | 53.5±2.6 | 93.3±1.3 |

CSL(41, 12) of depth 2 and 3 are isomorphic. Therefore, when working with EGO and EGO+ policies, we need depth 4 to distinguish those two classes.

### C.2.5 ZINC12K

Throughout our experimentation, we used the same train, validation and test splits in Dwivedi et al. (2020), whose prescribed overall optimization and benchmarking setup we verbatim follow. We used Mean Absolute Error (MAE) as the training loss and evaluation performance metric. Our models are trained with batches of size 128 and an Adam optimizer whose learning rate is initially set to 0.001, reduced by 0.5 whenever the validation loss does not improve after a patience value set to 20 epochs. Training is stopped when the learning rate reduces below $10^{-5}$. We ran each experiment for 10 different initialization seeds, and reported the mean performances (along with standard deviation) at the time of the early stopping. All models feature 4 GNN layers, as common in this benchmark. Importantly, in order to comply with the 100k parameter budget advocated in Dwivedi et al. (2020), we set the embedding dimension of all GNN modules employed in our models to 64. For the EGO and EGO+ policies we used depth 3 ego-nets centered at each node.

DS-GNN implements $R_{\text{subgraphs}}$ as averaging of node representations while $E_{\text{sets}}$ is a two-layer *invariant* DeepSets (Zaheer et al., 2017) with final mean readout. Specifically, we tune $\phi$ and $\rho$ to either be both a single layer or a 2-layers MLP, with dimensions fixed to 96. As for DSS-GNN, we used the mean aggregator for the feature matrix and used the adjacency matrix of the original graph. Module $R_{\text{subgraphs}}$ applies averaging over node representations and, in turn, $E_{\text{sets}}$ averages the obtained subgraph encodings.

Table 8: Results of our stochastic sampling approach on OGB and RNI datasets, where each graph sees $100\%, 50\%, 20\%, 5\%$ of the subgraphs in the selected policy. Gray background indicates that ESAN outperforms the base encoder.

| | | OGBG-MOLHIV | OGBG-MOLTOX21 | EXP | CEXP |
|---|---|---|---|---|---|
| GIN (Xu et al., 2019) | | 75.58±1.40 | 74.91±0.51 | 51.2±2.1 | 70.2±4.1 |
| **DS-GNN (GIN) (ED)** | 100% | 76.43±2.12 | 75.12±0.50 | 100±0.0 | 100±0.0 |
| | 50% | 76.29±1.33 | 74.59±0.71 | 100±0.0 | 100±0.0 |
| | 20% | 76.57±1.48 | 75.67±0.89 | 100±0.0 | 99.9±0.2 |
| | 5% | 77.82±1.00 | 76.39±1.11 | 99.7±0.4 | 99.9±0.2 |
| **DS-GNN (GIN) (ND)** | 100% | 76.19±0.96 | 75.34±1.21 | 100±0.0 | 100±0.0 |
| | 50% | 77.23±1.32 | 74.82±1.05 | 100±0.0 | 99.9±0.2 |
| | 20% | 77.65±0.84 | 75.66±0.46 | 100±0.0 | 99.9±0.2 |
| | 5% | 78.26±1.02 | 76.51±1.04 | 97.2±1.1 | 99.8±0.8 |
| **DS-GNN (GIN) (EGO)** | 100% | 78.00±1.42 | 76.22±0.62 | 100±0.0 | 100±0.0 |
| | 50% | 76.52±0.72 | 75.98±0.72 | 100±0.0 | 99.9±0.2 |
| | 20% | 77.49±1.32 | 75.88±0.50 | 99.9±0.2 | 96.8±1.5 |
| | 5% | 73.92±1.78 | 74.95±0.54 | 93.5±1.3 | 83.9±3.8 |
| **DS-GNN (GIN) (EGO+)** | 100% | 77.40±2.19 | 76.39±1.18 | 100±0.0 | 100±0.0 |
| | 50% | 76.91±1.22 | 75.69±1.17 | 100±0.0 | 99.9±0.2 |
| | 20% | 75.92±1.59 | 75.84±0.63 | 99.7±0.4 | 97.0±1.4 |
| | 5% | 73.46±1.80 | 75.08±0.96 | 93.7±2.7 | 83.2±2.6 |
| **DSS-GNN (GIN) (ED)** | 100% | 77.03±1.81 | 76.71±0.67 | 100±0.0 | 100±0.0 |
| | 50% | 77.50±1.82 | 76.40±0.84 | 100±0.0 | 100±0.0 |
| | 20% | 76.82±1.83 | 76.31±0.90 | 100±0.0 | 100±0.0 |
| | 5% | 76.71±1.46 | 76.84±0.54 | 99.8±0.3 | 100±0.0 |
| **DSS-GNN (GIN) (ND)** | 100% | 76.63±1.52 | 77.21±0.70 | 100±0.0 | 100±0.0 |
| | 50% | 76.96±1.71 | 76.92±0.94 | 100±0.0 | 100±0.0 |
| | 20% | 76.23±1.48 | 77.07±1.03 | 100±0.0 | 100±0.0 |
| | 5% | 76.74±1.67 | 76.54±0.86 | 97.7±1.0 | 99.9±0.2 |
| **DSS-GNN (GIN) (EGO)** | 100% | 77.19±1.27 | 77.45±0.41 | 100±0.0 | 100±0.0 |
| | 50% | 76.42±1.38 | 76.37±1.02 | 100±0.0 | 100±0.0 |
| | 20% | 76.41±1.05 | 77.47±0.65 | 100±0.0 | 100±0.0 |
| | 5% | 76.38±1.48 | 77.40±0.58 | 99.2±0.6 | 100±0.0 |
| **DSS-GNN (GIN) (EGO+)** | 100% | 76.78±1.66 | 77.95±0.40 | 100±0.0 | 100±0.0 |
| | 50% | 76.88±0.93 | 76.42±0.93 | 100±0.0 | 100±0.0 |
| | 20% | 76.93±1.45 | 76.45±0.81 | 100±0.0 | 100±0.0 |
| | 5% | 75.97±0.80 | 76.70±0.56 | 99.5±0.6 | 100±0.0 |

Table 9: Number of parameters of the best models

| | MUTAG | PTC | PROTEINS | NCI1 | NCI109 | IMDB-B | IMDB-M |
|---|---|---|---|---|---|---|---|
| GIN (Xu et al., 2019) | 8296 | 8692 | 8164 | 9286 | 9319 | 39305 | 40091 |
| **ESAN (GIN)** | 18625 | 5601 | 10865 | 47137 | 20481 | 33729 | 26883 |

## C.3  ADDITIONAL EXPERIMENTS

### C.3.1  STOCHASTIC SAMPLING RESULTS

We performed our stochastic sampling using GIN as a base encoder. We considered three different subgraph sub-sampling ratios: 5%, 20% and 50%. While training, we randomly sampled a subset of subgraphs for each graph in each epoch. During evaluation, we performed majority voting using 5 different subgraph sampling sets for each graph. We performed hyper parameter tuning as detailed in Appendix C.2, and compared the performances of the sub-sampling method both to the base encoder, and to the corresponding approach without the sub-sampling procedure (reported as 100%). Results for the TUDatasets can be found in Table 7. As can be seen in the table, the performance of our approaches do not generally degrade when seeing only a fraction of the subgraphs. We showcased the importance of sampling by adding experiments on RDT-B dataset, which contains large graphs (430 nodes and 498 edges per graph on average) and sampling is employed to alleviate the computational burden. We used the same hyperparameter tuning and evaluation strategy of the other TUDatasets but we fixed the batch size to 32. ESAN outperforms the base encoder even when only seeing 5% of the subgraphs. Table 8 shows the performance of our stochastic variant on the RNI datasets. Even a small fraction of subgraphs is enough to achieve good results on EXP and CEXP. Notably, DSS-GNN works better than DS-GNN for the fraction 5%. Similarly, Table 8 also reports the performances on the OGB datasets. We additionally experimented with stochastic sampling policies on the CSL dataset. Although isomorphic graphs may be mapped to different representations and possibly different classes due to the randomness introduced by the sampling procedure, we observed that sampling does not deteriorate performance in practice: ESAN was able to still obtain perfect accuracy. We hypothesise that different samplings of the same bag are

Table 10: Results with larger number of trainable weights

|  | MUTAG | PROTEINS | NCI1 | NCI109 |
|---|---|---|---|---|
| GIN (Xu et al., 2019) | 89.4±5.6 | 76.2±2.8 | 82.7±1.7 | 82.2±1.6 |
| BIG-GIN | 89.9±4.9 | 75.8±5.5 | 82.9±1.8 | 81.6±1.5 |
| **ESAN (GIN)** | 91.1±7.0 | 77.1±4.6 | 83.8±2.4 | 83.1±0.8 |

Table 11: Timing comparison per epoch on a RTX 2080 GPU. Time taken for a single epoch with batch size 32 on the NCI1 dataset and with batch size 128 on the ZINC dataset. All values are in seconds.

|  | NCI1 | | | ZINC | |
|---|---|---|---|---|---|
|  | BASELINE | 100% SUBGRAPHS | 20% SUBGRAPHS | BASELINE | 100% SUBGRAPHS |
| GIN (Xu et al., 2019) | 1.00±0.05 | - | - | 1.45±0.01 | - |
| **DS-GNN (GIN) (ED)** | - | 2.07±0.01 | 1.73±0.01 | - | 3.56±0.03 |
| **DS-GNN (GIN) (ND)** | - | 2.08±0.03 | 1.71±0.01 | - | 3.42±0.02 |
| **DS-GNN (GIN) (EGO)** | - | 1.96±0.01 | 1.72±0.01 | - | 3.02±0.04 |
| **DS-GNN (GIN) (EGO+)** | - | 2.01±0.02 | 1.73±0.01 | - | 3.09±0.04 |
| **DSS-GNN (GIN) (ED)** | - | 3.26±0.07 | 2.94±0.01 | - | 4.25±0.01 |
| **DSS-GNN (GIN) (ND)** | - | 3.19±0.07 | 2.91±0.02 | - | 4.12±0.03 |
| **DSS-GNN (GIN) (EGO)** | - | 3.14±0.04 | 2.79±0.02 | - | 3.63±0.02 |
| **DSS-GNN (GIN) (EGO+)** | - | 3.19±0.03 | 2.90±0.01 | - | 3.69±0.05 |

still mapped closely in the embedding space, this allowing our models to easily predict the same isomorphism class for all of them.

### C.3.2 SIZE OF THE MODELS

As already discussed, on TUDatasets, our ESAN variants tend to outperform their base encoders in the majority of the cases. We made sure to fairly tune these baseline models specifically for each dataset, aiming at a valid comparison with our approach. However, one may still wonder whether the improvements of ESAN are due to an overall larger number of parameters. This aspect is investigated in the following for the case of GIN base graph encoders.

First, we compared the number of trainable parameters of the best baseline model with that of our best ESAN configuration; numbers are reported in Table 9. We found the best ESAN configuration to utilise *fewer* learnable parameters in three out of seven datasets: IMDB-B, IMDB-M and PTC.

In order to verify the fairness of our results on the remaining datasets, we conducted additional experiments, training larger GIN architectures obtained by enlarging the dimension of the hidden layers in a way to approximately match the overall number of ESAN parameters. We dub these models "BIG-GIN". The results are reported in Table 10. We observed that, on average, adding more parameters to baselines produced slightly worse results and did not change the ranking of the methods. Using baselines with more parameters yielded marginally worse results for two datasets (PROTEINS, NCI109) and only insignificantly better results on the remaining ones (NCI1, MU-TAG). This demonstrates that, while a larger number of parameters tend to correlate with larger model capacity, it does not necessarily correlate with generalization performance, which is what we are really interested in.

We finally refer readers to Table 3, where we report the results obtained on the ZINC12K datasets. On this benchmark, all ESAN configurations significantly outperformed their GIN base encoder while being all compliant to the 100k parameter budget.

### C.3.3 TIMING

We focus our analysis on NCI1 dataset from Morris et al. (2020a) (4110 graphs, 30 nodes/edges per graph on average) and estimated the time to perform a single training epoch on a RTX 2080 GPU. We used GIN as a base encoder and compared the times of: (1) DS-GNN and DSS-GNN without sampling (100% subgraphs – full bags); (2) DS-GNN and DSS-GNN with sampling (20% of subgraphs); (3) the baseline GIN method. We kept the hyperparameters the same for all methods

to allow a fair comparison. Results are reported in Table 11. As can be seen in the table, our method takes around 2x the time of the corresponding GIN baseline for DS-GNN, and around 3x for DSS-GNN. Sampling reduces these times by 10-15%, showcasing how our stochastic version can be beneficial in practice. Note that these times consider a single process completing a single training epoch. In our implementation, we perform the cross validation in parallel, making use of multiprocessing. The overall speedup gained by the parallelism comes at the cost of a fixed overhead. This added overhead reduces significantly the gap between our method and the baseline. In practice, we take less than twice the time of GIN on the NCI1 dataset, even without sampling. Indeed, DS-GNN (GIN) takes around 50 minutes to perform the entire 10-fold cross validation with 350 epochs for all policies and full bags, with DSS-GNN (GIN) taking a few minutes longer. GIN takes around 30 minutes for the same amount of work.

Further timing experiments on ZINC12K confirmed the empirically tractable computational complexity of our approach: for a training epoch on 10k graphs, DS-GNN required only between 3 and 3.5 seconds (depending on the policy), while DSS-GNN required between 3.7 and 4.25 seconds (the base encoder ran in slightly less than 1.5 seconds). Full results are reported in Table 11.

## C.4 DISCUSSION

Do our experimental results answer the research questions we raised at the beginning of Section 4? We discuss them in specific here below.

(1) *Is our approach more expressive than the base graph encoders in practice?* ESAN perfectly solves the RNI and CSL expressiveness benchmarks with a variety of subgraph selection policies, while the base encoder (GIN) performs as a random guesser (see Table 6). These results provide a (strong) positive answer to this question.

(2) *Can our approach achieve better performance on real benchmarks?* On real world graph datasets, ESAN has outperformed its base encoder in most of its configurations. Superior performance has been recorded across benchmarks, independently of the choice of the base encoder and the subgraph selection policy. In particular, we found ESAN to outperform its base graph encoder in 91% and 75% of the cases for, respectively, the DSS-GNN and DS-GNN variants. Overall, ESAN attains performance comparable to state-of-the-art methods on the TUDatasets and, in particular, on ZINC12K, where it significantly outperforms all provably expressive, domain agnostic GNNs. ESAN appears generally less competitive on the OGB dataset with respect to state-of-the-art methods (Bodnar et al., 2021a; Bouritsas et al., 2022; Beaini et al., 2021). We hypothesise that a different choice of base graph encoders may lead to further improvements on this benchmark and defer such investigations to future work.

(3) *How do the different subgraph selection policies compare?* This question has a less definite answer. We found certain policies to attain more competitive results on some benchmarks (e.g., ND on PROTEINS for DS-GNN (GIN) and EGO(+) on ZINC), while they all seemed to perform equally well on others (e.g., on IMDB-M and on NCI1 for the DSS-GNN variant). Our results do not strongly support correlations between the performance of a specific policy and the application domain. At the same time, the EGO(+) policies were found generally more consistent in their results across datasets.

(4) *Can our efficient stochastic variant offer the same benefits as the full approach?* Our results suggest a generally affirmative answer. From Table 7, it emerges that, barring a few notable exceptions, our subgraph sampling scheme has marginal to no impact to the overall performance of ESAN, even for drastic sampling rates. This is particularly evident over the IMDB-B and IMDB-M social benchmarks, as well as the PROTEINS bioinformatics one. In these and other cases, subgraph sampling sometimes slightly outperforms even the full, deterministic approach. Similar conclusions can be drawn from Table 8. An interesting trend is worth reporting: In the case of the DS-GNN variant, subgraph sampling negatively affects EGO(+) policies, sometimes consistently. This is clearly evident on the MUTAG, NCI1 and NCI109 benchmarks, as well as the OGBG-MOLHIV one. Interestingly, the DSS-GNN variant appears more robust in this regard (see paragraph below).

(5) *Are there any empirical advantages to DSS-GNN versus DS-GNN?* Overall, we observe a tendency for DSS-GNN to perform better than DS-GNN. This observation is more pronounced in certain conditions and datasets, and on ZINC12K in particular (see Table 3). Another interesting

finding is that DSS-GNN appears significantly more robust across different scenarios. As already noted above, this variant does not exhibit the same performance drop showed by DS-GNN with EGO(+) policies, and is, in particular, less sensitive to subgraph sampling (see Table 7). In certain cases, we observe DSS-GNN to be (more) robust to the choice of the base graph encoder, as it can be noted by comparing the performance of the two variants on the OGB datasets with GCN encoders (see Table 2 and Table 5). These advantages come at the cost of a slightly larger computational complexity, as observed in Table 11.

When is the ESAN framework advantageous, in general? Given the above, we can conclude that applying our approach to real world datasets is generally beneficial whenever some additional computational resources are available. The DSS-GNN variant, when equipped with EGO(+) policies, is a competitive and versatile architecture, showing strong performance across various benchmarks and conditions. DS-GNN may offer slightly more favourable computational complexity, and it generally works well in conjunction with ND and ED policies.

More specifically, our results indicate that the application of ESAN is recommended in those cases where the task at hand may require expressive power beyond 1-WL. There is some empirical evidence that certain molecular benchmarks fall in this category; our results on the ZINC12K dataset validate this hypothesis by showing that our approach strongly outperforms GIN, while performing best amongst all provably expressive, domain agnostic GNNs (see Table 3). Finally, ESAN is especially suitable in those cases where no domain knowledge is available about graph substructures relevant to the task being solved. As it requires little pre-engineering, our method represents a valid, flexible and provably expressive approach in these conditions.

We are not aware of any particular scenario where the application of ESAN has proved to be consistently unsuccessful. We have observed it to not work well in conjunction with GCN encoders over the OGB datasets, but this would require further future investigation before drawing definite conclusions.

# D  WL VARIANTS AND PROOFS FOR SECTION 3.2

This appendix includes additional details for Section 3.1 and proofs for the theoretical results in Section 3.2 of the main paper, along with additional context and results.

## D.1  PRELIMINARIES

Let us first introduce some notation and required definitions. We begin with the definition of *vertex coloring* (Rattan & Seppelt, 2021).

**Definition 1** (Vertex coloring)**.** *A vertex coloring is a function mapping a graph and one of its nodes to a "color" from a fixed color palette.*

Generally, we can define a vertex coloring as a function $c : \mathcal{V} \to C, (G, v) \mapsto c_v^G$, where $\mathcal{V}$ is the set of all possible tuples of the form $(G, v)$ with $G = (V, E) \in \mathcal{U}$ (the set of all finite graphs) and $v \in V$. Concretely we write $c : (G, v) \mapsto c_v^G$ and we sometimes drop the superscript $G$ whenever clear from the context. In order to design and compare vertex colorings, it is important to introduce the notion of *refinement*.

**Definition 2** (Vertex color refinement)**.** *Let c, d be two vertex colorings. We say that d refines c (or d is a refinement of c) when for all graphs $G = (V^G, E^G), H = (V^H, E^H)$ and all vertices $v \in V^G, u \in V^H$ we have that $d_v^G = d_u^H \implies c_v^G = c_u^H$. We write $d \sqsubseteq c$.*

---

**Algorithm 1** WL Test

**Require:** Graphs $G^1, G^2$, initial $\iota : v \mapsto l_v$ (optional)
1: $t \leftarrow 0$, *converged* $\leftarrow$ **false**
2: **for** $v \in G^1 \cup G^2$ **do**
3: $\quad c_v^0 \leftarrow \bar{c}$ or $\iota(v)$ if provided
4: $c_{G^1}^0 \leftarrow \{\!\!\{ c_v^0 \mid v \in G^1 \}\!\!\}$
5: $c_{G^2}^0 \leftarrow \{\!\!\{ c_v^0 \mid v \in G^2 \}\!\!\}$
6: **while** $c_{G^1}^t = c_{G^2}^t$ and not *converged* **do**
7: $\quad$ **for** $i \in \{1, 2\}$ **do**
8: $\quad\quad$ **for** $v \in G^i$ **do**
9: $\quad\quad\quad N_v^t \leftarrow \{\!\!\{ c_w^t \mid w \in \mathcal{N}^i(v) \}\!\!\}$
10: $\quad\quad\quad c_v^{t+1} \leftarrow \text{HASH}(c_v^t, N_v^t)$
11: $\quad\quad c_{G^i}^{t+1} \leftarrow \{\!\!\{ c_v^{t+1} \mid v \in G^i \}\!\!\}$
12: $\quad$ **if** $c_{G^1}^t \sqsubseteq c_{G^1}^{t+1}$ and $c_{G^2}^t \sqsubseteq c_{G^2}^{t+1}$ **then**
13: $\quad\quad$ *converged* $\leftarrow$ **true**
14: $\quad t \leftarrow t + 1$
15: **if** $c_{G^1}^t \neq c_{G^2}^t$ **then**
16: $\quad$ **return** non-isomorphic
17: **else**
18: $\quad$ **return** possibly isomorphic

---

When working with a specific graph pair $G^1, G^2$, we write $d \sqsubseteq_{G^1, G^2} c$ when for all $v \in V^{G^1}$ and all $u \in V^{G^2}$, if $d_v^{G^1} = d_u^{G^2}$, then $c_v^{G^1} = c_u^{G^2}$.

The 1-WL test, sometimes known as "naïve vertex refinement" (and sometimes simply as "WL" in this manuscript) is an efficient heuristic to test (non-)isomorphism between graphs. We report the related pseudocode in Algorithm 1.

The algorithm represents a graph with the multiset (or histogram) of colors associated with its nodes. colors naturally induce a partitioning of the nodes into color classes, in the sense that two nodes belong to the same partition if and only if they are equally colored. We denote with $c^{-1}(v)$ the partition to which node $v$ belongs to, according to coloring $c$. Starting from an initial vertex coloring, the algorithm iterates rounds whereby the node partitioning gets finer and finer. These rounds are referred to as "refinement steps" since, if $c$ indicates the colorings computed by the WL algorithm, $c^{t+1} \sqsubseteq_{G, G} c^t$:

**Proposition 4.** *Let G be a graph and $c_S^t$ be the vertex coloring computed by WL. Then, $\forall t > 0$, $c^{t+1} \sqsubseteq_{G, G} c^t$.*

*Proof.* For those nodes $v, u$ in the vertex set of G such that $c_v^{t+1} = c_u^{t+1}$ we have $\text{HASH}(c_v^t, N_v^t) = \text{HASH}(c_u^t, N_u^t)$ and, necessarily, $c_v^t = c_u^t$ due to the injectivity of the HASH function. $\qquad\square$

The refinement is at convergence when the node partitioning is *stable*, that is, when it does not vary across steps. This means that both $c^{t+1} \sqsubseteq_{G, G} c^t$ and $c^t \sqsubseteq_{G, G} c^{t+1}$ hold. At convergence on both the

two input graphs, the test deems them non-isomorphic if assigned distinct multisets of node colors, and it is inconclusive otherwise.

We report the full DSS-WL test in Algorithm 2. Let us first note that both DSS-WL and DS-WL produce color refinements at the level of each subgraph:

**Proposition 5.** *Let $G$ be a graph, $\pi$ be a subgraph selection policy and $\mathcal{G}$ be the bag induced by $\pi$ on $G$. Let $c_S^t$ be the vertex coloring computed by DSS-WL (DS-WL) over subgraph $S \in \mathcal{G}$ at round $t$. Then, $\forall S \in \mathcal{G}, \forall t > 0, \ c_S^{t+1} \sqsubseteq_{S,S} c_S^t$.*

*Proof.* In the case of DS-WL, the assertion is immediately proved by observing that the algorithm performs independent WL refinements on each subgraph, and we know the standard WL procedure is indeed a vertex refinement one.

The assertion trivially holds also in the case of DSS-WL. Suppose there exist nodes $v, u$ in the vertex set of $S$ such that $c_{v,S}^{t+1} = c_{u,S}^{t+1}$. This implies $\mathrm{HASH}(c_{v,S}^t, N_{v,S}^t, C_v^t, M_v^t) = \mathrm{HASH}(c_{u,S}^t, N_{u,S}^t, C_u^t, M_u^t)$ and, necessarily, $c_{v,S}^t = c_{u,S}^t$ due to the injectivity of the HASH function. □

An immediate consequence of the above considerations is that, for vertex refinement algorithms, if two vertices belong to distinct partitions at time step $t$, they will for any other $\bar{t} > t$. This allows us to conclude that both DSS-WL and DS-WL tests eventually terminate on any pair of (finite) input graphs.

**Proposition 6.** *Let $G^1, G^2$ be any pair of finite graphs. There exists $\bar{t} < \infty$ such that the DSS-WL (DS-WL) test converges when run on $G^1, G^2$ with any subgraph selection policy $\pi$.*

---

**Algorithm 2** DSS-WL Test

**Require:** $G^1, G^2$, policy $\pi$, $\iota : v \mapsto l_v$ (optional)
1: $t \leftarrow 0, \ \ converged \leftarrow$ **false**
2: $\mathcal{G}^1 \leftarrow \pi(G^1), \ \ \mathcal{G}^2 \leftarrow \pi(G^2)$
3: **for** $S \in \mathcal{G}^1 \cup \mathcal{G}^2$ **do**
4:      **for** $v \in S$ **do**
5:          $c_{v,S}^0 \leftarrow \bar{c}$ or $\iota(v)$ if provided
6:      $c_S^0 \leftarrow \mathrm{HASH}(\{\!\!\{ c_{v,S}^0 \mid v \in S \}\!\!\})$
7: $c_{G^1} \leftarrow \{\!\!\{ c_S^0 \mid S \in \mathcal{G}^1 \}\!\!\}, \ \ c_{G^2} \leftarrow \{\!\!\{ c_S^0 \mid S \in \mathcal{G}^2 \}\!\!\}$
8: **while** $c_{G^1} = c_{G^2}$ and not $converged$ **do**
9:      **for** $i \in \{1, 2\}$ **do**
10:          **for** $S \in \mathcal{G}^i$ **do**
11:              **for** $v \in S$ **do**
12:                  $N_{v,S}^t \leftarrow \{\!\!\{ c_{w,S}^t \mid w \in \mathcal{N}^S(v) \}\!\!\}$
13:                  $C_v^t \leftarrow \{\!\!\{ c_{v,R}^t \mid R \in \mathcal{G}^i \}\!\!\}$
14:                  $M_v^t \leftarrow \{\!\!\{ C_w^t \mid w \in \mathcal{N}(v) \}\!\!\}$
15:                  $c_{v,S}^{t+1} \leftarrow \mathrm{HASH}(c_{v,S}^t, N_{v,S}^t, C_v^t, M_v^t)$
16:              $c_S^{t+1} \leftarrow \mathrm{HASH}(\{\!\!\{ c_{v,S}^{t+1} \mid v \in S \}\!\!\})$
17:          $c_{G^i} \leftarrow \{\!\!\{ c_S^{t+1} \mid S \in \mathcal{G}^i \}\!\!\}$
18:      **if** $\forall S \in \mathcal{G}^1 \cup \mathcal{G}^2, \ c_S^t \sqsubseteq c_S^{t+1}$ **then**
19:          $converged \leftarrow$ **true**
20:      $t \leftarrow t + 1$
21: **if** $c_{G^1} \neq c_{G^2}$ **then**
22:      **return** non-isomorphic
23: **else**
24:      **return** possibly isomorphic

---

*Proof.* Let $\mathcal{G}^1, \mathcal{G}^2$ be the bags induced by $\pi$ on $G^1, G^2$. The two tests converge when, on all subgraph, the node partitioning induced by the coloring does not vary across two consecutive steps. Due to Proposition 5, we know that their colorings are vertex refinements on each of the subgraphs. This implies that, at iteration $t + 1$, the partitioning on a subgraph is either as fine as, or finer than the one at iteration $t$. In other words, if $N_{t,S}$ is the number of color classes at round $t$ on subgraph $S$, we have that $N_{t+1,S} \geq N_{t,S}$. At the same time, the value $N_{t,S}, \forall t \geq 0$ is upper bounded by $N_S < \infty$, i.e. the cardinality of the vertex set of $S$. This implies that the number of possible iterations before convergence must be upper-bounded by the value $\nu = \max_i \left( \sum_{S \in \mathcal{G}^i} N_S \right)$, which is, itself, finite. □

The concept of vertex color refinement is linked to the ability to distinguish between non-isomorphic graphs. This is a consequence of the following Lemma, whose proof is adapted from Bodnar et al. (2021b).

**Lemma 2.** *Let $c, d : \mathcal{V} \to \mathbb{N}$ be coloring functions implementing maps in the form $(A, a) \mapsto c_a, (A, a) \mapsto d_a$, with $(A, a) \in \mathcal{V}$ such that $A$ is a set and $a \in A$. Let $A, B$ be two sets such that*

$(A, \cdot), (B, \cdot) \in \mathcal{V}$ and $\mathcal{A}_\phi, \mathcal{B}_\phi$ be multisets of colors defined as: $\mathcal{A}_\phi = \{\!\{\phi_a | a \in A\}\!\}, \mathcal{B}_\phi = \{\!\{\phi_b | b \in B\}\!\}, \phi \in \{c, d\}$ Suppose that for all $a, b \in A \cup B$, it holds that $d_a = d_b \implies c_a = c_b$ (we write $d \sqsubseteq_{A,B} c$). Then $\mathcal{A}_c \neq \mathcal{B}_c \implies \mathcal{A}_d \neq \mathcal{B}_d$.

*Proof.* If $\mathcal{A}_c \neq \mathcal{B}_c$, we assume, w.l.o.g., that there is a color $\bar{c}$ which is represented more in $\mathcal{A}_c$ than in $\mathcal{B}_c$. In other words, if we define the two sets:

$$A^{\bar{c}} = \{a \in A | c_a = \bar{c}\}, \quad B^{\bar{c}} = \{b \in B | c_b = \bar{c}\}$$

then $|A^{\bar{c}}| > |B^{\bar{c}}|$. These sets naturally define the partitioning:

$$A = A^{\bar{c}} \cup \tilde{A}^{\bar{c}}, \quad B = B^{\bar{c}} \cup \tilde{B}^{\bar{c}}$$

where $\tilde{A}^{\bar{c}} = \{a \in A | c_a \neq \bar{c}\}, \tilde{B}^{\bar{c}} = \{b \in B | c_b \neq \bar{c}\}$. Let us now consider the multisets:

$$\mathcal{A}_d^* = \{\!\{d_a | a \in A^{\bar{c}}\}\!\}, \quad \mathcal{B}_d^* = \{\!\{d_b | b \in B^{\bar{c}}\}\!\}$$

First, each of these multisets have the same cardinality of the generating sets, that is $|\mathcal{A}_d^*| = |A^{\bar{c}}|, |\mathcal{B}_d^*| = |B^{\bar{c}}|$ so, we necessarily have $\mathcal{A}_d^* \neq \mathcal{B}_d^*$. Second, in view of the fact that $d \sqsubseteq_{A,B} c$, although the two multisets may generally contain more than one color, these colors cannot appear outside of these last ones. More formally, by construction we have that

$$\forall x \in A^{\bar{c}} \cup B^{\bar{c}}, \quad y \in \tilde{A}^{\bar{c}} \cup \tilde{B}^{\bar{c}} \quad c_x \neq c_y$$

and, due to $d \sqsubseteq_{A,B} c$ it must also hold that:

$$\forall x \in A^{\bar{c}} \cup B^{\bar{c}}, \quad y \in \tilde{A}^{\bar{c}} \cup \tilde{B}^{\bar{c}} \quad d_x \neq d_y$$

This implies that, if we define multisets

$$\tilde{\mathcal{A}}_d^* = \{\!\{d_a | a \in \tilde{A}^{\bar{c}}\}\!\}, \quad \tilde{\mathcal{B}}_d^* = \{\!\{d_b | b \in \tilde{B}^{\bar{c}}\}\!\}$$

then $(\mathcal{A}_d^* \cup \mathcal{B}_d^*) \cap (\tilde{\mathcal{A}}_d^* \cup \tilde{\mathcal{B}}_d^*) = \varnothing$.

Now, we have that $\mathcal{A}_d = \mathcal{A}_d^* \cup \tilde{\mathcal{A}}_d^*, \mathcal{B}_d = \mathcal{B}_d^* \cup \tilde{\mathcal{B}}_d^*$, and $\mathcal{A}_d^* \neq \mathcal{B}_d^*$. Say that, w.l.o.g., this last inequality holds due to a color $\bar{d}$ appearing $n_A$ times in $\mathcal{A}_d^*$ and $n_B$ times in $\mathcal{B}_d^*$ with $n_A > n_B$. Then, a necessary condition for $\mathcal{A}_d = \mathcal{B}_d$ is that $\bar{d}$ appears $n_A - n_B$ times more in $\tilde{\mathcal{B}}_d^*$ than in $\tilde{\mathcal{A}}_d^*$. However, this condition cannot be met due to the disjointness of the multisets as shown above. $\square$

**Definition 3.** *Let $c$ be a vertex coloring. A readout map $\mathcal{A}_c : (V, E) \mapsto \phi(\{\!\{c_v^G | v \in V\}\!\})$ is function representing a graph $G = (V, E)$ with the multiset of node colors generated by $c$ on $V$, or an injection thereof ($\phi$).*

Readout maps allow us to leverage colorings to test (non-)isomorphism between graphs: we can define algorithmic tests whereby two graphs are distinguished when their readout representations are distinct. This is the case of the WL test, where $\phi$ is the identity function and $c$ is the coloring induced by a certain number of rounds of the refinement procedure.

The following is an immediate consequence of Lemma 2.

**Corollary 2.** *Let $c, d$ be two vertex coloring functions such that $d \sqsubseteq c$. Let $\mathcal{A}_c, \mathcal{A}_d$ be readout maps associated with, respectively, vertex colorings $c, d$. Then, any two graphs distinguished by $\mathcal{A}_c$ are also distinguished by $\mathcal{A}_d$. We say $\mathcal{A}_d$ refines $\mathcal{A}_c$ and write $\mathcal{A}_d \sqsubseteq \mathcal{A}_c$.*

When $\mathcal{A}_d \sqsubseteq \mathcal{A}_c$ then $\mathcal{A}_d$ is said to be at least as powerful as $\mathcal{A}_c$. When the opposite holds, that is $\mathcal{A}_c \sqsubseteq \mathcal{A}_d$, then the two have the same discriminative power and we write $\mathcal{A}_c \equiv \mathcal{A}_d$. On the contrary, if there exist a pair of graphs distinguished by $\mathcal{A}_d$ which are indistinguishable by $\mathcal{A}_c$, then we say $\mathcal{A}_d$ is *strictly* more powerful than $\mathcal{A}_c$, and write $\mathcal{A}_d \sqsubset \mathcal{A}_c$.

Let us lastly introduce some general, auxiliary lemmas that will be required for the proofs in the following subsection.

**Lemma 3.** *Let $A$, $B$ be two multisets with elements in $\mathcal{D}$ such that $A = \{\!\{a_i\}\!\}_{i=1}^m = \{\!\{b_i\}\!\}_{i=1}^m = B$. For any function $\varphi : \mathcal{D} \to \mathcal{C}$ it holds that $\varphi(A) = \{\!\{\varphi(a_i)\}\!\}_{i=1}^m = \{\!\{\varphi(b_i)\}\!\}_{i=1}^m = \varphi(B)$.*

*Proof.* Let $\varphi$ be any *function* (not necessarily injective). Suppose $A = B$, but $\varphi(A) \neq \varphi(B)$. There is some $\bar{c}$ such that w.l.o.g $\bar{c}$ appears more in $\varphi(A)$ than $\varphi(B)$. Defining

$$\mathcal{A} = \{\!\{d_A : \varphi(d_A) = \bar{c}, d_A \in A\}\!\}, \quad \mathcal{B} = \{\!\{d_B : \varphi(d_B) = \bar{c}, d_B \in B\}\!\} \tag{5}$$

Thus, $\mathcal{A} \neq \mathcal{B}$ because $|\mathcal{A}| > |\mathcal{B}|$. Note that $A = \mathcal{A} \cup \tilde{\mathcal{A}}$ for the complementary set $\tilde{\mathcal{A}}$ that is disjoint from $\mathcal{A}$ defined as $\tilde{\mathcal{A}} = \{\!\{d_A : \varphi(d_A) \neq \bar{c}, d_A \in A\}\!\}$, and similarly $B = \mathcal{B} \cup \tilde{\mathcal{B}}$.

Since $|\mathcal{A}| > |\mathcal{B}|$, we can choose a $d \in \mathcal{A}$ such that the multiplicity $\#_d(\mathcal{A})$ of $d \in \mathcal{A}$ is greater than $\#_d(\mathcal{B})$. Further, since $A = \mathcal{A} \cup \tilde{\mathcal{A}}$ and $\mathcal{A}$ is disjoint from $\tilde{\mathcal{A}}$, we have that $\#_d(A) = \#_d(\mathcal{A})$. Likewise, $\#_d(B) = \#_d(\mathcal{B})$. However, this is a contradiction, because then

$$\#_d(A) = \#_d(\mathcal{A}) > \#_d(\mathcal{B}) = \#_d(B), \tag{6}$$

so $A \neq B$. $\qquad\square$

**Lemma 4.** *Let $A$, $B$ be two multisets with elements in $\mathcal{D}^1 \times \ldots \times \mathcal{D}^k$ such that $A = \{\!\{(a_i^1, \ldots, a_i^k)\}\!\}_{i=1}^m, B = \{\!\{(b_i^1, \ldots, b_i^k)\}\!\}_{i=1}^m$. Let $A^j = \{\!\{a_i^j\}\!\}_{i=1}^m, B^j = \{\!\{b_i^j\}\!\}_{i=1}^m, \forall j = 1, \ldots, k$. Then $A = B$ implies $A^j = B^j$ for each $j$.*

*Proof.* The above is proven by considering that, for any $j$, it is possible to construct the mapping $\varphi^j : \mathcal{D}^1 \times \ldots \times \mathcal{D}^k \to \mathcal{D}^j$ such that $(a_i^1, \ldots, a_i^k) \mapsto a_i^j$. Then, $\varphi^j(A) = A^j$ and $\varphi^j(B) = B^j$. At the same time, by Lemma 3, $A = B$ implies $\varphi^j(A) = \varphi^j(B)$ and thus $A^j = B^j$. $\qquad\square$

### D.2 Proofs for results in Section 3.2

#### D.2.1 DSS-WL vs. DS-WL

We start by formally proving that DSS-WL is at least as expressive as DS-WL. We will require the following lemma, which describes a refinement relation between DSS-WL and WL node colorings at the level of subgraphs.

**Lemma 5.** *Let $\pi$ be any subgraph selection policy, $G^1, G^2$ be any two graphs and $\mathcal{D}^1, \mathcal{D}^2$ the $\pi$-bags thereof. Let $a^{(t)}, b^{(t)}$ generally refer to the coloring generated by $t$ steps of, respectively, the WL and DSS-WL algorithms, with WL running independently on each subgraph. Then, for any $t \geq 0$, $\forall G = (V_G, E_G), H = (V_H, E_H) \in \mathcal{D}_1 \cup \mathcal{D}_2, \ b^{(t)} \sqsubseteq_{G,H} a^{(t)}$.*

*Proof.* We seek to show that $\forall v \in V_G, u \in V_H, b_v^{G,(t)} = b_u^{H,(t)} \implies a_v^{G,(t)} = a_u^{H,(t)}$. As usual, let us prove the lemma by induction on $t$.

*(Base case)* The thesis trivially holds for $t = 0$ given that all nodes in all subgraphs are assigned the same color. Let us assume the thesis holds true for $t > 0$, we seek to show it also holds for $t + 1$.

*(Inductive Step)* Suppose that, for nodes $v$, $u$ we have $b_v^{G,(t+1)} = b_u^{H,(t+1)}$. The equality can be rewritten as $\mathrm{HASH}(b_v^{G,(t)}, N_v^{G,(t)}, C_v^{(t)}, M_v^{(t)}) = \mathrm{HASH}(b_u^{H,(t)}, N_u^{H,(t)}, C_u^{(t)}, M_u^{(t)})$, which implies the equality between each of the inputs to the HASH function. Importantly, this means that $b_v^{G,(t)} = b_u^{H,(t)}$ and $N_v^{G,(t)} = N_u^{H,(t)}$. The inductive hypothesis immediately gives us $a_v^{G,(t)} = a_w^{H,(t)}$ and $\{\!\{a_i^{G,(t)} | i \in \mathcal{N}^G(v)\}\!\} = \{\!\{a_j^{H,(t)} | j \in \mathcal{N}^H(w)\}\!\}$, by the definition of $N_v^{G,(t)}$ and $N_w^{H,(t)}$ and in view of Lemma 2. These are equalities between the (only) inputs in the WL color refinement step, which applies an injective HASH to obtain $a_v^{G,(t+1)}$ and $a_u^{H,(t+1)}$. It then immediately follows that $a_v^{G,(t+1)} = a_u^{H,(t+1)}$. $\qquad\square$

As a consequence, we have the following corollary:

**Corollary 3.** *Let $\pi$ be any subgraph selection policy, $G^1, G^2$ be any two graphs and $\mathcal{D}^1, \mathcal{D}^2$ the $\pi$-bags thereof. Let $a_G^{(t)}$ and $b_G^{(t)}$ refer to the subgraph colors assigned to subgraphs $G$ at step $t$ by, respectively, DS-WL and DSS-WL. For any pair of subgraphs $G, H \in \mathcal{D}_1 \cup \mathcal{D}_2$, and time step $t \geq 0$, if DS-WL assigns a distinct color to $G$ and $H$ then DSS-WL does so as well: $a_G^{(t)} \neq a_H^{(t)} \implies b_G^{(t)} \neq b_H^{(t)}$.*

*Proof.* In both algorithms, subgraph colors are obtained by injectively hashing the multisets of node colors therein. Furthermore, in DS-WL, the node colors at time step $t$ on generic subgraph $G$ are obtained by independently running $t$ rounds of the WL procedure on $G$. The thesis therefore holds in view of Lemma 5, Lemma 2 and Lemma 3. □

Equipped with the above, we can now prove the following:

**Proposition 7.** *Let $\pi$ be any subgraph selection policy. When both are run on the same $\pi$-induced bags, DSS-WL is at least as powerful as DS-WL in distinguishing non-isomorphic graphs, i.e., any two non-isomorphic graphs $G^1, G^2$ distinguished by DS-WL via policy $\pi$ are also distinguished by DSS-WL via the same policy.*

*Proof.* Let $\mathcal{D}^1 = \{\!\!\{ G^1_k \}\!\!\}^m_{k=1}, \mathcal{D}^2 = \{\!\!\{ G^2_h \}\!\!\}^m_{h=1}$ be the bags induced by $\pi$ on $G_1, G_2$. Let $a^{(t)}$ and $b^{(t)}$ refer to the *subgraph* coloring computed at step $t$ by, respectively, DS-WL and DSS-WL (these assign each subgraph a color by hashing the multiset of node colors at time $t$). Also, suppose DS-WL distinguishes between $G^1, G^2$ at time step $T$. The theorem immediately follows in light of Corollary 3 and Lemma 2. Let $\mathcal{D}^1, \mathcal{D}^2$ correspond to, respectively sets $A, B$ from Lemma 2, and let $a^{(T)}, b^{(T)}$ correspond to, respectively, colorings $c, d$ from the same proposition. If $G^1, G^2$ are distinguished by DS-WL at step $T$, then we have $\mathcal{A}_c \neq \mathcal{B}_c$. Also, in view of Corollary 3, it holds that $b^{(T)} \sqsubseteq_{\mathcal{D}^1, \mathcal{D}^2} a^{(T)}$, that is $d \sqsubseteq_{A,B} c$. All hypotheses in Lemma 2 hold and we can then conclude that $\mathcal{A}_d \neq \mathcal{B}_d$ that is $\mathcal{A}_{b^{(T)}} \neq \mathcal{B}_{b^{(T)}}$. This shows DSS-WL assigns the two bags distinct multisets of colors, and thus, as DS-WL, it distinguishes between the two graphs $G^1, G^2$. □

### D.2.2 EXPRESSIVENESS OF WL VARIANTS

In this subsection we will characterize the expressive power of our variants in relation to the standard WL test. We will first show that DSS-WL and DS-WL are indeed at least as powerful as the standard WL test when equipped with non-trivial subgraph selection policies. These will form the necessary premises to prove Theorem 1.

We start by characterizing DSS-WL, and giving some required definitions and lemmas.

**Definition 4.** *Let $G = (V, E)$ be any graph. Let $\pi$ be any subgraph selection policy generating an $m$-subgraph bag $\mathcal{S}^\pi_G = \{G_i = (V_i, E_i)\}^m_{i=1}$ over $G$. Policy $\pi$ is said to be* vertex set preserving *if $V_i = V$ for each $i = 1, \ldots, m$.*

Intuitively, a vertex set preserving policy is a policy which does not perturb the original vertex set; subgraphs are obtained by taking subsets of edges. Note that the ND policy is vertex set preserving if, instead of removing one node and all the related connectivity, it removes its incident edges only and leaves the isolated node in the vertex set.

An important component of the DSS-WL algorithm is the HASH input $C^{(t)}_v$, which represents the multiset of colors for node $v$ across the subgraphs in the bag.

**Definition 5.** *Let $G = (V, E)$ be any graph. Let $\pi$ be a* vertex set preserving *policy generating an $m$-subgraph bag $\mathcal{S}^\pi_G$. The* needle node color *for node $v \in V$ at time step $t \geq 0$, denoted $C^{(t)}_v$, is defined as $\{\!\!\{ c^{(t)}_{v,S} \}\!\!\}_{S \in \mathcal{S}^\pi_G}$, where $c^{(t)}_{v,k}$ is the node color of node $v$ on subgraph $k$ at time step $t$. We remark that the* needle node color $C^{(t)}_v$ *is employed by the DSS-WL algorithm to obtain a refinement at time step $(t+1)$.*

We now introduce a first result allowing us to compare DSS-WL and WL.

**Lemma 6.** *Let $\pi$ be any* vertex set preserving *policy. Let $b^\pi$ denote the (needle) node coloring from DSS-WL running on bags induced by $\pi$, and $a$ be the node coloring from the standard WL algorithm. Then $b^\pi \sqsubseteq a$, that is, for all graphs $G^1 = (V^1, E^1)$ and $G^2 = (V^2, E^2)$ and all nodes $v \in V^1, w \in V^2$ we have that $b^\pi_v = b^\pi_w \implies a_v = a_w$.*

*Proof.* Let $G^1, G^2$ be any two graphs, $a^{(t)}$ refer to the WL coloring at iteration $t$, and $b^{(t)}$ refer to the needle coloring induced by DSS-WL at the same iteration (we drop the superscript $\pi$ to ease the notation). We seek to prove that $b^{(t)} \sqsubseteq a^{(t)} \forall t \geq 0$. We prove the lemma by induction on time step $t$.

In particular, let us refer with $a_v^{(t)}$ to the WL-color assigned to node $v$ at time step $t$, and with $b_v^{(t)}$ to the needle node color DSS-WL assigns to the same node at the same iteration. We remark that this last is defined as $b_v^{(t)} = C_v^{(t)} = \{\!\{c_{v,S}^{(t)}\}\!\}_{S \in \mathcal{S}_G^\pi}$, with $c_{v,k}^{(t)}$ the DSS-WL color assigned to node $v$ at time step $t$ on subgraph $S$.

*(Base case)* At $t = 0$, it clearly holds that $b^{(0)} \sqsubseteq a^{(0)}$ given that all nodes are initialised with the same constant color $\bar{c}$.

*(Step)* Let us assume the thesis holds true at time step $t$: $b_v^{(t)} = b_w^{(t)} \implies a_v^{(t)} = a_w^{(t)}$. We seek to prove that if that is the case, it also holds true at time step $t + 1$.

Suppose $b_v^{(t+1)} = b_w^{(t+1)}$. Unrolling one step backwards in time we get:

$$\{\!\{g(b_{v,S}^{(t)}, N_{v,S}^{(t)}, C_v^{(t)}, M_v^{(t)})\}\!\}_{S \in \mathcal{S}_{G^1}^\pi} = \{\!\{g(b_{w,R}^{(t)}, N_{w,R}^{(t)}, C_w^{(t)}, M_w^{(t)})\}\!\}_{R \in \mathcal{S}_{G^2}^\pi}$$

where $g \equiv \text{HASH}$. In view of Lemma 3, the following equality also holds:

$$\{\!\{g^{-1} \circ g(b_{v,S}^{(t)}, N_{v,S}^{(t)}, C_v^{(t)}, M_v^{(t)})\}\!\}_{S \in \mathcal{S}_{G^1}^\pi} = \{\!\{g^{-1} \circ g(b_{w,R}^{(t)}, N_{w,R}^{(t)}, C_w^{(t)}, M_w^{(t)})\}\!\}_{R \in \mathcal{S}_{G^2}^\pi}$$

since, due to the injectivity of $g$, the left inverse $g^{-1} : Im(g) \to D(g)$ exists (and is injective). Thus we have:

$$\{\!\{(b_{v,S}^{(t)}, N_{v,S}^{(t)}, C_v^{(t)}, M_v^{(t)})\}\!\}_{S \in \mathcal{S}_{G^1}^\pi} = \{\!\{(b_{w,R}^{(t)}, N_{w,R}^{(t)}, C_w^{(t)}, M_w^{(t)})\}\!\}_{R \in \mathcal{S}_{G^2}^\pi}$$

where, in view of Lemma 4, we also have:

$$\{\!\{C_v^{(t)}\}\!\}_{S \in \mathcal{S}_{G^1}^\pi} = \{\!\{C_w^{(t)}\}\!\}_{R \in \mathcal{S}_{G^2}^\pi}, \quad \{\!\{M_v^{(t)}\}\!\}_{S \in \mathcal{S}_{G^1}^\pi} = \{\!\{M_w^{(t)}\}\!\}_{R \in \mathcal{S}_{G^2}^\pi}$$

The elements of these multisets do not depend on index $k$, therefore we can rewrite the equalities as the set equalities:

$$\{(C_v^{(t)}, |\mathcal{S}_{G^1}^\pi|)\} = \{(C_w^{(t)}, |\mathcal{S}_{G^2}^\pi|)\}, \quad \{(M_v^{(t)}, |\mathcal{S}_{G^1}^\pi|)\} = \{(M_w^{(t)}, |\mathcal{S}_{G^2}^\pi|)\}$$

which obviously imply

$$C_v^{(t)} = C_w^{(t)}, \quad M_v^{(t)} = M_w^{(t)}$$

First, we note that $b_v^{(t)} = C_v^{(t)} = C_w^{(t)} = b_w^{(t)}$, which by the induction hypothesis gives us $a_v^{(t)} = a_w^{(t)}$.

Second, we note that $M_v^{(t)} = \{\!\{b_i^{(t)} | i \in \mathcal{N}(v)\}\!\}$ and $M_w^{(t)} = \{\!\{b_j^{(t)} | j \in \mathcal{N}(w)\}\!\}$. By the induction hypothesis, Lemma 2 gives us that $M_v^{(t)} = M_w^{(t)} \implies \{\!\{a_i^{(t)} | i \in \mathcal{N}(v)\}\!\} = \{\!\{a_j^{(t)} | j \in \mathcal{N}(w)\}\!\}$. This, along with the result above, implies $a_v^{(t+1)} = a_w^{(t+1)}$, since both the two inputs to hash $g$ at time step $t$ coincide. The proof terminates. $\square$

Finally, by leveraging the above lemma, we show that DSS-WL is at least as powerful as the WL test in distinguishing between non-isomorphic graphs, when equipped with vertex set preserving policies.

**Lemma 7.** *Let $\pi$ be any* vertex set preserving *policy. DSS-WL on $\pi$-induced bags is at least as powerful as WL in distinguishing non-isomorphic graphs, i.e., any two non-isomorphic graphs $G^1, G^2$ distinguished by WL are also distinguished by DSS-WL when run on the respective $\pi$-induced bags.*

*Proof.* Throughout the proof, we will denote with $a$ colorings generated by the WL algorithm, while $b$ will denote needle colorings generated by DSS-WL and $c$ its node colorings on subgraphs. Additionally, we denote as $\mathcal{D}^1, \mathcal{D}^2$ the $\pi$-bags of, respectively, $G^1, G^2$. We suppose the two bags have both cardinality $m$, and are defined as $\mathcal{D}^1 = \{\!\{G_k^1 = (V^1, E_k^1)\}\!\}_{k=1}^m, \mathcal{D}^2 = \{\!\{G_h^2 = (V^2, E_h^2)\}\!\}_{h=1}^m$. We do not explicitly focus on the case $|\mathcal{D}^1| \neq |\mathcal{D}^2|$ as the two graphs would then be (trivially) distinguished by DSS-WL.

Let us start by recalling that at time step $t$, two graphs are deemed non-isomorphic by the WL algorithm when the two are assigned two distinct multisets of *node colors*:

$$\{\!\{a_v^{(t)} | v \in V^1\}\!\} \neq \{\!\{a_w^{(t)} | w \in V^2\}\!\}$$

while the DSS-WL algorithm deems them non-isomorphic when the two are assigned two distinct multisets of *subgraph colors*, that is:

$$\{\!\!\{ b_{G_k^1}^{(t)} \}\!\!\}_{k=1}^m \neq \{\!\!\{ b_{G_h^2}^{(t)} \}\!\!\}_{h=1}^m$$

where $b_{G_j^i}^{(t)}$ generally indicates the color computed for subgraph $j$ of graph $G^i$: $\{\!\!\{ c_{v,j}^{(t)} | v \in V^i \}\!\!\}$.

Suppose WL distinguishes between the two graphs at iteration $T$. In view of Lemma 6, we know that $b^{(T)} \sqsubseteq a^{(T)}$, where $b^{(T)}$ is the needle node coloring generated by DSS-WL at time step $T$. At the same time, it can be shown that the needle coloring at $T$ is refined by the coloring generated by DSS-WL at $T+1$ on any pair of subgraphs: $\forall G, H \in \mathcal{D}^1 \cup \mathcal{D}^2 \ \ c^{(T+1)} \sqsubseteq_{G,H} b^{(T)}$. Let us shortly prove it. We seek to show that for all nodes $v$ in $G$, $u$ in $H$:

$$c_{v,G}^{(T+1)} = c_{u,H}^{(T+1)} \implies b_v^{(T)} = b_u^{(T)}$$

The antecedent implies the equality of all inputs to the HASH function at time step $T$, that is: $c_{v,G}^{(T)} = c_{u,H}^{(T)}, N_{v,G}^{(T)} = N_{u,H}^{(T)}, M_v^{(T)} = M_u^{(T)}$ and, importantly, $C_v^{(T)} = C_u^{(T)}$. By the definition of needle node colors, this last equality is rewritten as $b_v^{(T)} = b_u^{(T)}$, which proves the refinement.

This tells us that, by transitivity, that $\forall G, H \in \mathcal{D}^1 \cup \mathcal{D}^2 \ \ c^{(T+1)} \sqsubseteq_{G,H} a^{(T)}$. Therefore, by Lemma 2, if two graphs are deemed non-isomorphic by the WL algorithm at step $T$, then at step $T+1$, DSS-WL assigns a distinct color to any pair of subgraphs from the respective bags, that is:

$$b_{G_k^1}^{(T+1)} = \{\!\!\{ c_{v,k}^{(T+1)} | v \in V^1 \}\!\!\} \neq \{\!\!\{ c_{w,h}^{(T+1)} | w \in V^2 \}\!\!\} = b_{G_h^2}^{(T+1)} \quad \forall h, k \in \{1, \ldots, m\}$$

This immediately leads DSS-WL to distinguish between graphs $G^1, G^2$ as their corresponding multisets of subgraph colors are disjoint. $\square$

Let us now characterize the expressive power of DS-WL. We start by proving that, under certain subgraph selection policies, it can be made at least as expressive as the WL test. Recall that for a policy $\pi$, the augmented policy $\widehat{\pi}$ is defined by $\widehat{\pi}(G) = \pi(G) \cup \{G\}$, which augments $\pi$ with the original graph $G$.

**Lemma 8.** *Let $\widehat{\pi}$ be any augmented subgraph selection policy. DS-WL on $\widehat{\pi}$-induced bags is at least as powerful as WL in distinguishing non-isomorphic graphs, i.e., any two non-isomorphic graphs $G^1, G^2$ distinguished by WL are also distinguished by DS-WL when run on the respective $\widehat{\pi}$-induced bags.*

*Proof.* We focus on the non-trivial case in which the two graphs have the same number of edges (and nodes) and policy $\widehat{\pi}$, on the two graphs, generates bags of same cardinality. Let us remark that subgraph colors are obtained by DS-WL by running an independent WL procedure on each of them.

Let $G^1, G^2$ be two graphs distinguished by WL at round $T \geq 1$. Let $a^1, a^2, a^1 \neq a^2$ be the representations WL assigns to the two input graphs at time step $T$. Also, let us denote by $C^1 = \{\!\!\{ c_k^1 \}\!\!\}_{k=0}^m, C^2 = \{\!\!\{ c_k^2 \}\!\!\}_{k=0}^m$ the multisets of *subgraph* colors computed by DS-WL on the two input $\widehat{\pi}$-bags at round $T$. In particular, if $c_0^1, c_0^2$ are the subgraph colors associated with $G_0^1 = G^1, G_0^2 = G^2$, clearly, $c_0^1 = a^1, c_0^2 = a^2, c_0^1 \neq c_0^2$.

Now, any other subgraph from any bag has a number of edges smaller than the one in the original graphs. This implies that, if $S$ is a proper subgraph of $G$, then WL assigns it a color distinct from the one assigned to $G$ already at the first round. Then, $\forall k = 1, \ldots, m, i = 1, 2, c_k^i \notin \{c_0^1, c_0^2\}$. This entails the presence of a color in $C^1$ which is not in $C^2$ (and vice-versa), thus $C^1 \neq C^2$ which implies DS-WL distinguishes between the two graphs. $\square$

As for DS-WL, the inclusion of the original graphs in the generated bags guarantees a lower bound on the expressive power of the test, which corresponds to that of the standard WL algorithm. Nonetheless, there exist families of graphs undistinguishable by WL, which can be are separated by DS-WL when equipped with policies which do not include the original graph in the induced bags. Lemma 1 provides examples of them: $\mathrm{CSL}(n, 2)$ graphs can be distinguished from any $\mathrm{CSL}(n, k)$ with $k \in [3, n/2 - 1]$ with either ND, EGO and EGO+ policy. We recall the definition of these example graphs:

**Circulant Skip Link Graphs.** For parameters $n \geq 7$ and $k \in [2, n-2]$, $\mathrm{CSL}(n, k)$ is a 4-regular graph with vertex set $V = \{0, \ldots, n-1\}$ and edges between node $i$ and $i \pm 1 \mod n$ as well as between $i$ and $i \pm k \mod n$ (Murphy et al., 2019). For an infinite subset of non-isomorphic pairs of these graphs, we prove that DS-WL with two different policies can distinguish pairs that 1-WL cannot.

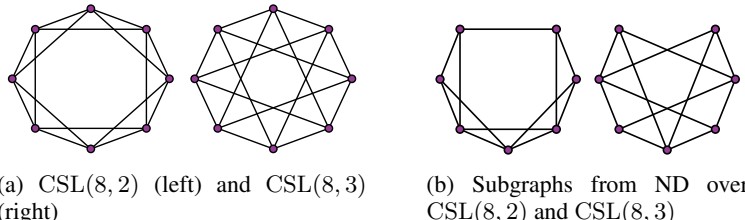

(a) $\mathrm{CSL}(8,2)$ (left) and $\mathrm{CSL}(8,3)$ (right)

(b) Subgraphs from ND over $\mathrm{CSL}(8,2)$ and $\mathrm{CSL}(8,3)$

Figure 4: Graphs $\mathrm{CSL}(8,2)$ and $\mathrm{CSL}(8,3)$ (left) and their node-deleted subgraphs (right).

*Proof of Lemma 1.* We start with a lemma that reduces the problem to considering a single subgraph of each CSL graph.

**Lemma 9.** *All node-deleted subgraphs of $G = \mathrm{CSL}(n, k)$ are isomorphic. All depth-1 ego-nets of $G = \mathrm{CSL}(n, k)$ are isomorphic.*

*Proof.* For $i \in \{0, \ldots, n-1\}$, there is an isomorphism from node-deleted subgraphs $G - v_0$ to $G - v_i$ given by mapping node $j$ in $G - v_0$ to node $j + i \mod n$ in $G - v_i$. This same isomorphism works for mapping the ego-net rooted at $v_0$ to the ego-net rooted at $v_i$.

$\square$

Let the nodes of $\mathrm{CSL}(n, 2)$ be denoted $v_0^{(1)}, \ldots, v_{n-1}^{(1)}$ and similarly denote the nodes of $\mathrm{CSL}(n, k)$ as $v_0^{(2)}, \ldots, v_{n-1}^{(2)}$. First, we start with the node-deleted case.

**Node-deleted policy.** Denote the subgraph $G_1 = \mathrm{CSL}(n, 2) - v_0^{(1)}$ and $G_2 = \mathrm{CSL}(n, k) - v_0^{(2)}$. Due to Lemma 9, it suffices to show that $G_1 \neq_{1-\mathrm{WL}} G_2$.

After deleting the nodes, in $G_1$ we note that $v_1^{(1)}, v_{n-1}^{(1)}, v_2^{(1)}$, and $v_{n-2}^{(1)}$ have degree 3 while all other nodes have degree 4. Also, these degree 3 nodes each have at least one neighbor of degree 3.

In $G_2$, the nodes of degree 3 are $v_1^{(2)}, v_{n-1}^{(2)}, v_k^{(2)}$, and $v_{n-k}^{(2)}$. However, it can be seen that none of these nodes have a neighbor of degree 3.

In fact, it suffices to prove that just $v_1^{(2)}$ has no neighbors of degree 3. To see this, note that $v_1^{(2)}$ has neighbors $v_2^{(2)}, v_{1+k}^{(2)}$, and $v_{n-k+1}^{(2)}$. The constraints $n \geq 7$ and $k \in [3, n/2 - 1]$ show that these neighbors are disjoint from the degree 3 nodes $S = \{v_{n-1}^{(2)}, v_k^{(2)}, v_{n-k}^{(2)}\}$. This is because these constraints imply that the indices of the degree 3 nodes in $S$ satisfy

$$3 < n/2 < n - 1 < n$$
$$3 \leq k \leq n/2 - 1$$
$$4 < n/2 + 1 \leq n - k \leq n - 3.$$

On the other hand, the indices of the neighbors of $v_1^{(2)}$ (i.e. $v_2^{(2)}, v_{k+1}^{(2)}, v_{n-k+1}^{(2)}$) satisfy:

$$0 < 2 < 3$$
$$k < k + 1 \leq n/2$$
$$n/2 + 2 \leq n - k + 1 < n - 2.$$

Which together suffice to show that the indices do not overlap. Indeed, 2 is less than all of the indices of degree 3 nodes. $k + 1$ is less than $n - k$, greater than $k$, and less than $n - 1$. Finally, $n - k + 1$ is less than $n - 1$, greater than $k$, and greater than $n - k$.

Thus, $v_1^{(2)}$ has no degree 3 neighbors, so that 1-WL distinguishes $G_1$ and $G_2$ after two iterations.

**EGO policies.** Let $G_1$ be the depth-1 ego-net of $\mathrm{CSL}(n,2)$ rooted at $v_0^{(1)}$, and let $G_2$ be the depth-1 ego-net of $\mathrm{CSL}(n,k)$ rooted at $v_0^{(2)}$. This proof works for both the EGO and EGO+ cases, as we will simply show the degree distribution of $G_1$ and $G_2$ differ. This is because $G_1$ has 7 edges total — 4 edges from $v_0^{(1)}$ to $v_{n-1}^{(1)}, v_{n-2}^{(1)}, v_1^{(1)}$, and $v_2^{(1)}$, as well as edges $(v_{n-2}^{(1)}, v_{n-1}^{(1)})$, $(v_1^{(1)}, v_2^{(1)})$, and $(v_{n-1}, v_1)$. However, $G_2$ only has 4 edges — those incident to the root $v_0^{(2)}$. The proof that no other edges exist is the same as the above proof that the nodes of degree 3 in the node-deleted subgraphs of $G_2$ have no other neighbors of degree 3. As the degree distributions differ, 1-WL can distinguish $G_1$ and $G_2$.

The derivations above are enough to prove the thesis hold for DS-WL under policies ND, EGO and EGO+. Proposition 7 ensures that these graphs are also distinguished by DSS-WL and allows us to conclude the proof. □

We now exhibit an example of a graph pair separated by our variants with ED policy (and undistinguishable by WL).

**Definition 6** ($C_6$ and $2 \times C_3$)**.** *The $C_6$ graph is a 2-regular graph composed by a chordless cycle of 6 nodes. The $2 \times C_3$ graph is 2-regular and is a graph formed by two disconnected cycles of 3 nodes each (triangles). These graphs are illustrated in Figure 5a.*

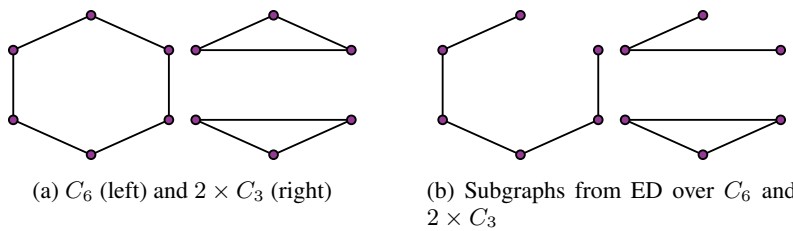

(a) $C_6$ (left) and $2 \times C_3$ (right)  (b) Subgraphs from ED over $C_6$ and $2 \times C_3$

Figure 5: Graphs $C_6$, $2 \times C_3$ (left) and their edge-deleted subgraphs (right).

It is easy to verify that, when run on this pair, the WL test converges at the first iteration: each node is assigned a color representing degree 2. This is a consequence of being 2-regular. We now show that they are indeed distinguished by our variants with ED policy.

**Lemma 10.** *The $C_6$ and $2 \times C_3$ are disambiguated by DS-WL and DSS-WL under ED policy.*

*Proof.* We discuss the DS-WL variant; if this is able to distinguish between the two graphs, then DSS-WL is as well, in light of Proposition 7. We first note that, on each of the two graphs, the ED policy induces bags of only one isomorphism class (shown in Figure 5b). This means that it is sufficient to show these two subgraphs are distinguished by WL to prove DS-WL separates them. We simulate two WL rounds on these subgraphs starting from a constant color initialisation.

*(Iter. 1)* Colors after the first WL refinement uniquely represent node degrees. Each of the two subgraphs will therefore contain two nodes with color 1 and four nodes with color 2. The subgraphs are not yet separated.

*(Iter. 2)* After the second WL refinement, we note that the number of nodes of color $(2, \{\!\{1, 2\}\!\})$ is two in $C_6$, while this color is absent in $2 \times C_3$ because there is no node having both a 1-degree and a 2-degree node in its neighborhood. At this step the subgraphs are separated. □

We are now ready to prove Theorem 1.

*Proof of Theorem 1.* *(DSS-WL)* Policies ND, ED, EGO, EGO+ are vertex set preserving. Thus, in view of Lemma 7, DSS-WL refines WL under these subgraph selection strategies and we know that there exists exemplary graph pairs distinguished by DSS-WL under these policies and not by WL. In particular, Lemma 1 exhibits exemplary pairs for policies ND, EGO, EGO+: these CSL pairs

are indeed indistinguishable by WL (Murphy et al., 2019; Chen et al., 2019). Lemma 10 exhibits another such exemplary pair for policy ED. This is sufficient to prove the thesis for DSS-WL.

*(DS-WL)* Let us consider augmented policies $\widehat{\text{ND}}$, $\widehat{\text{ED}}$, $\widehat{\text{EGO}}$, $\widehat{\text{EGO}+}$. In view of Lemma 8, these policies are such that DS-WL refines WL. Additionally, we note that Lemma 1 and Lemma 10 trivially hold also when considering these policies, thus providing counterexamples of graphs indistinguishable by WL but separated by DS-WL under these policies. This allows us to conclude the proof. □

### D.2.3 NEURAL COUNTERPARTS

We will now focus on characterizing the relation between the proposed WL variants and neural counterparts such as the DSS-GNN architecture introduced in the main text, seeking to prove Theorem 2. In this subsection we will always work with a DSS-GNN architecture with GraphConv encoders (Morris et al., 2019) and entry-wise max adjacency matrix aggregation. Accordingly, the representation for node $v$ in subgraph $i$ is given by:

$$h_{v,i}^{t+1} = \sigma\big(W_1^1 h_{v,i}^t + W_2^1 \sum_{u\sim_i v} h_{u,i}^t + W_1^2 \sum_{j=1}^m h_{v,j}^t + W_2^2 \sum_{w\sim v}\sum_{j=1}^m h_{w,j}^t\big) \tag{7}$$

where $\sigma$ is the ReLU non-linearity, $\sim_i$ indicates the connectivity over subgraph $i$ and $\sim$ indicates the connectivity induced by the aggregated subgraph adjacency matrices. Let us remark that, for all policies of interest in this work, each edge in the original graph is present in at least one subgraph. In these cases, $\sim$ corresponds to the connectivity in the original graph. We conveniently formalize this intuition in the following definition:

**Definition 7** (Edge-covering policies). *A subgraph selection policy $\pi$ is said to be* edge covering *if for any graph $G = (V, E)$, $\forall e \in E$, $\exists S = (V_S, E_S) \in \pi(G)$ such that $e \in E_S$.*

Clearly, policies ND, ED, EGO(+) are all edge-covering. Let us now show that in the case of edge-covering policies, there exists a DSS-GNN architecture with layers in the form of Equation (7) that, when working on graphs of bounded size, can "simulate" the color refinement steps of the DSS-WL algorithm.

**Lemma 11.** *Let $\mathcal{G} = \{\!\!\{G_i = (V_i, E_i)\}\!\!\}_{i=1}^m$ denote the bag generated by an edge-covering selection policy on any $G$ from a family $\mathcal{F}$ of graphs with bounded size, optionally endowed with node labels from a finite support. Let $c_{v,k}^{(T)}$ be the color assigned by DSS-WL to node $v$ in subgraph $k$, after $T < \infty$ iterations. There exists a DSS-GNN model $\mathcal{M}$ with $D = 10T$ layers of the form described in Equation (7) such that $c_{v,k}^{(T)}$ is uniquely represented by a one-hot encoding $h_{v,k}^{(D)}$, the representation for node $v$ in subgraph $k$ computed by such a model.*

*Proof.* We will prove the lemma by describing how to parameterize the DSS-GNN layers in a way to simulate the DSS-WL color refinement step on each subgraph.

*(Preliminaries)* We assume to work with finite graphs whose size is upper-bounded by a positive integer $M \in \mathbb{N}$ and whose degree is upper-bounded by $N \in \mathbb{N}$. From these assumptions it immediately follows that any employed subgraph selection policy generates bags of bounded cardinalities, whose subgraphs are finite graphs with size upper-bounded by $M$ and degree upper-bounded by $N$. Let us denote with $B < \infty$ the maximum possible bag cardinality under the considered selection policy on $\mathcal{F}$. Additionally, if initial node labels are present, it is assumed that they are defined over a finite support. For a DSS-GNN to simulate color refinement steps, it is necessary to encode colors in vector/matrix form. Under the aforementioned boundedness assumptions, at any step, each of the four inputs in the DSS-WL color refinement equation belongs to a finite set of bounded size. The domain of the HASH function is therefore represented by the cross-product of such these sets and is therefore finite and of bounded size. Due to the injectiveness of HASH, its image must therefore be finite and bounded as well. The application of this injective function effectively computes a unique representation of the inputs by assigning them a color.

*(Encoding)* Given these premises, it is clear that, at each time step $t$, including $t = 0$, it is possible to represent colors as one-hot vectors of proper dimension without losing any information. This allows the application of our DSS-GNN model. In the following, we indicate with $t \in \mathbb{N}$ a time

step in the DSS-WL procedure, and with $d_t \in \mathbb{N}$ the corresponding layer of the described DSS-WL model. Note that, in general, $d_t \geq t$ as more than one layer is required to simulate one color refinement step. Let $c_{v,k}^{(t)}$ be the color assigned to node $v$ on subgraph $k$ at time-step $t$ and $h_{v,k}^{(d_t)}$ be its representation computed by our DSS-GNN model, of dimension $a_{d_t}$. At time step $t = 0$, each node in each subgraph is assigned a constant color $c_{v,k}^{(0)} = \bar{c}$; if initial colors are available, they are retained. We consider $h_{v,k}^{(d_0)} = h_{v,k}^{(0)}$ to be a proper one-hot encoding of these initial colors. For a color refinement from time step $t$ to $t+1$ we consider inputs to layer $d_t$ to be one-hot encoded. The refinement is simulated with layers $d_t$ to $d_{t+1}$, where the final output of these is a one-hot encoding uniquely representing the result of the HASH function.

*(Computation)* Starting from the one-hot color representation in input to layer $d_t$, the refinement step is implemented by: (i) computing unique representations for all the four inputs to the HASH function; (ii) simulating the HASH function by mapping them to an output, unique, one-hot encoding. Let us start by describing step (i). An important observation is that the summation is an injective aggregator over multisets when elements are represented by one-hot encodings. In other words, we can uniquely represent a multiset of one-hot encoded elements by simply summing its members. This implies that there exists one DSS-GNN layer which computes a unique representation for each of the first three inputs to the HASH function: $c_{v,k}^{(t)}, N_{v,k}^{(t)}, C_v^{(t)}$. In this layer, each matrix $W$ has dimension $3 \cdot a_{d_t} \times a_{d_t}$ and $b$ has dimension $3 \cdot a_{d_t}$. In particular: $W_1^1 = [I, \mathbf{0}, \mathbf{0}], W_2^1 = [\mathbf{0}, I, \mathbf{0}], W_1^2 = [\mathbf{0}, \mathbf{0}, I], W_2^2 = \mathbf{0}, b = \mathbf{0}$, where $[\cdot, \cdot]$ indicates vertical concatenation.

In output from this layer are vectors in the form $h_{v,k}^{(d_t+1)} = [h_{v,k}^{(d_t)}, h_{v,k}^{(d_t,N)}, h_v^{(d_t,C)}]$, with the last two components representing unique encodings of, respectively, inputs $N_{v,k}^{(t)}, C_v^{(t)}$. In order to also compute a unique representation for the fourth input ($M_v^{(t)}$), we cannot directly apply summation on the just computed $h_v^{(d_t,C)}$, because these are not one-hot encoded anymore, and summation is, generally, not injective. To overcome this, we employ additional layers to first uniquely encode each $h_{v,k}^{(d_t,C)}$ with a one-hot scheme so that the original neighborhoods $\mathcal{N}(v)$ would be uniquely encoded by the summation over them, inherent in the DSS-GNN layer. This one-hot encoding scheme is possible because there is only a finite number of multisets of size $m \leq B$ that can be constructed from a finite color palette of cardinality $a_{d_t}$. Additionally, this encoding can be computed by stacking 4 DSS-GNN layers with ReLU activations as shown by the following:

**Lemma 12.** *Let $\mathcal{H}$ be a finite set. Let elements in $\mathcal{H}$ be uniquely encoded by $h(\cdot)$. There exists a 4-layer DSS-GNN model which computes the map $\mu : h(a) \mapsto e, a \in \mathcal{H}$ with $e$ being a unique one-hot encoding for element $a$.*

*Proof.* We consider an ordering relation $<$ determining an overall sorting $s$ of the elements in $\mathcal{H}$. This can be obtained, for example, by a lexicographic ordering of the elements in the encodings $h(\cdot)$. This sorting associates any element with an integer index, that is $s : \mathcal{H} \to \mathbb{N}, s : a \mapsto i$. We can therefore denote each (encoded) element with $h_i, i = 1, \ldots, |\mathcal{H}|$. We seek to implement the mapping $\mu : h_i \mapsto e_i$, such that: $e_i \in \{0, 1\}^{|\mathcal{H}|}$ and $(e_i)_j = 1$ if $j = i$, $(e_i)_j = 0$ otherwise.

We define the dataset $\mathcal{D} = \{(h_i, e_i)\}_{i=1}^{\mathcal{H}}$. This dataset satisfies Assumption 3.1 in Yun et al. (2019): all $h_i$'s are distinct and labels $e_i$'s are in $[-1, +1]^{d_y}, d_y = |\mathcal{H}|$. Moreover we have that $\forall i, e_i \in 0, 1^{d_y}$. Thus, according to Proposition 3.2 in the same work, there exists a 4-layer ReLU network parameterised by $\theta$ and with $\Omega(\sqrt{|\mathcal{H}|} + |\mathcal{H}|)$ neurons which perfectly fits $\mathcal{D}$. This network computes $\mu$ and is exactly implemented by a 4-layer DSS-GNN model with matrices $W_2^{1(l)}, W_1^{2(l)}, W_2^{2(l)} = \mathbf{0}, l = 1, \ldots, 4$ and $\theta = (W_1^{1(l)}, b^{(l)})_{l=1}^4$. $\square$

We invoke Lemma 12 with $\mathcal{H}$ being the (finite) set of all possible $B$-size multisets over node colorings $h^{(d_t)}$, which are from a finite palette of size $a_{d_t}$. Accordingly, there exists a stacking of 4 DSS-GNN layers such that $h_{v,k}^{(d_t+5)} = [h_{v,k}^{(d_t)}, h_{v,k}^{(d_t,N)}, \bar{h}_v^{(d_t,C)}]$, with $\bar{h}_v^{(d_t,C)}$ being the one hot encoding of $h_v^{(d_t,C)}$, dimension $e_{t,C}$. If $(\tilde{W}_1^{(l)}, \tilde{b}^{(l)})_{l=d+1}^{d+5}$ are the parameters from Lemma 12 one-

hot encoding $h_v^{(d_t, C)}$, then these layers are obtained simply by $W_1^{1(l)} = [[I|\mathbf{0}], [\mathbf{0}|\tilde{W}_1^{1(l)}]], b^l = [\mathbf{0}, \tilde{b}^{(l)}], W_2^{1(l)}, W_1^{2(l)}, W_2^{2(l)} = \mathbf{0}$. Here $[\cdot|\cdot]$ indicates horizontal concatenation and $I$ is always of size $2 \cdot a_{d_t} \times 2 \cdot a_{d_t}$. Overall, each $h_{v,k}^{(d_t+5)}$ has dimension $a_{d_t+5} = (2 \cdot a_{d_t}) + e_{t,C}$. Importantly, we notice that component $\bar{h}_v^{(d_t, C)}$ is computed for node $v$ but independently of $k$, and is thus shared across all subgraphs. Layer $d_t + 6$ now computes the encoding of $M_v^{(t)}$. Each weight matrix has dimension $(a_{d_t+5} + e_{t,C}) \times a_{d_t+5}$ with: $W_1^1 = [I, \mathbf{0}]$ ($I$ size $a_{d_t+5}$), $W_2^1, W_1^2 = \mathbf{0}, W_2^2 = [\mathbf{0}, [\mathbf{0}_\perp | \frac{1}{m} I]]$ ( $\mathbf{0}_\perp$ size $e_{t,C} \times (2 \cdot a_{d_t})$, $I$ size $e_{t,C}$), $b = \mathbf{0} \in \mathbb{R}^{a_{d_t+5}}$.

Now, $h_{v,k}^{(d_t+6)} = [h_{v,k}^{(d_t)}, h_{v,k}^{(d_t,N)}, \bar{h}_v^{(d_t,C)}, \bar{h}_v^{(d_t,M)}]$, of dimension $a_{d_t+6} = a_{d_t+5} + e_{t,C}$, constructed as the concatenation of unique representations of the inputs to the HASH function. It is only left to "simulate" the application of this last. We do this by computing a one-hot encoding of vector $h_{v,k}^{(d_t+6)}$. As already discussed, this encoding is possible due the fact that HASH is an injection with domain of finite and bounded cardinality. In other words, vectors $h_{v,k}^{(d_t+6)}$ can only assume a finite, bounded number of distinct values. Again, this encoding is computable by 4 DSS-GNN layers, as per Lemma 12 invoked considering the (finite) set of all possible colorings attainable by encodings $h^{(d_t+6)}$.

Finally, we have that $h_{v,k}^{(d_{t+1})} = h_{v,k}^{(d_t+10)}$. This construction can be arbitrarily repeated to exactly simulate $T$ iterations of the DSS-WL color refinement step with $D = 10T$ layers, which concludes the proof. $\square$

The following Lemma shows that there exists a DS-GNN architecture with layers of the form in Equation (7) that, when working on graphs of bounded size, can "simulate" the color refinement steps of the DS-WL algorithm.

**Lemma 13.** *Let $\mathcal{G} = \{\!\{G_i\}\!\}_{i=1}^m$ denote the bag generated by a subgraph selection policy applied on any $G$ from a family $\mathcal{F}$ of graphs with bounded size. Let $c_{v,k}^{(T)}$ be the color assigned by DS-WL to node $v$ in subgraph $k$, after $T$ iterations. There exists a DS-GNN model $\mathcal{M}$ with $D = 5T$ layers of the form described in Equation (7) such that $c_{v,k}^{(T)}$ is uniquely represented by a one-hot encoding $h_{v,k}^{(D)}$, the representation for node $v$ in subgraph $k$ computed by such a model.*

*Proof.* The proof closely follows the one for Lemma 11; we work in the same regime (finite graphs of bounded size and degree, initial labels from a finite support) and assume the network encodes node colors with a proper one-hot encoding at each refinement step being simulated. The difference is that, in this case, the described DS-GNN model always has parameters $W_1^2, W_2^2 = \mathbf{0}$ and does not need to encode the HASH inputs $C_v^{(t)}, M_v^{(t)}$ (not present in its update equations).

Let $c_{v,k}^{(t)}$ be the color assigned to node $v$ on subgraph $k$ at time-step $t$ and $h_{v,k}^{(d_t)}$ be its representation computed by our DS-GNN model, of dimension $a_{d_t}$ and properly one-hot encoded. We seek to describe a stacking of DS-GNN layers producing $h_{v,k}^{(d_{t+1})}$, a unique, one-hot encoding of color $c_{v,k}^{(t+1)}$.

Once more, since summation is an injective aggregator over multisets when elements are represented by one-hot encodings, there exists one DS-GNN layer which computes a unique representation for the two inputs to the HASH function: $c_{v,k}^{(t)}, N_{v,k}^{(t)}$. In this layer, each matrix $W$ has dimension $2 \cdot a_{d_t} \times a_{d_t}$ and $b$ has dimension $2 \cdot a_{d_t}$. In particular: $W_1^1 = [I, \mathbf{0}], W_2^1 = [\mathbf{0}, I], b = \mathbf{0}$ where $[\cdot, \cdot]$ indicates vertical concatenation.

In output from this layer are vectors in the form $h_{v,k}^{(d_t+1)} = [h_{v,k}^{(d_t)}, h_{v,k}^{(d_t,N)}]$, with the last component representing a unique encoding of $N_{v,k}^{(t)}$. In order to obtain $h_{v,k}^{(d_{t+1})}$ it is only left to one-hot encode $h_{v,k}^{(d_t+1)}$. We perform this operation with a stacking of 4 DS-GNN layers leveraging only on matrix $W_1^1$ (in module $L_1$) and bias $b$. This stacking is described by Lemma 12, invoked considering the (finite) set of all possible colorings attainable by encodings $h^{(d_t+1)}$. $\square$

Let us now move to prove Theorem 2 from the main text.

*Proof of Theorem 2. (DS(S)-GNN $\sqsubseteq$ DS(S)-WL)* We will directly describe a DSS-GNN architecture satisfying the requirements under edge-covering policies. According to Equation (2), a DSS-GNN architecture computes a representation for a graph in input as: $h_G = E_{\text{sets}} \circ R_{\text{subgraphs}} \circ E_{\text{subgraphs}}(G)$. Let $T$ be the time step at which DSS-WL (resp. DS-WL) distinguishes two input graphs $G^1, G^2$. At this round, the overall coloring is such that the multisets of subgraph colors for the two graphs are distinct:

$$\{\!\{c^{(T)}_{G^1_k}\}\!\}^m_{k=1} \neq \{\!\{c^{(T)}_{G^2_h}\}\!\}^m_{h=1}$$

where $c^{(T)}_{G^i_j}$ generally indicates the color computed for subgraph $j$ of graph $G^i$: $\text{HASH}(\{\!\{c^{(T)}_{v,j}|v \in V^i\}\!\})$. Now, let $E_{\text{subgraphs}}$ be the module obtained by the stacking described in Lemma 11 (resp. Lemma 13). We know that this module uniquely encodes each subgraph node color $c^{(T)}_{v,j}$ with one-hot representations $h^{(d_T)}_{v,j}$. Therefore, it is only left to describe modules $R_{\text{subgraphs}}, E_{\text{sets}}$ in a way to injectively aggregate such representations into an overall graph embedding.

Given that subgraph node colors are (uniquely) represented by one-hot encodings, the required $R_{\text{subgraphs}}$ can be simply obtained by summation over node representations on each subgraph. This is because summation is an injective aggregator when multiset elements are described in a one-hot fashion. Let us generally refer to the computed subgraph representations as $h_j$.

Let us finally describe module $E_{\text{sets}}$. The graphs of interest have a number of nodes upper-bounded by a positive integer and so $h_j$'s uniquely represent bounded-sized multisets of colors from a finite palette. The set of all possible such multisets is finite, so it is possible to compute a one-hot encoding for these subgraph representations with a 4-layer ReLU MLP applied in a siamese fashion. Lemma 12 ensures the existence of such a network as the described stacking of DSS-GNN layers effectively boils down to a Siamese MLP. This would constitute the first 4 layers of $E_{\text{sets}}$. To conclude, one last layer applying summation to these one-hot encodings would suffice to output an embedding uniquely describing the multiset of subgraph colors.

*(DS-WL $\sqsubseteq$ DS-GNN)* As for DS-GNN and DS-WL, we show in the following that, if a DS-GNN distinguishes two graphs, then DS-WL also distinguishes these two graphs. Suppose DS-GNN distinguishes between $G^1$ and $G^2$. If the bags $S_{G^1}$ and $S_{G^2}$ are of different sizes, then DS-WL separates the two graphs as well because it hashes multisets of different sizes to different outputs. Assume they have the same size. Letting 1-WL: $\mathcal{G} \to \Sigma$ denote the 1-WL graph coloring function and $\text{MPNN} : \mathcal{G} \to \Sigma$ denote the MPNN base graph encoder employed by DS-GNN, we have:

$$\{\!\{\text{MPNN}(G^1_1), \ldots, \text{MPNN}(G^1_m)\}\!\} \neq \{\!\{\text{MPNN}(G^2_1), \ldots, \text{MPNN}(G^2_m)\}\!\} \tag{8}$$

By Morris et al. (2019); Xu et al. (2019), we know that 1-WL $\sqsubseteq$ MPNN, i.e. that 1-WL$(H^1) = $ 1-WL$(H^2)$ implies that $\text{MPNN}(H^1) = \text{MPNN}(H^2)$ for any graphs $H^1$ and $H^2$. Thus, Lemma 2 tells us that

$$\{\!\{\text{1-WL}(G^1_1), \ldots, \text{1-WL}(G^1_m)\}\!\} \neq \{\!\{\text{1-WL}(G^2_1), \ldots, \text{1-WL}(G^2_m)\}\!\}. \tag{9}$$

Hence, the final readout of DS-WL maps these multisets to different values, so that DS-WL distinguishes $G^1$ and $G^2$.

This is enough to prove the second part of the thesis, terminating the overall proof. $\qquad\square$

# E    PROOFS FOR SECTION 3.3: DESIGN CHOICE EXPRESSIVENESS

This appendix section contains proofs for Section 3.3. First, we note that we use DS-GNN and DS-WL interchangeably in proofs, as they are equivalent in power for distinguishing graphs (as shown in Theorem 2).

## E.1    PROOFS OF DSS VERSUS DS

*Proof of Prop 1.* That DSS-GNN is at least as powerful as DS-GNN follows directly from the definition. For any DS-GNN, there is a DSS-GNN with $L^2 = 0$ in each layer that implements the DS-GNN. To show that DSS-GNN is strictly more powerful, we provide two examples.

**A trivial example.** Consider the subgraph selection policy $SE : G = (V, E) \mapsto \{(V, \{e\}) \mid e \in E\}$. Each subgraph is a single edge from the original graph. Now, let $G_1$ be the path on 4

nodes, and let $G_2$ be the 3-star. In other words, $G_1$ has a vertex set of $\{1, 2, 3, 4\}$ and edges $\{(1, 2), (2, 3), (3, 4)\}$, while $G_2$ has a vertex set of $\{1, 2, 3, 4\}$ and edges $\{(1, 2), (1, 3), (1, 4)\}$.

Now, we show that DS-GNN cannot distinguish $G_1$ and $G_2$. Note that the bags are of the same size: $|SE(G_i)| = 3$. Moreover, any other subgraph is a single edge, and these single edges are isomorphic and hence 1-WL equivalent. Thus, the multisets of subgraph colors are equal

$$\{\!\{1\text{-WL}(H) : H \in SE(G_1)\}\!\} = \{\!\{1\text{-WL}(H) : H \in SE(G_2)\}\!\}, \tag{10}$$

so that DS-WL cannot distinguish these graphs, and hence DS-GNN cannot distinguish these graphs by Theorem 2.

To see that DSS-GNN can distinguish these graphs, we let $L^1(A, X) = 0$ and $L^2(A, X) = A\left(\frac{1}{3}X\right)$. These operations are simple and can be implemented by MPNNs with sum pooling; for instance, GIN (Xu et al., 2019) and GraphConv (Morris et al., 2019) can both implement $L^1$ and $L^2$. Letting the initial node features in each subgraph $i$ be $X_i^{(0)} = \vec{1} = [1 \ldots 1]^\top$, we compute the node features after one layer as

$$X_i^{(1)} = L^1(A_i, \vec{1}) + L^2\left(\sum_j A_j, \sum_j \vec{1}\right) \tag{11}$$

$$= 0 + L^2\left(A, 3\vec{1}\right) \tag{12}$$

$$= A\vec{1}. \tag{13}$$

Thus, the $k$th component of $X_i^{(1)}$ is the degree of node $k$. Since $G_1$ has no nodes of degree 3 while $G_2$ has one node of degree 3, these node features differ between $G_1$ and $G_2$. Thus, DSS-GNN distinguishes these two graphs. This is because the multiset of node features in $G_1$ differs from the multiset of node features in $G_2$ after this iteration, so a final set pooling across nodes can map these two graphs to different outputs (for instance, a max pooling outputs 2 for $G_1$ and 3 for $G_2$).

This example is trivial because it is easy to make a change to the policy that would allow DS-GNN to distinguish these two graphs. Since the two original graphs have different degree distributions, they are directly distinguished by 1-WL. Thus, adding the original graph to the bags of subgraphs would allow DS-GNN to distinguish these graphs.

**A strengthened result.** For an extension of the above example, we consider the policy $\widehat{SE}$ that is given by adding the original graph to the $SE$ policy, so $\widehat{SE}(G) = SE(G) \cup \{G\}$. For this policy, we can again show that DS-GNN cannot distinguish a pair of graphs that DSS-GNN can, though we need to allow a slightly different type of aggregation for DSS-GNN. In particular, when computing the features for subgraph $i$, we use an $L^2$ that is mentioned in Section 2.1: $L^2\left(\sum_j A_j, \sum_{j \neq i} X_j\right)$.

Now, consider the pair of non-isomorphic graphs: $G_1$ is two triangles $2 \times C_3$ and $G_2$ is the 6-cycle $C_6$ (these graphs are depicted in Figure 5). Denote the subgraph corresponding to the single edge $(i, j)$ as $G^{(i,j)}$, and the corresponding adjacency as $A^{(i,j)} \in \{0, 1\}^{6 \times 6}$.

We show that DS-GNN cannot distinguish $G_1$ and $G_2$. First, note that the bag is of size $|\widehat{SE}(G_i)| = 7$ for each graph — they each have 6 edges and the original graph adds one more element to the bag. Note that $G_1$ and $G_2$ are regular of the same number of nodes and degree, so they are 1-WL equivalent. As any other subgraph is a single edge, we once again know that these single edges are isomorphic and thus 1-WL equivalent. Hence, the multisets of subgraph colors are equal

$$\{\!\{1\text{-WL}(H) : H \in \widehat{SE}(G_1)\}\!\} = \{\!\{1\text{-WL}(H) : H \in \widehat{SE}(G_2)\}\!\}, \tag{14}$$

so that similarly to the previous example, DS-GNN cannot distinguish these graphs.

Now, we show that DSS-GNN can distinguish $G_1$ and $G_2$. We give a choice of parameters for DSS-GNN that distinguishes these graphs. Let the input node features be all one, so $X^{(0)} = \vec{1} = (1, \ldots, 1) \in \mathbb{R}^{n \times 1}$. Let the first layer graph encoders be given by:

$$L^1(A, X) = AX, \qquad L^2(A, X) = 0 \tag{15}$$

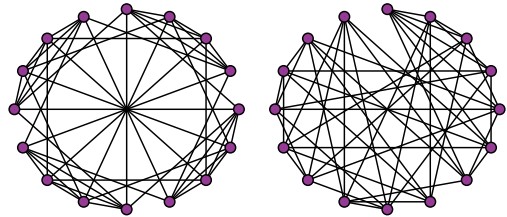

Figure 6: The Rook's graph (left) and Shrikhande graph (right) (Arvind et al., 2020) are non-isomorphic strongly regular graphs of the same parameters. They thus cannot be distinguished by 3-WL.

Once again, these two graph encoders can be implemented by both GIN and GraphConv. Then the $k$th element of the representation of the nodes of $A^{(i,j)}$ after one layer is:

$$X^{(1)}_{(i,j),k} = (L(\mathcal{A}, \mathcal{X}))^{(i,j)}_k = L^1(A^{(i,j)}, \vec{1})_k \tag{16}$$

$$= (A^{(i,j)}\vec{1})_k \tag{17}$$

$$= \mathbb{1}_{k \in \{i,j\}}. \tag{18}$$

In other words, the feature on node $k$ for $A^{(i,j)}$ is one if $k$ is $i$ or $j$, and zero otherwise. Also, a similar computation shows that subgraph $G$ corresponding to the original graph has node feature $2\vec{1}$ after this layer.

Now, let the second layer graph encoders be:

$$L^1_{(2)}(A, X) = 0, \qquad L^2_{(2)}(A, X) = \left(\frac{1}{2}A\right)\left(-X + 4\vec{1}\right), \tag{19}$$

where $L^2$ takes the standard sum pooled $\sum_j A_j$ as well as the modified pooled $\sum_{j \neq i} X_j$. These can be implemented by GraphConv with bias terms. Then the $k$th element of the representation of the nodes of $A^{(i,j)}$ after the second layer is:

$$X^{(2)}_{(i,j),k} = L^2_{(2)}\left(A + \sum_{(i,j)\in E} A^{(i,j)}, \; X^{(1)}_G + \sum_{(v,w)\in E\setminus\{(i,j)\}} X^{(1)}_{(v,w)}\right)_k \tag{20}$$

$$= L^2_{(2)}\left(2A, \; 2\vec{1} + \sum_{(v,w)\in E\setminus\{(i,j)\}} X^{(1)}_{(v,w)}\right)_k \tag{21}$$

$$= L^2_{(2)}\left(2A, \; Z_{(i,j)}\right)_k \tag{22}$$

$$= \left(A(-Z_{(i,j)} + 4\vec{1})\right)_k \tag{23}$$

where $Z_{(i,j),l}$ is 4 if $l \notin \{i,j\}$ and is 3 if $l \in \{i,j\}$. Thus, $-Z_{(i,j),l} + 4$ is 0 if $l \notin \{i,j\}$ and 1 if $l \in \{i,j\}$. We show that $X^{(2)}_{(i,j)}$ differs between $G_1$ and $G_2$.

For $G_1$, let $k$ be the other node in the triangle that node $i$ and $j$ are in. Then $X^{(2)}_{(i,j),k} = 2$. On the other hand, for $G_2$, there is no value of $k$ such that $X^{(2)}_{(i,j),k}$ is 2. This is because there is no node $k$ that is adjacent to both $i$ and $j$ (there are no triangles in $G_2$). Thus, we have shown that DSS-GNN distinguishes these two graphs after the second layer, and we are done. $\square$

### E.2 PROOFS OF INCREASING ENCODER EXPRESSIVENESS

First, we introduce certain graphs that provide useful examples for our proofs, and that have been used to study expressiveness of graph algorithms more generally.

**Rook's graph and Shrikhande graph.** We also introduce a particular pair of commonly-used example graphs that will be useful for us. Figure 6 illustrates the 4-by-4 Rook's graph and the Shrikhande graph. These are non-isomorphic graphs that are strongly regular of the same parameters, so they cannot be distinguished by 3-WL / 2-FWL (Arvind et al., 2020; Balcilar et al., 2021).

We number the nodes for ease of reference in the below proofs. For the either graph, let the vertex set be $V = \{(x, y) : x \in \{0, 1, 2, 3\}, y \in \{0, 1, 2, 3\}\}$, so each of the 16 nodes are viewed as a point on a grid in the plane. For each node $(x, y) \in V$, the Rook's graph has edges $((x, y), (x', y))$ for $x' \neq x$ and $((x, y), (x, y'))$ for $y' \neq y$. The Shrikhande graph has edges $((x, y), (x, y \pm 1 \bmod 4))$, $((x, y), (x \pm 1 \bmod 4, y))$, and $((x, y), (x \pm 1 \bmod 4, y \pm 1 \bmod 4))$. We number the nodes on the grid from left to right then top to bottom, so node 1 is (0,3), node 2 is (1,3), node 5 is (0,2), node 6 is (1,2), and so on.

*Proof of Prop 2.* Note that the generalization of DS-GNN to 3-WL encoders is straightforward, as we simply use the graph representation vectors from the readout of the 3-WL encoder as the subgraph representations in the DeepSets module. In other words, letting 3-WL-GNN($G$) be the final coloring of $G$ that is obtained from a GNN that is equivalent to 3-WL, we apply a DeepSets module to $\{\!\{3\text{-WL-GNN}(G_i) : G_i \in S_G\}\!\}$ for DS-GNN with a 3-WL encoder. This DS-GNN with 3-WL is at least as strong as DS-GNN with 1-WL since 3-WL is at least as strong as 1-WL. Also, DS-GNN with 3-WL is at least as strong as 3-WL since the original graph is included in any augmented policy $\widehat{\pi}$.

To show that DS-GNN with 3-WL is strictly stronger than the other two models, we show that it can distinguish the Rook's graph and the Shrikhande graph, which the other two models cannot. Recall that 3-WL cannot distinguish the two because they are both strongly regular graphs of the same parameters (Arvind et al., 2020).

Now, note that all depth-1 ego-nets of the Rook's graph consist of two disjoint triangles with a root node that is connected to all nodes. All depth-1 ego-nets of the Shrikhande graph consist of a 6-cycle with a root node that is connected to all nodes. It is easy to see that 1-WL cannot distinguish these ego-nets, as 1-WL cannot distinguish two disjoint triangles from a 6-cycle. Thus, DS-GNN with the depth-1 $\widehat{\text{EGO}}$ or $\widehat{\text{EGO}}$+ policy cannot distinguish the two graphs.

On the other hand, 3-WL can distinguish these ego-nets as 3-WL can distinguish two disjoint triangles from a 6-cycle; for instance, this can be seen due to 3-WL being able to count triangles at initialization (Chen et al., 2020). □

### E.3 Proofs of Policy Choice and Expressiveness

Before restricting to strongly regular graphs, we give an example that shows how the power of the EGO policy can depend in interesting ways with the depth — increasing ego-net depth may either increase or decrease the expressivity of DS-GNN.

**Proposition 8.** *Our DS-GNN model with depth-1 or depth-3 EGO policies do not distinguish CSL(12,3) and CSL(12,5), while the depth-2 EGO policy does distinguish them.*

*Proof of Prop 8.* The depth-1, depth-2, and depth-3 ego-nets of $CSL(12, 3)$ and $CSL(12, 5)$ are all isomorphic, so DS-GNN with a 1-WL encoder can distinguish the two graphs if and only if 1-WL can distinguish the ego-nets. The ego-nets are plotted in Figure 7.

The depth-1 ego-nets of both $CSL(12, 3)$ and $CSL(12, 5)$ are $K_{1,4}$ — the star with one root node of degree 4 and four nodes of degree 1 connected to the root. As these are isomorphic, 1-WL does not distinguish any subgraphs, so our DS-GNN cannot distinguish $CSL(12, 3)$ and $CSL(12, 5)$ with depth-1 EGO policy.

The depth-2 ego-nets are displayed in Figure 7. It can be seen that the depth-2 ego-nets have different degree distributions: for instance, that of $CSL(12, 3)$ has degree 3 nodes while the ego-nets of $CSL(12, 5)$ do not have any degree 3 nodes. Thus, 1-WL distinguishes these ego-nets, and hence DS-GNN does as well with the depth-2 EGO policy.

As both $CSL(12, 3)$ and $CSL(12, 5)$ have diameter 3, the depth-3 ego-nets are in both cases just the original graph. As both graphs are regular of the same number of nodes and same degree, 1-WL

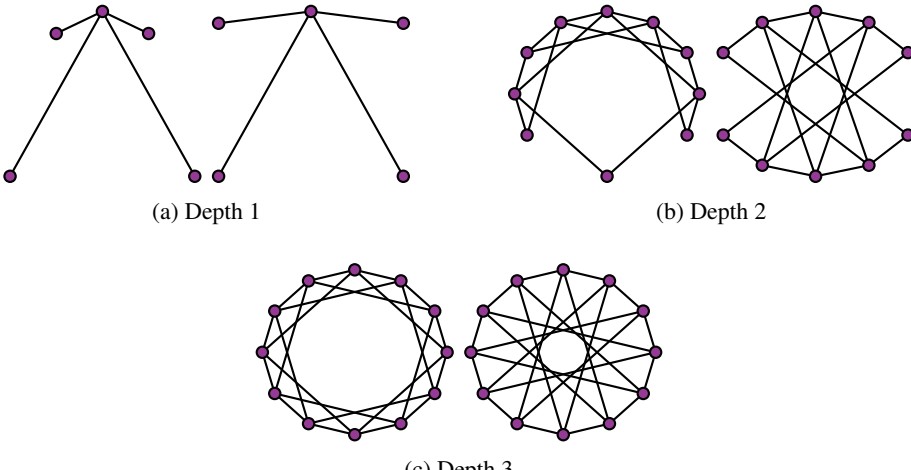

(a) Depth 1             (b) Depth 2

(c) Depth 3

Figure 7: Ego-nets of $\mathrm{CSL}(12, 3)$ (left) and $\mathrm{CSL}(12, 5)$ (right) for depths 1, 2, and 3. These are used in Proposition 8.

does not distinguish the subgraphs, so DS-GNN cannot distinguish $\mathrm{CSL}(12, 3)$ and $\mathrm{CSL}(12, 5)$ with depth-3 EGO policy. □

### E.4 STRONGLY REGULAR GRAPH PROOFS

In this Section, we prove Proposition 3 on the power of different policies for distinguishing strongly regular graphs.

#### E.4.1 ND AND EGO+ ON STRONGLY REGULAR GRAPHS

**Lemma 14.** *DS-GNN with the ND policy can distinguish any strongly regular graphs of different parameters, but cannot distinguish any strongly regular graphs of the same parameters.*

*Proof.* We show that the 1-WL algorithm converges to the same coloring of *any* node-deleted subgraph of a strongly regular graph in at most 3 iterations, and that the coloring only depends on the strongly regular parameters. Let $G$ be a strongly regular graph of parameters $(n, k, \lambda, \mu)$, meaning that $G$ has $n$ nodes, is regular of degree $k$, any two adjacent vertices have $\lambda$ common neighbors, and any two non-adjacent vertices have $\mu$ common neighbors. We step through the 1-WL iterations on the node-deleted subgraph $G - u$:

- Iteration 1: Initialize each node to have color a.

- Iteration 2: Degree $k - 1$ nodes are colored b and degree $k$ nodes are colored c.

- Iteration 3: All degree $k - 1$ nodes are colored d, as they each had color b and have $\lambda$ b neighbors and $(k - 1 - \lambda)$ c neighbors. To see this, note that these nodes were adjacent to the deleted node $u$ in the original graph $G$, so they had exactly $\lambda$ common neighbors with $u$. Since the neighbors of $u$ are now degree $k - 1$ in $G - u$, they have color b at iteration 2. Thus, there are exactly $\lambda$ b neighbors, and the remaining color is c, so there are also exactly $(k - 1 - \lambda)$ c neighbors.

  Also, all degree $k$ nodes are colored e, as they had color c and have $\mu$ b neighbors and $(k - \mu)$ c neighbors. This is because these nodes are not adjacent to $u$ in the original $G$, so they have exactly $\mu$ common neighbors with $u$ in $G$. Thus, they have exactly $\mu$ b neighbors and $(k - \mu)$ c neighbors.

As the colors of iteration 2 and 3 are isomorphic, 1-WL has converged. Since this coloring is the same for any node-deleted subgraphs of any strongly regular graph of the same parameters, DS-WL and hence DS-GNN cannot distinguish any strongly regular graphs of the same parameters.

Likewise, this coloring differs for any strongly regular graphs $G_1$ and $G_2$ of different parameters. The coloring differs at iteration 1 if $n$ differs, at iteration 2 if $k$ differs, and at iteration 3 if $\lambda$ or $\mu$ differ. Thus, DS-WL and hence DS-GNN is able to distinguish any two such graphs. $\square$

**Lemma 15.** *DS-GNN with the depth-$n$ $\widehat{\text{EGO}}+$ policy can distinguish any strongly regular graphs of different parameters, but cannot distinguish any strongly regular graphs of the same parameters.*

*Proof.* Consider a depth-$n$ ego-net rooted at $u$. This proof is similar to that of the ND case in Lemma 14. We compute the 1-WL iterations:

- Iteration 1: $u$ has color a1 and all other nodes have color b

- Iteration 2: $u$ has color a2 (as it is the only node with previous color a1).

  The $k$ neighbors of $u$ have color c (they each have previous color b, have the one neighbor $u$ of color a1, and have $k - 1$ neighbors of the remaining color b).

  The $n - k - 1$ non-neighbors of $u$ have color d (they each have previous color b, have no neighbors of color a1, and thus have $k$ neighbors of color b).

- Iteration 3: $u$ has color a3 (it is the only node with previous color a2)

  The $k$ neighbors of $u$ have color e. They each had previous color c, have one neighbor of color a2, have $\lambda$ neighbors of color c because they are adjacent to $u$ and all neighbors of u had color c, and have $k - \lambda - 1$ neighbors of the remaining color d.

  The $n - k - 1$ non-neighbors of $u$ have color f (they had previous color d, have no neighbors of color a2, have $\mu$ neighbors of color c because they are non-adjacent to $u$ and all neighbors of u had color c, and have $k - \mu$ neighbors of the remaining color d).

This coloring is stable, so the algorithm terminates. Note that this coloring only depends on the strongly regular parameters. Once again, it differs for any strongly regular graphs of different parameters, and is the same for any strongly regular graphs of the same parameters. Also, note that besides the unique color of the root node $u$, this coloring is the same as that of the 1-WL coloring of the node-deleted graph $G - u$. $\square$

### E.4.2 ED ON STRONGLY REGULAR GRAPHS

**Lemma 16.** *3 iterations of 1-WL on any edge-deleted subgraph of a strongly regular graph gives a coloring that only depends on the strongly regular parameters. This coloring distinguishes any strongly regular graphs of different parameters.*

*Proof.* Say we have deleted edge $(u_1, u_2)$. The 1-WL iterations are as follows:

- In iteration 1, each node has initial color a.

- In iteration 2, we have two nodes of degree $k - 1$ that are colored b, i.e. $u_1$ and $u_2$. There are $n - 2$ nodes of degree $k$ that are colored c.

- In iteration 3, we have two nodes of degree $k - 1$ that are colored d (because they are both only adjacent to degree $k$ nodes previously colored c). For the degree $k$ nodes, there are three possible colors, which we call e, f, g: the color e is for nodes adjacent to 2 b nodes and $k - 2$ c nodes, f is for nodes adjacent to 1 b node and $k - 1$ c nodes, and g is for nodes adjacent to $k$ c nodes. We show that the number of colors e, f, and g only depend on the strongly regular graph parameters.

First, there are $\lambda$ nodes of color e, as the nodes adjacent to 2 b nodes are common neighbors of $u_1$ and $u_2$, which are adjacent in the original graph $G$. By definition there are $\lambda$ common neighbors of $u_1$ and $u_2$.

Next, there are $2(k-1-\lambda)$ nodes of color f. This is because each node of color f is only adjacent to one of $u_1$ or $u_2$. Each of $u_1$ and $u_2$ has $(k-1-\lambda)$ neighbors that are not in common with the other, so there are in total $2(k-1-\lambda)$ nodes of color f.

The only other choice is color g, so we can compute the number by subtracting all the other color counts from the total number of nodes $n$. There are thus $n - 2 - \lambda - 2(k-1-\lambda) = n + \lambda - 2k$ nodes of color g.

It is clear that if $G_1$ and $G_2$ are strongly regular graphs that differ in any of the $n, k,$ or $\lambda$ parameters, then three iterations of 1-WL distinguishes due to the above coloring differing. Now, if $G_1$ and $G_2$ differ in the $\mu$ parameter, the same argument holds, because the $\mu$ parameter for a strongly regular graph is related to the other parameters by

$$(n - k - 1)\mu = k(k - \lambda - 1) \tag{24}$$

(See Theorem 1.1 in Cameron (2004)). Thus, $G_1$ and $G_2$ must differ in at least one of $n, k,$ or $\lambda$, and hence they are distinguished by 3 iterations of 1-WL.

$\square$

**Lemma 17.** *The $4 \times 4$ Rook's graph and the Shrikhande graph are distinguished by DS-GNN and DSS-GNN with the ED policy.*

*Proof.* We prove that DS-GNN can distinguish these two graphs by directly computing 1-WL colorings of edge-deleted subgraphs. It can be checked computationally that all edge-deleted subgraphs of the Rook's graph are isomorphic, and likewise all edge-deleted subgraphs of the Shrikhande graph are isomorphic. Thus, it suffices to show that a 1-WL coloring of one edge-deleted subgraph of the Rook's graph differs from a 1-WL coloring of one edge-deleted subgraph of the Shrikhande graph.

We use the numbering of nodes as we defined in the beginning of Appendix E.2. Each node starts with an initial color that we denote 'a'. See Tables 12 and 13 for the coloring, which differs for the two graphs at iteration 4 (as an aside, note that the first 3 iterations were already computed in the proof of Lemma 16). Thus, these two graphs are distinguished by DS-GNN, and hence DSS-GNN, with the ED policy.

| Node | Iter 1 | 2 | 3 | 4 |
|------|--------|---|---|---|
| 1 | a | b | d | h |
| 2 | a | b | d | h |
| 3 | a | c | e | i |
| 4 | a | c | e | i |
| 5 | a | c | f | j |
| 6 | a | c | f | j |
| 7 | a | c | g | k |
| 8 | a | c | g | k |
| 9 | a | c | f | j |
| 10 | a | c | f | j |
| 11 | a | c | g | k |
| 12 | a | c | g | k |
| 13 | a | c | f | j |
| 14 | a | c | f | j |
| 15 | a | c | g | k |
| 16 | a | c | g | k |

| HASH($\mathcal{M}(v)$) | $\mathcal{M}(v)$ |
|------------------------|------------------|
| a | initialize |
| b | (a, aaaaa) |
| c | (a, aaaaaa) |
| d | (b, ccccc) |
| e | (c, bbcccc) |
| f | (c, bccccc) |
| g | (c, cccccc) |
| h | (d, eefff) |
| i | (e, ddeggg) |
| j | (f, dfffgg) |
| k | (g, effggg) |

Table 12: 1-WL on an edge-deleted subgraph of the Rook's graph, where we delete edge (1,2). The table on the right shows $\mathcal{M}(v)$, which is a tuple containing the current color of node $v$ along with the multiset of neighbor colors. In each iteration of 1-WL, $\mathcal{M}(v)$ is updated to HASH($\mathcal{M}(v)$).

$\square$

| Node | Iter 1 | 2 | 3 | 4 |
|------|--------|---|---|---|
| 1 | a | b | d | h |
| 2 | a | b | d | h |
| 3 | a | c | f | j |
| 4 | a | c | f | j |
| 5 | a | c | f | l |
| 6 | a | c | e | m |
| 7 | a | c | f | l |
| 8 | a | c | g | n |
| 9 | a | c | g | k |
| 10 | a | c | g | k |
| 11 | a | c | g | k |
| 12 | a | c | g | k |
| 13 | a | c | e | m |
| 14 | a | c | f | l |
| 15 | a | c | g | n |
| 16 | a | c | f | l |

| HASH($\mathcal{M}(v)$) | $\mathcal{M}(v)$ |
|------------------------|------------------|
| a | initialize |
| b | (a, aaaaa) |
| c | (a, aaaaaa) |
| d | (b, ccccc) |
| e | (c, bbcccc) |
| f | (c, bccccc) |
| g | (c, cccccc) |
| h | (d, eefff) |
| j | (f, dfffgg) |
| k | (g, effggg) |
| l | (f, defggg) |
| m | (e, ddffgg) |
| n | (g, ffffgg) |

Table 13: 1-WL on an edge-deleted subgraph of the Shrikhande graph, where we delete edge (1,2).

### E.5 PROOF OF PROPOSITION 3

With the above lemmas for different policies we may formally prove Proposition 3.

*Proof of Prop 3.*

(1) Lemmas 14 and 15 show that ND and depth-$n$ $\widehat{\text{EGO}}+$ can distinguish strongly regular graphs of different parameters. Lemma 16 shows the same for ED.

(2) is proven in Lemmas 14 and 15.

(3) is directly proven by Lemma 17, as DS-GNN with ED distinguishes the Rook's graph and the Shrikhande graph, which are strongly regular of the same parameters.

Finally, note that 3-WL cannot distinguish strongly regular graphs of the same parameters (Arvind et al., 2020), so ED is not less powerful than 3-WL in general, and is strictly more powerful than 3-WL on the family of strongly regular graphs. While we have only provided one pair of strongly regular graphs of the same parameters that ED can distinguish, the general case for all strongly regular graphs is hard; while a lot of work has gone into distinguishing non-isomorphic strongly regular graphs, there is no known polynomial-time algorithm to do so (Babai et al., 2013). $\qquad\square$

## F COMPLEXITY

### F.1 FORWARD PASS

Table 14: Complexity of graph networks that are more expressive than 1-WL. $\Delta_{\max}$ denotes the maximum degree over all nodes.

| | PPGN | 3-IGN | 3-GNN | GSN | CWN | Ours |
|---|------|-------|-------|-----|-----|------|
| Time | $\mathcal{O}(n^3)$ | $\mathcal{O}(n^3)$ | $\mathcal{O}(n^4)$ | $\mathcal{O}(n\Delta_{\max})$ | $\mathcal{O}\left(\sum_{p=1}^{2} n_p(B_p + 2\binom{B_p}{2})\right)$ | $\mathcal{O}(|S|n\Delta_{\max})$ |
| Space | $\mathcal{O}(n^2)$ | $\mathcal{O}(n^3)$ | $\mathcal{O}(n^3)$ | $\mathcal{O}(n + n\Delta_{\max})$ | $\mathcal{O}\left(n + \sum_{p=1}^{2} n_p(1 + B_p + 2\binom{B_p}{2})\right)$ | $\mathcal{O}(|S|(n + n\Delta_{\max}))$ |

Here, we analyze the complexity of our method, when using MPNNs as graph encoders, compared to other graph networks that are more expressive than 1-WL. In Table 14 we summarize the results. We assume that the feature dimensionality is a constant.

For dense graphs, our method requires $\mathcal{O}(|S|n^2)$ time, as each of the $|S|$ subgraphs is processed using a MPNN of time complexity $\mathcal{O}(n^2)$. When $|S| = \mathcal{O}(n)$ as in the case of node-deleted and ego-net policies, the time complexity is $\mathcal{O}(n^3)$, just as in the case of PPGN (Maron et al., 2019a)

and 3-IGN (Maron et al., 2019b). When $|S| = \mathcal{O}(n^2)$ as in the case of edge-deleted policies, the time complexity is $\mathcal{O}(n^4)$ like 3-GNN (Morris et al., 2019).

More specifically, the time complexity of our method improves to $\mathcal{O}(|S|n\Delta_{\max})$ where $\Delta_{\max}$ is the maximum node degree, which is low for sparse graphs. This is a major benefit of our method, as PPGN, k-IGN do not have time complexity improvements for sparse graphs, as they perform global aggregations instead of just local neighborhood aggregations. For sparse graphs where $\Delta_{\max} = \mathcal{O}(1)$, we gain a factor of $\mathcal{O}(n)$ in time complexity over PPGN and 3-IGN and a factor of $\mathcal{O}(n^2)$ over 3-GNN.

Table 14 also reports the time complexity of two provably expressive, *sparse* architectures: GSN (Bouritsas et al., 2022) and CWN (Bodnar et al., 2021a). The first method has the same complexity of standard GNNs as long as the number of considered subgraphs in the bank is a small constant. ESAN can match the complexity of GSN only in the case where $|S| = \mathcal{O}(1)$, which is only the case for stochastic policies that we consider. The remarkable efficiency of GSN, however, is paid in terms of preprocessing complexity (discussed in the next subsection). As for CWN, we report the time complexity for a generic lifting procedure generating a 2-dimensional cellular complex, with $n_p$ denoting the number of cells at dimension $p$ and $B_p$ the maximum considered boundary size. Here, $n_0 = n$, $n_1 = \mathcal{O}(n\Delta_{\max})$ refer to, respectively, the number of nodes and edges. In the case of ring-liftings with ring size upper-bounded a small constant, the complexity can be rewritten as $\mathcal{O}(n\Delta_{\max} + n_2)$. In the case of molecules, the number of rings, $n_2$, is generally contained. However, the number of 2-cells may grow even exponentially for general graph distributions. When employing policies ND, ED, EGO(+), on the contrary, our approach works with bags whose cardinality never exceeds the number of edges, irrespective of the graph distribution.

The space complexity of our method is $\mathcal{O}\left(|S|(n + n\Delta_{\max})\right)$, as we need to compute $n$ node features for each subgraph, while keeping the subgraph connectivity in memory. This evaluates to $\mathcal{O}(n^2 + n^2\Delta_{\max})$ for node-deleted and ego-net policies and $\mathcal{O}(n^2\Delta_{\max} + (n\Delta_{\max})^2)$ for the edge-deleted policy. We also note that the space complexity of our method can be improved to $\mathcal{O}(n + n\Delta_{\max})$ when using the DS-GNN architecture, as we may compute the GNN embedding of each subgraph online and accumulate them in one tensor that represents the sum over all subgraphs — though this requires sacrificing parallelism across subgraphs. Also, note that certain policies that significantly reduce the size of subgraphs, such as low-depth ego-net policies, are even more efficient because the number of edges in the subgraphs are much lower.

## F.2 PREPROCESSING

In order to perform message passing on subgraphs, it is first required to apply a selection policy and generate the bag. This amounts to a one-off computation that can be performed prior to training. Simple implementations of the ND and ED policies have complexities $\mathcal{O}(nm)$ and $\mathcal{O}(m^2)$, respectively, where $n$ is the number of nodes and $m$ is the number of edges. As noted above, $m$ can be upper-bounded by $n\Delta_{\max}$, with $\Delta_{\max}$ typically small in sparse graphs. EGO policies are easily implemented in $\mathcal{O}(n(n+m))$, where $\mathcal{O}(n+m)$ is the complexity of a generic Breadth First Search algorithm employed to recover the $k$-hop neighborhood of a source node. More sophisticated implementations of these selection policies may be possible, making the preprocessing step even more parsimonious. As pointed out above, the preprocessing complexity directly depends on the size and density of the graphs at hand; while practical run-times may be drastically reduced when working with sparse graphs, we remark that they can never exceed the aforementioned theoretical bounds, independently of the nature of graphs characterising the task being solved.

This is in contrast with the preprocessing complexity of GSN and CWN, which is generally $\mathcal{O}(n^k)$ for a generic substructure of size $k$. Preprocessing times are remarkably reduced in practice for certain families of graphs (e.g. planar graphs) and substructures for which specialised algorithms exists (e.g. triangles and rings). For example, these methods have been observed to attain tractable practical run-times on molecular datasets. Still, the worst-case exponential complexity may hinder their application in the presence of graphs of non-characterised distribution or when substructures of more efficient matching are not known to play a relevant role.

Finally, we remark that all these preprocessing strategies are embarrassingly parallel w.r.t. the different graphs in a dataset and parallelisation techniques can be applied to dramatically reduce the empirical run-time.

