# OpenReview forum: "Equivariant Subgraph Aggregation Networks"
_ICLR.cc/2022/Conference — ICLR 2022 Spotlight_

### Official Review · Reviewer_B3oK · 2021-10-20

**Correctness:** 3
**Technical Novelty And Significance:** 2
**Empirical Novelty And Significance:** 1
**Recommendation:** 6
**Confidence:** 4

**Main Review:**

### Strong Points
1. Searching more expressive GNN is hot topic and has a lot of importance both academic and industrial perspective. The main motivation and their approach is definitely eligible. This problem is one of the most important problems in GNN and recently there are a lot of different approaches.
2. The way of explanations and writing manner of the author are nearly perfect. Seems they spent a lot of effort to present the idea clear and didactic way.
3. Theorems and their proofs are clear and sufficient.
4. The researches that are in the same track with this paper, usually use predefined handcrafted features and/or structures which needs domain expertise among nearly infinite number of selections. Proposed approach does not need that kind of engineering efforts.

### Weak Points
1. Increasing expressive power of MPNN by applying baseline MPNN onto each subgraph of given graph and later somehow aggregate them in order to get graph level representation is very smart idea. However, It is not new at all. Recently, one (maybe concurrent research) already published the node deleted subgraph idea in [1]. Their proposal almost the same with DS-GNN in this paper. In addition their proposal of sampling strategy is exact the same. Just the differences is that in inference time, they used full sampling to get invariant graph representation.  That paper in [1] was not seen in this paper. I think the differences between these two papers are not significant.  But I am happy to hear the differences by author point of view.

2. If the baseline MPNN has linear time and memory complexity, proposed method with full sampling has quadratic complexity (when one node deleted subgraphs is used) that makes it infeasible easily w.r.t increasing number of nodes. Stochastic sampling of subgraphs seems reasonable solution, but this time the method lose the invariance property. Though this shortcoming, it may gives comparable results in the benchmark datasets, but how can we say it is more powerful than 1-WL in terms of separation power? It may separate non-isomorphic graphs but also it separates isomorphic graphs as well. Basically it separates all given pair of graphs because of stochastic sampling in inference time.

3. Experimental part of the paper is very weak unlikely to the theoretic parts.  Since TU datasets have limited number of instance and results standard deviation is very high, we cannot make reliable comparison. I think, TU datasets result does not deserve place in main paper but maybe in appendix for just giving some idea. In the paper, just EXP datasets is the unique result that gives idea about separation power of GNN. However, this dataset is very small and all pairs are 1-WL equivalent. The paper also mentioned SR dataset which consist of 3-WL equivalent graphs. I would like to see separation power analysis of SR dataset. I guess it is missing because of stochastic sampling. It basically separates all graph pairs even though pairs are isomorphic. Also Circular Skip Link dataset in (Murphy  2019) would be great expressive test of proposal method. But I strongly think that proposed methods fails, because that dataset includes isomorphic graphs that needs to be map on the same point in latent space. It is known that even though having more powerful GNN does not mean we will have better generalization on various datasets. But at least in some dataset such as Zinc12K, powerful methods outperformed 1-WL ones with a huge margin. I would like to see your method result on ZINC with compare to relevant recent powerful GNNs. In addition to Zinc, we know that in some artificial task such as substructure counting, 1-WL equivalent MPNNs fail ridiculously. So that kind of tasks would be great test to show your methods expressive power experimentally.

### Reference
[1] Cotta, L., Morris, C., & Ribeiro, B. (2021). Reconstruction for Powerful Graph Representations. NeurIPS 2021.

**Summary Of The Paper:**

This paper proposed a novel way of increasing MPNN expressive power without using higher order node tuples representation, thus proposed method has lower complexity compare to naive powerful GNN methods. The core idea is to create a set of subgraphs for each graph in the given dataset by using single node or edge deleting,...etc policy. They create a virtual graph that adjacency is summation of subgraph's adjacencies and node feature matrix is the summation of subgraph's feature matrix. Each subgraph's next layer representation will be obtained by aggregation of self subgraph's new representation by baseline MPNN and also this virtual graph's representation by another MPNN. In application this virtual graph is replaced by original graph or in another version is basically neglected. After arbitrary number of layers, they apply subgraph level pooling to obtain invariant representations. They see all subgraph's final representation as a set of representations. Thus to obtain final graph level representation, they apply another layer on set of subgraph representations such deepsets.




**Summary Of The Review:**

Since the idea is not original if Cotta et al. work is seen, and it has lack of experimental tests in initial submission as mentioned above, I would recommend rejection to this work.

But as the author mentioned that Cotta et al. work should be seen as contemporaneous work and there are a lot of additional test in second version, I changed my rate to 6.

---

> ### Author Response · Authors · 2021-11-14
> **Official Response to Reviewer B3oK**
>
> We are glad the Reviewer has found several, diverse strengths in our work. They have highlighted the industrial and academic importance of the tackled research problem, the clarity of presentation and completeness of theoretical analysis and the little hand-engineering required by the proposed approach, in contrast to recent related methods. The Reviewer has also expressed concerns regarding the novelty of our approach and the completeness of our experimental analysis. We address these here below.

---

> > ### Author Response · Authors · 2021-11-14
> > **Official Response to Reviewer B3oK 1/4**
> >
> > __Q__: _“Increasing expressive power of MPNN by applying baseline MPNN onto each subgraph of given graph and later somehow aggregate them in order to get graph level representation is very smart idea. However, It is not new at all. Recently, one (maybe concurrent research) already published the node deleted subgraph idea in [1] [...] That paper in [1] was not seen in this paper. I think the differences between these two papers are not significant. But I am happy to hear the differences by author point of view.”_
> >
> > __A__:  First, we would like to clarify that the work of Cotta et al. [1] appeared on arXiv only on Oct 1st, 2021, thus after the abstract submission deadline (Sept 29th, 2021). While we were not aware of this work at the time of our submission to ICLR 2022, we kindly remind the Reviewer that the ICLR instruction clearly qualifies such a paper as “contemporaneous work”:
> >
> > > We consider papers contemporaneous if they are published (available in online proceedings) __within the last four months__. That means, since our full paper deadline is October 5, if a paper was published (i.e., at a peer-reviewed venue) on or after June 5, 2021, authors are __not required__ to compare their own work to that paper.
> >
> > Hence, the aforementioned paper of Cotta et al. __is and must be considered concurrent to ours__ and shall not constitute a basis for diminishing the novelty of our work.
> >
> > However, as we believe it is beneficial for the community, and in order to address the Reviewer’s concerns, we will provide a comparison between our approach and that of Cotta et al. in the next revised version of our manuscript.
> >
> > The Reviewer has specifically asked to point out differences between the two works. Our framework is significantly more general and directly includes the Reconstruction GNN of Cotta et al. as a special case (Reconstruction GNN is exactly our DS-GNN with k-node-deleted policy). There are multiple substantial differences, which we discuss below.
> >
> > __Architecture for processing bags of subgraphs__: In the terminology of our paper, Cotta et al. use a DS-GNN architecture, while we perform a rigorous and more specific analysis of equivariance, and argue in favor of a different and more powerful DSS-GNN architecture. We show that DS-GNN is theoretically too restrictive in terms of the invariance it enforces (see p. 5, footnote 3 in our paper, and compare with the original DSS paper of Maron et al. 2020) and prove that DSS-GNN is strictly stronger than DS-GNN for certain policies (our Proposition 1). Our approach is thus theoretically more general and powerful than that of Cotta et al. Moreover, our DSS-GNN achieves better empirical results than DS-GNN in most of the tested settings, so this theoretical benefit leads to practical gains.
> >
> > __General subgraph policies__: We advocate general subgraph policies, four of which are studied in the paper (node-deletion, edge-deletion, ego-nets and ego-nets+). There are many other possible subgraph policies that fit into our framework, e.g. those based on spanning trees. In contrast, Cotta et al. only study the k-node-deletion policies.
> >
> > __More general theoretical analysis__: We provide a more general theoretical analysis. First, since we consider general subgraph selection policies (instead of just node-deleted policies as in [1]), we are able to prove results for many policies in the same way (e.g. our Proposition 3), whereas all of the results in [1] are obviously restricted to node-deletion. Second, we consider higher-order graph encoders in our general framework, whereas Cotta et al. only considers MPNNs and (the unrealistic case of) universal feed forward networks. We prove results on the integration of higher-order encoders in our ESAN framework in Proposition 2.
> >
> > __Formulation of new WL variants__: Finally, we also develop and analyze new variants of the Weisfeiler-Lehman test (DS-WL and DSS-WL). We then characterize DS-GNN and DSS-GNN in terms of these variants (Theorem 2), which helps us to prove other results in Section 3.3, and will help in proofs for future work. In comparison, there is no such WL variant developed by Cotta et al.
> >
> > __Basic motivation__: The paper of Cotta et al. is mainly motivated by the Reconstruction Conjecture in graph theory. In contrast, our paper is motivated by the recent work by [Maron et al. 2020] on equivariant architectures w.r.t. product symmetry groups arising when dealing with sets of objects containing internal symmetries.
> >
> > [Maron et al. 2020] Maron H, Litany O, Chechik G, Fetaya E. “On learning sets of symmetric elements.” ICML 2020

---

> > > ### Author Response · Authors · 2021-11-14
> > > **Official Response to Reviewer B3oK 2/4**
> > >
> > > __Q__: _“If the baseline MPNN has linear time and memory complexity, proposed method with full sampling has quadratic complexity (when one node deleted subgraphs is used) that makes it infeasible easily w.r.t increasing number of nodes.”_
> > >
> > > __A__: In most practical applications considered in our paper (including all molecular and social graph benchmarks we performed), no sampling was necessary as it was possible to run our approach in a deterministic (and fully invariant) fashion, both in training and inference, with more than tractable runtimes. Empirical runtimes reported in Table 8 Appendix D: irrespective of the selection policy, one training epoch on the 4,110 graphs of NCI1 is conducted in around 2 seconds with the DS-GNN(GIN) architecture and slightly more than 3 seconds with the DSS-GNN(GIN) one, while GIN takes around 1 second for the same amount of work. As recommended by the Reviewer, we have also experimented on ZINC12k (see below). This datasets comprises 10k training graphs. From further timing experiments we observed that a single epoch takes between 3 and 3.5 seconds with DS-GNN, and between 3.7 and 4.25 seconds with DSS-GNN (while the base encoder requires slightly less than 1.5 seconds).
> > >
> > >
> > > __Q__: _“Stochastic sampling of subgraphs seems reasonable solution, but this time the method lose the invariance property. Though this shortcoming, it may gives comparable results in the benchmark datasets, but how can we say it is more powerful than 1-WL in terms of separation power? It may separate non-isomorphic graphs but also it separates isomorphic graphs as well.”_
> > >
> > > __A__: We study stochastic sampling of subgraphs with the specific aim of easing the practical application of our framework in those use cases where the graphs are too large. We would like to stress that the stochasticity inherent in the sampling approach is not a native, intrinsic component of our theoretical framework (contrary to other related works such as those making use of random node identifiers [2]). For this reason, we believe that extending Theorem 1 to the subsampling regime – and even formalizing these research questions in the first place – is a significant theoretical endeavor warranting a separate paper and thus out of scope of our present work. We are grateful for this suggestion and will explore it in future work.
> > >
> > > In practice, while it is indeed true that stochastic sampling generally breaks the theoretical invariance of our approach, there might still be approximate invariance. Graph representations obtained from different samplings of the same bag would nonetheless be relatively close, allowing our models to map them to the same target class. This is empirically verified on both real world and synthetic datasets: we refer the reviewer to Tables 6,7 and, in particular, to the results on the CSL dataset that we report in the following.
> > >
> > > __Q__: _“Experimental part of the paper is very weak unlikely to the theoretic parts.”_
> > >
> > > __A__: We respectfully disagree with the Reviewer on this point and kindly ask to reconsider this point in light of our response below.
> > > We believe that the obtained empirical results clearly answer the research questions of interest that we have listed at the beginning of Section 4, and the experimental section of our work is fairly comprehensive and in line with the validation performed in related papers published in the field. We have experimented with 7 TU, 2 RNI and 2 OGB datasets, for a total of 11 distinct benchmarks. On each of them, we have studied the performance of all 4 subgraph selection policies analysed in the paper (ND, ED, EGO, EGO+) and of distinct base graph encoders, including GIN, GraphConv and GCN. In addition to this, we have further experimented with subgraph sampling at 3 distinct rates (5%, 20%, 50%). Moreover, we followed the suggestion of the Reviewer and experimented also on ZINC and CSL datasets (other than the additional RDT-B dataset). These new results will be reported in the next paper revision.
> > >
> > > __Q__: _“In the paper, just EXP datasets is the unique result that gives idea about separation power of GNN. However, this dataset is very small and all pairs are 1-WL equivalent.”_
> > >
> > > __A__: We believe that this dataset represents, on the contrary, a sound testbed to empirically verify the discriminative power of our method. It is in view of the fact that all graph pairs are 1-WL equivalent that this benchmark is well-suited to practically validate Theorem 1 in the main paper, arguably amongst the key contributions in our work.

---

> > > > ### Author Response · Authors · 2021-11-14
> > > > **Official Response to Reviewer B3oK 3/4**
> > > >
> > > > __Q__: _“Also Circular Skip Link dataset in (Murphy 2019) would be great expressive test of proposal method. But I strongly think that proposed methods fails, because that dataset includes isomorphic graphs that needs to be map on the same point in latent space.”_
> > > >
> > > > __A__: The Reviewer recommends experimenting with the CSL dataset. We note that even this dataset contains 1-WL-equivalent graph pairs and is therefore another suitable testbed for validating our theoretical results. We have therefore benchmarked our method over this dataset as well and we will report the results in the next paper revision. Contrary to what is (strongly) hypothesized by the Reviewer, we were able to run our DS-GNN model in a deterministic fashion with no subgraph sampling, achieving  __100% test accuracy__ with our deterministic node and edge-deletion policies, for both GIN and GraphConv base encoders.
> > > > We understand that the Reviewer’s concern was also specifically related to the potential loss of invariance when stochastic sampling is adopted. In order to address this, we additionally experimented with stochastic subgraph selection with sampling rates of 5%, 20% and 50%. Again, we obtained perfect test accuracy on this dataset with the node and edge-deletion selection policies. This numerically confirms our intuition that the studied stochastic selection policies still guarantee a form of approximate invariance which allows our models to map isomorphic graphs to the same target isomorphism class. Our final version will include the results of our method on the CSL dataset for all policies.

---

> > > > > ### Author Response · Authors · 2021-11-14
> > > > > **Official Response to Reviewer B3oK 4/4**
> > > > >
> > > > > __Q__: _“[...] But at least in some dataset such as Zinc12K, powerful methods outperformed 1-WL ones with a huge margin. I would like to see your method result on ZINC with compare to relevant recent powerful GNNs.”_
> > > > >
> > > > > __A__: We would like to remark that we have already conducted extensive experimentations on two similar benchmarks (_ogbg-molhiv_ and _ogbg-moltox21_). On these molecular benchmarks, our approach has brought significant performance improvements compared to standard 1-WL upper-bounded base encoders (GCN and GIN). This experimentally verifies the intuition that larger expressive power may be beneficial in this kind of benchmarks.
> > > > >
> > > > > In addition to the above, we have also performed experiments on the ZINC12K benchmark, as suggested by the Reviewer. We followed all benchmarking guidelines indicated in [Dwivedi et al., 2020], including imposing a 100k parameter budget for fairness of comparison across all methods. Results are reported in the table below, where we have also reported the performance of other provably expressive GNNs. These results will be included in the next manuscript revision.
> > > > >
> > > > > | Model | ZINC12K test MAE |
> > > > > |--------------|:--------------------------:|
> > > > > | GIN | 0.252 ± 0.017 |
> > > > > | PNA | 0.188 ± 0.004 |
> > > > > | DGN | 0.168 ± 0.003 |
> > > > > | SMP | 0.138 ± ???  |
> > > > > | | |
> > > > > | DS-GNN (GIN) (ND) | 0.171 ± 0.010 |
> > > > > | DS-GNN (GIN) (ED) | 0.172 ± 0.008 |
> > > > > | DS-GNN (GIN) (EGO) | 0.126 ± 0.006 |
> > > > > | DS-GNN (GIN) (EGO+) | 0.116 ± 0.009 |
> > > > > | | |
> > > > > | DSS-GNN (GIN) (ND) |  0.166 ± 0.004  |
> > > > > | DSS-GNN (GIN) (ED) | 0.172 ± 0.005 |
> > > > > | DSS-GNN (GIN) (EGO) | 0.107 ± 0.005  |
> > > > > | DSS-GNN (GIN) (EGO+) |  ___0.102 ± 0.003___  |
> > > > > | | |
> > > > > | HIMP | 0.151 ± 0.006 |
> > > > > | GSN  | 0.108 ± 0.018 |
> > > > > | CIN | __0.094 ± 0.004__ |
> > > > >
> > > > > First, we observe that all ESAN configurations significantly outperform their base GIN encoder, while having a comparable number of learnable parameters. This is in line with the general trend emerging from the rest of experimental results already included in the manuscript.
> > > > >
> > > > > Second, we notice that our method achieves particularly competitive results, irrespective of the chosen subgraph selection policy. All our configurations outperform the provably expressive PNA model [Corso et al. 2020]. At the same time, when equipped with the EGO(+) selection policies, our approach outperforms HIMP [Fey et al. 2020], DGN [Beaini et al. 2021] as well as SMP [Vignac et al. 2020] and GSN [Bouritsas et al. 2020], whose expressive power is _strictly greater_ than standard MPNNs.
> > > > >
> > > > > CIN [Bodnar et al. 2021] (for which we report the budget-compliant results) is the only method outperforming ESAN. However, a large contribution to its performance is given by the explicit modeling of cycles / rings, which are listed beforehand in a preprocessing step. Ring structures are known as powerful inductive biases in organic chemistry and, according to the ablation study reported in the aforementioned paper (Table 9, Appendix E), this indeed represents a pivotal factor in explaining its test performance, more than its ability to disambiguate non-isomorphic graphs.
> > > > >
> > > > > Similarly to CIN, HIMP and GSN also explicitly employ information from rings and their counts, yet are outperformed by ESAN. This is an important result, because, contrary to these models, the policies considered in our paper are not tailored for specific application domains – something that, on the other hand, can be advantageous in other settings.
> > > > >
> > > > > Given the above, we can conclude that, on the ZINC12K benchmark:
> > > > > - ESAN is _the best performing model_ amongst all provably expressive, domain agnostic GNNs;
> > > > > - ESAN is competitive with (provably powerful) GNNs employing domain specific structural information, and it _often outperforms them_.
> > > > >
> > > > > We believe these to be strong results and that they positively address the Reviewer’s request.

---

> > > > > > ### Author Response · Authors · 2021-11-14
> > > > > > **Official Response to Reviewer B3oK 4/4 (references)**
> > > > > >
> > > > > > [Dwivedi et al., 2020] Vijay Prakash Dwivedi, Chaitanya K Joshi, Thomas Laurent, Yoshua Bengio, and Xavier Bresson. Benchmarking graph neural networks. arXiv preprint arXiv:2003.00982, 2020.
> > > > > >
> > > > > > [Corso et al. 2020] Gabriele Corso, Luca Cavalleri, Dominique Beaini, Pietro Liò, and Petar Velickovič. “Principal neighbourhood aggregation for graph nets”. NeurIPS 2020
> > > > > >
> > > > > > [Fey et al. 2020] M. Fey, J. G. Yuen, and F. Weichert. “Hierarchical inter-message passing for learning on molecular graphs”. ICML Graph Representation Learning and Beyond (GRL+) Workshop 2020.
> > > > > >
> > > > > > [Beaini et al. 2021] Dominique Beaini, Saro Passaro, Vincent Létourneau, William L Hamilton, Gabriele Corso, and Pietro Liò. “Directional graph networks”. ICML, 2021.
> > > > > >
> > > > > > [Vignac et al. 2020] Clément Vignac, Andreas Loukas, and Pascal Frossard. “Building powerful and equivariant graph neural networks with structural message-passing”. NeurIPS 2020
> > > > > >
> > > > > > [Bouritsas et al. 2020] Giorgos Bouritsas, Fabrizio Frasca, Stefanos Zafeiriou, and Michael M Bronstein. “Improving graph neural network expressivity via subgraph isomorphism counting”. arXiv preprint arXiv:2006.09252, 2020.
> > > > > >
> > > > > > [Bodnar et al. 2021] Cristian Bodnar, Fabrizio Frasca, Nina Otter, Yu Guang Wang, Pietro Liò, Guido Montúfar, and Michael Bronstein. “Weisfeiler and lehman go cellular: CW networks”. NeurIPS 2021.

---

> ### Comment · Reviewer_B3oK · 2021-11-25
> **Response to the authors**
>
> I really appreciate the extra efforts on additional tests. Zinc dataset results clearly provide experimental evidence on how powerful the ESAN is. I was wrong on CSL dataset. Because I assumed it consist of strongly regular graphs with the same parameter (3-WL equivalent) But actually they are strongly regular graph with different parameter (1-WL equivalent). So it is expected that any method powerful than 1-WL equivalent solve it. Still I think we should have tests on 3-WL equivalent graphs and more results on artificial dataset to show its expressive power experimentally such as substructure counting.
>
> For the  paper by Cotta et al, even though the motivations are not the same, at the end, the proposal is the same.
> However, the authors are right we should count them as contemporaneous work.
>
> Despite of there is no single subgraph selection policy (we do not know which and why we need to select the one) and mentioned missing experimental expressive power test, the theoretical study makes valuable contribution on how to obtain more powerful GNN, thus I changed score to 6.

---

### Official Review · Reviewer_tGsP · 2021-10-27

**Correctness:** 4
**Technical Novelty And Significance:** 4
**Empirical Novelty And Significance:** 3
**Recommendation:** 8
**Confidence:** 4

**Main Review:**

This paper develops a Equivariant Subgraph Aggregation Network to enhance the
expressive power of GNNs. The idea is to 1) represent graphs as bags of
subgraphs 2) use permutation-equivariant layers to encode the subgraphs 3)
aggregate the subgraph representations into an invariant graph representation.

Strength:

S1. The idea of using equivariant subgraph representation and aggregation is novel.

S2. Very solid theoretical analysis showing the expressive power of DS-GNN and DSS-GNN.

S3. The experiments are comprehensive as they conducted experiments on various
real and synthetic graph classification datasets and include most existing
methods on expressive GNNs. The results are mostly positive, showing the
benefits of DS/DSS-GNN architectures.

Weakness:

W1. It'll be better if the authors can perform a detailed breakdown. Of the
experimental results to show. To provide some insights above, when does the
DSS. Architectural works well and when does it not? The.

W2. The model introduces extra time overhead. Although the authors introduced
a stochastic strategy that can mitigate this issue to some extent, the time
cost is still several times larger than standard GNNs. Also, it's better to
show how the method scales with graph sizes.

W3. How will this method work for other graph related tasks beyond graph classification?


**Summary Of The Paper:**

This paper aims to improve GNNs' expressive power based on bag-of-subgraph
representation. The authors develop a Equivariant Subgraph Aggregation
Network, which uses equivariant layers to encode subgraphs and aggregate
subgraph representations. The authors have performed theoretical analysis of
the proposed framework, subgraph selection policies, and their expressive
power. The experiments show that such an encoding can lead to a better
expressive power on several real and synthetic graph classification datasets.


**Summary Of The Review:**

This is a solid paper on improving the expressive power of graph neural networks. The proposal approach of equivariant subgraph representation and aggregation is novel, the theoretical analysis is solid, and the results are positive.

---

> ### Author Response · Authors · 2021-11-14
> **Official Response to Reviewer tGsP**
>
> The reviewer has found our idea novel, while appreciating the solidity of the theoretical analysis, the comprehensiveness of our experimentation and the positive emerging results. They have nonetheless raised a few questions that we address below.
>
> __Q__: _“It'll be better if the authors can perform a detailed breakdown. Of the experimental results to show. To provide some insights above, when does the DSS. Architectural works well and when does it not?”_
>
> __A__: We will include a paragraph for a more general discussion of the experimental results in the next revision of our manuscript. In general, a strong emerging conclusion is that our approach tends to perform better than the utilised base encoder. In fact, in 91% of the cases DSS-GNN outperforms its base graph encoders, while this happens in 75% of the cases for the DS-GNN architecture. Thus, we believe that applying our ESAN approach is generally beneficial whenever some additional computational resources are available.
>
> More specifically, the application of our approach is recommended in all those cases where the task at hand may require expressive power beyond 1-WL. There is some empirical evidence that certain molecular benchmarks fall in this category; our new results on the ZINC dataset validate this hypothesis by showing that our approach strongly outperforms GIN (while performing best amongst all provably expressive, domain agnostic GNNs).
>
> Finally, ESAN is especially suitable in those cases where no domain knowledge is available about graph substructures relevant to the task being solved. As it requires little pre-engineering, our method represents a valid, flexible and provably expressive approach in these conditions.
>
> We are not aware of any particular scenario where the application of ESAN has proved to be consistently unsuccessful. We have observed it to not work well in conjunction with GCN encoders over the OGB datasets, but this would require further future investigation before drawing concrete conclusions.
>
>
> __Q__: _“The model introduces extra time overhead. Although the authors introduced a stochastic strategy that can mitigate this issue to some extent, the time cost is still several times larger than standard GNNs. Also, it's better to show how the method scales with graph sizes.”_
>
> __A__: Working with a bag of subgraphs inevitably introduces additional computational complexity. However, we would like to stress that, _in practice_, this has never prevented the application of our full deterministic method on any of the benchmarks we have utilized in our work, including the large scale _ogbg-molhiv_ molecular dataset. In fact, the additional complexity has always been observed to be tractable. The parallelism achieved by GPU kernels implementing tensor operations and the typical sparsity of real-world graphs play an important role in this sense.
>
> Moreover, we believe that the subgraph selection strategy we study in our work is a particularly promising strategy for the application of our approach to larger, denser graphs. Posterior to the first submission of our manuscript we have conducted additional experiments on the large RDT-B TU benchmark, with an average of ~ 430 nodes and ~ 498 edges per graph. A stochastic selection strategy with 5% (!) sampling rate allowed us to seamlessly apply our method on this dataset, while performing better than the base encoder for most ESAN configurations. We will include these results in the next revision of the manuscript.
>
> Aside from that, we will make efforts to add an experiment that shows the runtime vs the graph size as the reviewer requested during the discussion period, and in any case, we will include it in the final version.
>
>
> __Q__: _“How will this method work for other graph related tasks beyond graph classification?”_
>
> __A__: Our method is easy to adapt to other tasks, but we focus on graph-level prediction in this work (both theoretically and practically) and leave further theory and experiments to future works. We do expect that some of our policies would work well for different tasks. For instance, DropEdge [Rong et al. 2020] is a stochastic version of an edge-dropping policy, and it was originally introduced and tested on node classification, where it can substantially improve empirical results.
>
> [Rong et al. 2020] Rong Y, Huang W, Xu T, Huang J. “Dropedge: Towards deep graph convolutional networks on node classification”. ICLR 2020.

---

### Official Review · Reviewer_rtwc · 2021-11-01

**Correctness:** 4
**Technical Novelty And Significance:** 4
**Empirical Novelty And Significance:** 2
**Recommendation:** 8
**Confidence:** 4

**Main Review:**

Pros:
- This paper gives solid theoretical formulation and analysis of graph representation learning over bag of subgraphs. The technical contribution of this paper is sound and significant.
- This paper gives detailed formal analysis of design choices such as base graph encoders and subgraph selection policies.
- The presentation of this paper is clear.

Cons:
- In the experiment results, some DS(S)-GNN variants have inferior performance than basic graph encoders on some graphs.  Can there be any insights or guidance on whether to use DS(S)-GNNs for specific kinds of graph data? As we might want to quickly estimate this DS(S) architecture can be beneficial before actually conducting its heavier computation.


Some work that might be relevant:
- [1] is on the empirical side of using the aggregation of subgraph representation to represent a graph. It tries to factorize the original graph into disentangled subgraphs, then to utilize the factor subgraph embedding to represent the whole graph.
- [2] shares similar high-level ideas with this paper, which is to encode the graph through different subgraph channels then aggregate them.

[1] Factorizable Graph Convolutional Networks, Yiding Yang, Zunlei Feng, Mingli Song, Xinchao Wang, NeurIPS 2020

[2] Graph Neural Network with Automorphic Equivalence Filters, Fengli Xu, Quanming Yao, Pan Hui, Yong Li, arxiv 2020


**Summary Of The Paper:**

This paper proposes to improve the expressivity of MPNN by using H-Equivariant layers to process bags of subgraphs accounting for their natural symmetry. The proposed DS(S)-GNN is proven to be equivalently powerful as DS(S)-WL, which can be strictly more powerful than 1-WL. It also provides theoretical results about how design choices such as the subgraph selection policy and equivariant neural architecture affect the architecture’s expressive power. In addition, a stochastic sampling scheme is proposed to mitigate the computational overhead.


**Summary Of The Review:**

This paper provides a solid foundation on the theoretical formulation and analysis of graph representation learning over bag of subgraphs. The contribution is technically sound and the solution is well supported by detailed analysis of design choices. Thus I recommend acceptance.

---

> ### Author Response · Authors · 2021-11-14
> **Official Response to Reviewer rtwc**
>
> We are glad to notice that the Review deems our theoretical formulation solid and our technical contributions sound and significant, while appreciating the detailed formal analysis of design choices and the overall presentation.
>
> The Reviewer has pointed out the existence of two papers representing relevant previous work. We thank the Reviewer for this; we will make sure to refer to them in the next revision of our manuscript.
>
> Finally, the Reviewer had a question about some experimental results and on the general choice of our method.
>
> __Q__: _“In the experiment results, some DS(S)-GNN variants have inferior performance than basic graph encoders on some graphs. Can there be any insights or guidance on whether to use DS(S)-GNNs for specific kinds of graph data? As we might want to quickly estimate this DS(S) architecture can be beneficial before actually conducting its heavier computation.”_
>
> __A__: We thank the reviewer for their question. First, we would like to stress that in the vast majority of our experiments, DS(S) variants outperformed the base encoders: on TU datasets, DSS-GNN is better 91% of the time while DS-GNN is better 75% of the time. Therefore, we believe it is fair to say that if extra computational resources are available, in most cases, using DS(S)-GNN is a better option. It should also be noted that the extra computational cost is not as severe as it might seem. For example, a single epoch of the NCI1 (~ 4k training graphs) dataset takes around 2 seconds with DS-GNN, 3 seconds with DSS-GNN, and 1 second with the base encoder. On the ZINC12K (10k training graphs) a single epoch takes between 3 and 3.5 seconds with DS-GNN, and between 3.7 and 4.25 seconds with DSS-GNN (while the base encoder requires slightly less than 1.5 seconds).
>
> Having said that, we believe our theoretical results might also help guide practitioners: In Section 3.3, we theoretically study how various design choices of our DS(S)-GNN variants affect expressive power. Finally, in the next revision of the paper we will additionally include a paragraph for an overall discussion of the empirical results. We believe this may also help in this sense.

---

> > ### Comment · Reviewer_rtwc · 2021-11-25
> > **After response**
> >
> > Thanks to the authors for their explanations and efforts to make the practicality more clear. I believe this work makes nice contributions to this community.

---

### Official Review · Reviewer_Ebj3 · 2021-11-02

**Correctness:** 3
**Technical Novelty And Significance:** 3
**Empirical Novelty And Significance:** 2
**Recommendation:** 6
**Confidence:** 4

**Main Review:**

Strengths:
- The paper is well-written, easy to read and provides lots of details at the same time.
- The proposed idea is rather simple, which is kind of an advantage and a disadvantage at the same time.
- The proposed method shows (small) improvements over the base encoder in many cases.

Weaknesses:
- Even though the proposed method shows improvements, it should be mentioned that the improvements are rather smaller, especially given the increased complexity and computing requirements of the proposed method. Hence, it seems rather unlikely that it will be widely adopted in its current form.
- The paper does not provide details about the size of the models (e.g. in terms of number of trainable parameters). This problematic since it remains unclear if the performance improvement stems from the increases expressiveness or is simply due to a larger model size.
- Highlighting the top three models (e.g. in Table 1) is highly misleading since only one version of each prior method is compared against 16 versions of the proposed idea. In general, this comparison does not make sense since the propose method has 16 trials to achieve a good result while all reference models only have a single trial.
- No code is provided to verify the results.

**Summary Of The Paper:**

The paper addresses known limitations of the expressiveness of message passing neural networks (MPNNs). Key idea is to generate subgraphs from the original graph, to encode each subgraph with a GNN, and to aggregate the resulting set of subgraph-encodings. Since generating all possible subgraphs is computational infeasible for larger graphs, only a smaller subset of subgraphs is used in practice. The paper shows that their approach is more expressive than 1-WL. Moreover, the paper shows that different variants of the proposed method can outperform the base model.

**Summary Of The Review:**

To summarize, the paper proposes a fairly simple but reasonable idea that achieves some improvements. However, since the improvements are rather small and the complexity of the model increases substantially, it is unclear if this idea will receive a lot of attention.

---

> ### Author Response · Authors · 2021-11-14
> **Official Response to Reviewer Ebj3**
>
> We are delighted to see the Reviewer has appreciated the simplicity of the idea, the way it has been illustrated throughout the manuscript and the overall level of detail. At the same time, some issues were raised regarding the presentation of experimental results and their general significance in view of the additional computational complexity. We proceed here below in replying to these comments in specific.

---

> > ### Author Response · Authors · 2021-11-14
> > **Official Response to Reviewer Ebj3 1/3**
> >
> > __Q__: _“Even though the proposed method shows improvements, it should be mentioned that the improvements are rather smaller”_
> >
> > __A__: Our main experimental goal was to show that the presented ESAN framework can generally improve the performance of base encoders, and we believe that our results give us a strong positive answer in this regard: we could observe a general improvement trend spanning across different policies, encoder architectures and application domains.
> >
> > On the synthetic RNI datasets the improvement is more than evident and confirms our theoretical results.
> >
> > On the OGB datasets, we observed significant improvements w.r.t. base encoders. As we already reported in the paper, when using GIN, all ESAN variants improve the base encoder accuracy by up to __2.4%__ absolute improvement (__3.2%__ relative) on _ogbg-molhiv_ and __3%__ absolute improvement (__4%__ relative) on _ogbg-moltox21_. We report below the performance obtained by our best ESAN models:
> >
> > |              | ogbg-molhiv (DS-GNN / GIN / EGO) | ogbg-moltox21 (DSS-GNN / GIN / EGO+ ) |
> > |--------------|:--------------------------:|:------------------------:|
> > | Base  | 75.58 ± 1.40                 | 74.91 ± 0.51            |
> > | ESAN  | 78.00 ± 1.42                 | 77.95 ± 0.40            |
> >
> > We notice from these results that our mean test performance is ~ 1.7 (_ogbg-molhiv_) and ~ 6.0 (_ogbg-moltox21_) standard deviations above the mean performance of the base encoder.
> >
> > On TUDatasets, DSS-GIN and DSS-GraphConv almost always outperform their corresponding GIN and GraphConv base encoders on all datasets, irrespective of the chosen selection policy. In certain cases, the performance gap is relatively large (for these benchmarks):
> >
> > |              | PTC ( DSS-GNN / GIN / EGO+ ) | NCI1 ( DS-GNN / GIN / ND ) | IMDB-B ( DSS-GNN / GIN / EGO+ ) |
> > |--------------|:--------------------------:|:------------------------:|:------------------------------:|
> > | Base  | 64.6 ± 7.0                 | 82.7 ± 1.7               | 75.1 ± 5.1                    |
> > | ESAN  | 69.2 ± 6.5                 | 83.8 ± 2.4               | 77.1 ± 3.0                    |
> >
> > We deem these results to be significant in particular because of the following reasons:
> > - The hyperparameters of the base encoders were fairly tuned;
> > - The results obtained by our models are on-par, and in some cases better than, those of other state-of-the-art methods such as GSN [Bouritsas et al. 2020] and CIN [Bodnar et al. 2021]. Contrary to us, these use more complex message passing layers which explicitly include domain-specific information.
> >
> > Finally, we have additionally experimented with the ZINC12K benchmark. We observed ESAN to significantly outperform the base encoder (GIN), while obtaining results close to the state-of-the-art. Our model  _outperformed all provably expressive, domain agnostic GNNs_ (GIN, PNA [Corso et al. 2020], DGN [Beaini et al. 2021] , SMP [Vignac et al. 2020]), along with other GNNs employing domain-specific structural information (HIMP [Fey et al. 2020], GSN [Bouritsas et al. 2020]). We will report these results and discuss them in the next paper revision.
> >
> > | Model | ZINC12K test MAE |
> > |--------------|:--------------------------:|
> > | GIN | 0.252 ± 0.017 |
> > | PNA | 0.188 ± 0.004 |
> > | DGN | 0.168 ± 0.003 |
> > | SMP | 0.138 ± ???  |
> > | | |
> > | DS-GNN (GIN) (ND) | 0.171 ± 0.010 |
> > | DS-GNN (GIN) (ED) | 0.172 ± 0.008 |
> > | DS-GNN (GIN) (EGO) | 0.126 ± 0.006 |
> > | DS-GNN (GIN) (EGO+) | 0.116 ± 0.009 |
> > | | |
> > | DSS-GNN (GIN) (ND) |  0.166 ± 0.004  |
> > | DSS-GNN (GIN) (ED) | 0.172 ± 0.005 |
> > | DSS-GNN (GIN) (EGO) | 0.107 ± 0.005  |
> > | DSS-GNN (GIN) (EGO+) |  ___0.102 ± 0.003___  |
> > | | |
> > | HIMP | 0.151 ± 0.006 |
> > | GSN  | 0.108 ± 0.018 |
> > | CIN | __0.094 ± 0.004__ |
> >
> > [Corso et al. 2020] Gabriele Corso, Luca Cavalleri, Dominique Beaini, Pietro Liò, and Petar Velickovič. “Principal neighbourhood aggregation for graph nets”. NeurIPS 2020
> >
> > [Fey et al. 2020] M. Fey, J. G. Yuen, and F. Weichert. “Hierarchical inter-message passing for learning on molecular graphs”. ICML Graph Representation Learning and Beyond (GRL+) Workshop 2020.
> >
> > [Beaini et al. 2021] Dominique Beaini, Saro Passaro, Vincent Létourneau, William L Hamilton, Gabriele Corso, and Pietro Liò. “Directional graph networks”. ICML, 2021.
> >
> > [Vignac et al. 2020] Clément Vignac, Andreas Loukas, and Pascal Frossard. “Building powerful and equivariant graph neural networks with structural message-passing”. NeurIPS 2020
> >
> > [Bouritsas et al. 2020] Giorgos Bouritsas, Fabrizio Frasca, Stefanos Zafeiriou, and Michael M Bronstein. “Improving graph neural network expressivity via subgraph isomorphism counting”. arXiv preprint arXiv:2006.09252, 2020.
> >
> > [Bodnar et al. 2021] Cristian Bodnar, Fabrizio Frasca, Nina Otter, Yu Guang Wang, Pietro Liò, Guido Montúfar, and Michael Bronstein. “Weisfeiler and lehman go cellular: CW networks”. NeurIPS 2021.

---

> > > ### Author Response · Authors · 2021-11-14
> > > **Official Response to Reviewer Ebj3 2/3**
> > >
> > > __Q__: _“[...] improvements are rather smaller, especially given the increased complexity and computing requirements of the proposed method. Hence, it seems rather unlikely that it will be widely adopted in its current form.”_
> > >
> > > __A__: The reviewer mentions the “increased complexity and computing requirements” that may prevent the future application of our method. We deem it important to stress that, in practice, the additional computational complexity has always demonstrated to be empirically tractable in all our experiments. As an example, irrespective of the selection policy, one training epoch on the NCI1 dataset (4,110 graphs) only took around 2 seconds with the DS-GNN(GIN) architecture and slightly more than 3 seconds with the DSS-GNN(GIN) one (while GIN takes around 1 second), see table 8 in appendix D for further details. Further timing experiments on ZINC12K confirmed the empirically tractable computational complexity of our approach: for a training epoch on 10k graphs, DS-GNN required only between 3 and 3.5 seconds (depending on the policy), while DSS-GNN required between 3.7 and 4.25 seconds (the base encoder ran in slightly less than 1.5 seconds).
> > >
> > > This empirical tractability is mostly due to the intrinsic parallelism of tensor operations that can be exploited by GPU kernels. In addition, real-world graphs are typically sparse and of bounded degree, thus easing the practical complexity of the ED and EGO(+) policies.
> > >
> > > We would like to also stress the strongly positive results obtained by our experiments with stochastic subgraph selections, which can significantly mitigate the computational complexity of our approach. In practice, we observed this strategy to have minimum impact on the performance of our models (see Tables 6 and 7 in Appendix D). Furthermore, the sampling strategy allows us to apply our datasets to larger graphs. On new experiments over the RDT-B dataset, with ~ 430 nodes and ~ 498 edges per graph on average, stochastic sampling at 5% rate was sufficient to seamlessly run our model and outperform the base encoder in most of its configurations (results will be added in the paper revision). In view of the aforementioned considerations, we are confident that the increased complexity of our approach will not directly hinder practical applications, nor limit the attention that our theoretical framework may receive within the research community.
> > >
> > > We would finally like to remark that, in our paper, we have presented a _general_ framework offering complete flexibility in the choice of base encoders and selection policies. Future developments can explore potentially better combinations of these two, and even new selection policies that may be more suited to particular application domains for even stronger empirical results.

---

> > > > ### Author Response · Authors · 2021-11-14
> > > > **Official Response to Reviewer Ebj3 3/3**
> > > >
> > > > __Q__: _“The paper does not provide details about the size of the models (e.g. in terms of number of trainable parameters). This problematic since it remains unclear if the performance improvement stems from the increases expressiveness or is simply due to a larger model size.”_
> > > >
> > > > __A__: Similarly to our models, we have carefully optimized the hyperparameters of the baselines (i.e. GIN, GraphConv and GCN), which often produced better results than the results in their original papers. We also note that in many cases the best performing models are not necessarily the ones with the maximal number of parameters. This demonstrates that, while a larger number of parameters tend to correlate with larger model capacity, it does not necessarily correlate with generalization performance, which is what we are really interested in.
> > > > Nonetheless, in order to further verify the fairness of our results, we have conducted additional experiments, training larger GIN architectures over those TU datasets where we found our best ESAN variant to comprise a larger number of trainable weights. In each of these experiments, we have enlarged the dimension of the hidden layers of the GIN baseline in a way to approximately match the overall number of ESAN parameters. We dub these models “BIG-GIN”. The results are reported here in the table below and will be included in the next paper revision. Out of all the seven TU datasets, three of them (IMDB-B, IMDB-M, PTC), are those where ESAN uses _fewer_ parameters. On the four remaining ones, we observed that, on average, adding more parameters to the baselines produced slightly worse results and did not change the ranking of the methods. Using baselines with more parameters yielded marginally worse results for two datasets (PROTEINS, NCI109) and only insignificantly better results on the remaining two datasets (NCI1, MUTAG).
> > > >
> > > > _Number of parameters of best models_
> > > >
> > > > |    |  MUTAG  |  PTC | PROTEINS | NCI1 | NCI109 | IMDB-B | IMDB-M |
> > > > |:---|:---:|:---:|:---:|:---:|:---:|:---:|:---:|
> > > > | GIN              | 8296   | 8692 |  8164  |  9286  |  9319  | 39305 |  40091 |
> > > > | ESAN (GIN) | 18625 | 5601 | 10865 | 47137 | 20481 | 33729 |  26883 |
> > > >
> > > > _Results with larger number of trainable weights_
> > > >
> > > > |   |  MUTAG  |   PROTEINS | NCI1 | NCI109 |
> > > > |:---|:---:|:---:|:---:|:---:|
> > > > | GIN  | 89.4 ± 5.6   | 76.2 ± 2.8 | 82.7 ± 1.7 | 82.2 ± 1.6 |
> > > > | BIG-GIN  | 89.9 ± 4.9  | 75.8 ± 5.5 |  82.9 ± 1.8 | 81.6 ± 1.5  |
> > > > | ESAN  |  91.1 ± 7.0  | 77.1 ± 4.6 | 83.8 ± 2.4 | 83.1 ± 0.8 |
> > > >
> > > > We would also like to refer the Reviewer to the new results obtained on the ZINC benchmark, where both DS-GNN and DSS-GNN significantly outperformed the base encoder while having an even smaller number of parameters:
> > > >
> > > > | Model | Test MAE | Num. of parameters |
> > > > |--------------|:--------------------------:|:--------------------------:|
> > > > | GIN | 0.252 ± 0.017 | 139525 |
> > > > | DSS-GNN (GIN) (EGO+) | 0.102 ± 0.003 | 80729 |
> > > > | DS-GNN (GIN) (EGO+) | 0.116 ± 0.009 | 95949 |
> > > >
> > > > __Q__: _“Highlighting the top three models (e.g. in Table 1) is highly misleading since only one version of each prior method is compared against 16 versions of the proposed idea. In general, this comparison does not make sense since the propose method has 16 trials to achieve a good result while all reference models only have a single trial.”_
> > > >
> > > > __A__: We thank the Reviewer for bringing this point to our attention. We agree on the fact that the current colouring scheme may be misleading, and we will revise Tables 1 and 3 for clarity in the next revision. We will retain the colors, but allowing for only one ESAN configuration to be considered amongst the three top-scoring methods. We still believe this comparison to be fair as other methods reported in the table have their own hyperparameters that have been tuned in their respective works. On the other hand, we think it is worth reporting the results for all configurations of ESAN, in view of the research questions we have stated at the beginning of Section 4. We welcome further suggestions from the Reviewer to improve the presentation of these results.
> > > >
> > > > __Q__: _“No code is provided to verify the results.”_
> > > >
> > > > __A__: We wanted to follow one of the recommended methods of sharing code in the ICLR FAQ (https://iclr.cc/Conferences/2022/AuthorGuide):
> > > >
> > > > > After we open the discussion forums for all submitted papers, make a comment directed to the reviewers and area chairs and put a link to an anonymous repository.
> > > >
> > > > Thus, we will shortly proceed by sharing a link to the code in a message addressed to all Reviewers only, along with the Area Chair. We will additionally do our best to make our implementation available to the broader community in libraries for Deep Learning on graphs, such as PyTorch Geometric and Deep Graph Library.

---

### Author Response · Authors · 2021-11-14
**Official Response**

We would like to express our gratitude to all reviewers for their thorough reviews and for sharing their insightful comments. We are glad to see that our work has been positively received in general, with reviewers appreciating several, diverse aspects of our contribution. They find the problem of _“strong academic and industrial importance”_ (B3oK) and our approach to be _“novel”_ (tGsP) and _“sound and significant”_ (rtwc). Additionally, they consider our theoretical analysis to be _“solid”_ (tGsP, rtwc), _“clear and sufficient”_ (B3oK), while recognizing the level of detail (Ebj3, rtwc) and the _“comprehensive experimental validation”_ (tGsP). Finally, we are delighted to notice reviewers appreciated the quality of the presentation (Ebj3, tGsP), reporting that _“the way of explanations and writing manner of the author are nearly perfect”_ (B3oK).

A claim of “lack of novelty”, in light of the work of Cotta et al., was raised by Reviewer B3oK. As Cotta et al. appeared on arXiv on 1 Oct 2021, __we see this as a clear case of contemporaneous work__, as defined in the ICLR reviewer guidelines: any work that appeared after June 5 is considered contemporaneous. While we will certainly cite the aforementioned work appropriately in our paper, it must not be the ground for rejection. We therefore kindly ask Reviewer B3oK to revise their score. Furthermore, as we discuss in our response to Reviewer B3oK, despite the overall related approach, our work has substantial differences and advantages compared to Cotta et al.

All reviewers have provided interesting and actionable feedback to improve the quality of our work. In order to address their specific comments, we have proceeded by responding to each reviewer separately. We welcome further questions and clarifications.

__New experiments__. Several additional experiments were carried out in response to the reviewer's suggestion: (1) RDT - a graph benchmark that contains larger graphs; (2) CSL - a benchmark for GNN expressivity; (3) ZINC12K - a widely used large scale molecular dataset. Across all these experiments, ESAN outperformed the baseline encoder, sometimes by a considerable margin. On ZINC12K, ESAN also outperformed all provably expressive, domain agnostic GNNs, along with other GNNs employing domain-specific structural information. Details are provided in the individual responses.

We will follow up by uploading a new version of our manuscript during the discussion period. Our updated manuscript will include our new experimental results, and make the revisions as mentioned in the responses to reviewers. As mentioned in our responses, we will also make our code available in an anonymous repository.

---

### Author Response · Authors · 2021-11-18
**New manuscript revision**

We would like to signal the upload of a new manuscript revision. This includes the changes anticipated in the previous general comment and throughout the responses to each single reviewer.

The revision includes the following main additions (reported in the paper appendix):
- An in-depth comparison with the concurrent work of _Cotta et al._;
- Results from new experiments on the following benchmarks: `CSL`, `RDT-B`, `ZINC12k`;
- An analysis on how the number of parameters impact the performance of the baseline methods;
- An overall, general discussion of our experimental results.

Changes in the revision are visually highlighted in red.

---

### Decision · Program_Chairs · 2022-01-20

**Decision:**

Accept (Spotlight)

**Comment:**

Improving the expressiveness of GNN is an important problem in the current graph learning community. Its key idea is to generate subgraphs from the original graph, then encode the subgraphs into the message passing process of GNN. The proposed method is proven to be strictly more powerful than 1-WL. The authors also quantize how design choices such as the subgraph selection policy and equivariant neural architecture can affect the architecture’s expressive power.

After the rebuttal, all reviewers are glad to accept this submission.

During the discussion, while reviewer B3oK has shown some concerns on the concurrent works in NeurIPS 2021, it should not affect the decision of the submission once the authors have discussed them in the main text. The authors have done this in their revision, thus an acceptance (spotlight) is suggested.